# Last-Iterate Convergent Policy Gradient Primal-Dual Methods for Constrained MDPs

**Dongsheng Ding**[*]
University of Pennsylvania
dongshed@seas.upenn.edu

**Chen-Yu Wei**[*]
University of Virginia
chenyu.wei@virginia.edu

**Kaiqing Zhang**[*]
University of Maryland, College Park
kaiqing@umd.edu

**Alejandro Ribeiro**
University of Pennsylvania
aribeiro@seas.upenn.edu

## Abstract

We study the problem of computing an optimal policy of an infinite-horizon discounted constrained Markov decision process (constrained MDP). Despite the popularity of Lagrangian-based policy search methods used in practice, the oscillation of policy iterates in these methods has not been fully understood, bringing out issues such as violation of constraints and sensitivity to hyper-parameters. To fill this gap, we employ the Lagrangian method to cast a constrained MDP into a constrained saddle-point problem in which max/min players correspond to primal/dual variables, respectively, and develop two single-time-scale policy-based primal-dual algorithms with non-asymptotic convergence of their policy iterates to an optimal constrained policy. Specifically, we first propose a regularized policy gradient primal-dual (RPG-PD) method that updates the policy using an entropy-regularized policy gradient, and the dual variable via a quadratic-regularized gradient ascent, simultaneously. We prove that the policy primal-dual iterates of RPG-PD converge to a regularized saddle point with a sublinear rate, while the policy iterates converge sublinearly to an optimal constrained policy. We further instantiate RPG-PD in large state or action spaces by including function approximation in policy parametrization, and establish similar sublinear last-iterate policy convergence. Second, we propose an optimistic policy gradient primal-dual (OPG-PD) method that employs the optimistic gradient method to update primal/dual variables, simultaneously. We prove that the policy primal-dual iterates of OPG-PD converge to a saddle point that contains an optimal constrained policy, with a linear rate. To the best of our knowledge, this work appears to be the first non-asymptotic policy last-iterate convergence result for single-time-scale algorithms in constrained MDPs. We further validate the merits and the effectiveness of our methods in computational experiments.

## 1 Introduction

Constrained Markov decision process (Constrained MDP) is the classical model for constrained dynamic systems in the early stochastic control literature (e.g., [1, 2, 3, 4, 5]) and the recent constrained reinforcement learning (RL) literature (e.g., [6, 7, 8, 9, 10, 11]). It is applicable to many constrained control problems by integrating other system specifications in constraints, and admits a natural extension of constrained optimization and Lagrangian in policy space. Lagrangian-based policy search methods, especially policy-based primal-dual methods that work simultaneously with

---

[*]Alphabetical order.

37th Conference on Neural Information Processing Systems (NeurIPS 2023).

primal/dual variables, lie at the heart of recent successes of constrained MDPs, e.g., navigation [12], autonomous driving [13, 14], robotics [15], and finance [16]; see [17, 18, 19, 20] for more examples.

Despite the popularity of policy-based primal-dual algorithms, classical asymptotic convergence assumes that primal-dual updates are in two-time-scale[1] type [21, 6, 22, 16, 9] (and/or work in two nested loops[2]), and considerable global non-asymptotic convergence guarantee is measured via an average of past objective/constraint functions [23, 24, 25, 26] or a mixture of past policies [27, 28]. These results are unfavorable in constrained dynamic systems, especially safety-critical ones, due to three reasons: (i) Average and mixture performance of non-asymptotic convergence conceals oscillating (or even overshooting) objective/constraint functions of immediate policy iterates [29, 30], and oscillation-incurred constraint violation impedes a policy iterate being optimal; (ii) Asymptotic convergence is not instructive, because arbitrarily slow convergence, and oscillation and overshoot in any finite time can happen; (iii) Two-time-scale algorithms including algorithms with nested loops are sensitive to hyper-parameters and are therefore typically difficult to tune [16, 9]. Thus, we ask the following question in constrained MDPs:

> Can the *policy iterates* of a *single-time-scale* policy-based primal-dual algorithm
> converge to an optimal constrained policy with *non-asymptotic rate*?

By "single-time-scale", we refer to the classical methods [31, 32, 33] that iterate primal/dual variables concurrently (with the same constant stepsize). Only partial answers to this question are provided in recent studies [34, 35, 36] since they either do not work in the single-time-scale scheme or they do not have non-asymptotic convergence guarantees. In this work, we provide an affirmative answer in two methodologies. First, we initiate the design and analysis of single-time-scale policy-based primal-dual algorithms via regularization, while previous works [24, 28, 34] rely on two-time-scale schemes. Second, inspired by convex minimax optimization [37, 38, 39], we propose a new optimistic policy gradient for a single-time-scale policy-based primal-dual algorithm that solves a class of non-convex minimax problem. While preparing our work, we noticed a contemporaneous work [40], which has empirically validated the effectiveness of other optimistic methods in constrained MDPs, has further inspired the pursuit of our contributions, as outlined in detail below.

**Contributions**. To compute an optimal policy of an infinite-horizon discounted constrained MDP, we employ the Lagrangian method to cast it into a constrained saddle-point problem in which max/min players correspond to primal/dual variables, propose two single-time-scale policy-based primal-dual algorithms, and prove global non-asymptotic convergence of their policy iterates.

- *Nearly dimension-free sublinear last-iterate policy convergence.* We propose a regularized policy gradient primal-dual (RPG-PD) method that updates the policy using an entropy-regularized policy gradient, and the dual using a quadratic-regularized gradient ascent, simultaneously. We prove that the policy primal-dual iterates of RPG-PD converge to a regularized saddle point with a sublinear rate, and the policy iterates converge to an optimal constrained policy sublinearly.

- *Sublinear last-iterate policy convergence with function approximation.* We generalize RPG-PD for constrained MDPs with large state/action spaces by including function approximation in policy parametrization. We prove that the policy primal-dual iterates of an inexact RPG-PD converge to a regularized saddle point with a sublinear rate, but up to a function approximation error, and the policy iterates converge sublinearly to an optimal constrained policy when the error is small.

- *Problem-dependent linear last-iterate policy convergence.* We propose an optimistic policy gradient primal-dual (OPG-PD) method that employs the optimistic gradient method to update the primal/dual, simultaneously. We prove that the policy primal-dual iterates of OPG-PD converge to a saddle point that contains an optimal constrained policy with a problem-dependent linear rate.

While last-iterate convergence is of importance in its own right, by adding proper conservatism in the constraint, both methods can ensure *no* constraint violation for the *last policy iterate*, which perhaps is best for safety-critical tasks [10, 20]. As far as we know, this work shows the first non-asymptotic and policy last-iterate convergence for single-time-scale algorithms in the constrained MDP literature. We further exhibit the merits and the effectiveness of our methods in experiments.

---

[1] One update has relatively very large/small (fast/slow) stepsize than the other.

[2] We view an algorithm with two nested gradient-based loops as a two-time-scale algorithm.

**Technical comparisons with prior art**. Although global asymptotic last-iterate convergence has been established for single-time-scale algorithms very recently [36, 40], and value-average or policy-mixture non-asymptotic convergence have been established for other algorithms [23, 24, 25, 27, 28, 41, 42, 26], these studies did not investigate global *non-asymptotic* and *last-iterate* convergence for *single-time-scale* algorithms. Our results not only strengthen these prior guarantees, but also set up a new framework for analyzing policy-based primal-dual algorithms via the distance of primal-dual iterates to a saddle point that contains an optimal constrained policy. Our RPG-PD and OPG-PD keep the simplicity of single-time-scale primal-dual methods and output a nearly-optimal policy in the last iterate, which is more convenient than the history-average policies [28, 24] or the policies from subroutines [34, 35]. Compared with the policy-based methods [40], our OPG-PD is a projected policy gradient method that enjoys policy last-iterate convergence with linear rate. Compared with the constrained saddle-point problems [38, 39], our minimax optimization that results from constrained MDP is *non-convex*. Hence, our OPG-PD extends the last-iterate convergence guarantee from convex minimax optimization to a class of non-convex ones, while preserving a linear rate. Compared with the analysis in the two-player zero-sum Markov game [43, 44], there is no reduction from constrained MDPs to per-state bilinear games. Please see more details in Appendix A.

## 2 Preliminaries

We consider an infinite-horizon discounted constrained Markov decision process [3, 5, 8] – CMDP ($S, A, P, r, u, b, \gamma, \rho$) – where $S$ and $A$ are state/action spaces, $P$ is a transition kernel that specifies the transition probability $P(s' \mid s, a)$ from state $s$ to next state $s'$ under action $a \in A$, $r$, $u$ : $S \times A \to [0, 1]$ are reward/utility functions, $b$ is a constraint threshold, $\gamma \in [0, 1)$ is a discount factor, and $\rho$ is an initial state distribution. A stationary stochastic policy $\pi : S \to \Delta(A)$ determines a probability distribution over the action space $A$ based on current state, i.e., $a_t \sim \pi(\cdot \mid s_t)$ at time $t$, where $\Delta(A)$ is a probability simplex over $A$. Let $\Pi$ be the set of all possible stochastic policies. A policy $\pi \in \Pi$, together with the initial state distribution $\rho$, induces a distribution over trajectories $\tau = \{(s_t, a_t, r_t, u_t)\}_{t=0}^{\infty}$, where $s_0 \sim \rho$, $a_t \sim \pi(\cdot \mid s_t)$, $r_t = r(s_t, a_t)$, $u_t = u(s_t, a_t)$ and $s_{t+1} \sim P(\cdot \mid s_t, a_t)$ for all $t \geq 0$.

Given a policy $\pi$, the value functions $V_r^\pi$, $V_u^\pi : S \to \mathbb{R}$ associated with the reward function $r$ or the utility function $u$ are given by the expected sums of discounted rewards or utilities under policy $\pi$:

$$V_r^\pi(s) \;:=\; \mathbb{E}\left[\sum_{t=0}^{\infty} \gamma^t r(s_t, a_t) \mid s_0 = s\right] \text{ and } V_u^\pi(s) \;:=\; \mathbb{E}\left[\sum_{t=0}^{\infty} \gamma^t u(s_t, a_t) \mid s_0 = s\right]$$

where the expectation $\mathbb{E}$ is over the randomness of the trajectory $\tau$ induced by $\pi$. Their expected values under $\rho$ are $V_r^\pi(\rho) := \mathbb{E}_{s \sim \rho}[V_r^\pi(s)]$ and $V_u^\pi(\rho) := \mathbb{E}_{s \sim \rho}[V_u^\pi(s)]$. It is useful to introduce the discounted state visitation distribution, $d_{s_0}^\pi(s) = (1 - \gamma) \sum_{t=0}^{\infty} \gamma^t \Pr(s_t = s \mid \pi, s_0)$ which adds up discounted probabilities of visiting $s$ in the execution of $\pi$ starting from $s_0$. Denote $d_\rho^\pi(s) := \mathbb{E}_{s_0 \sim \rho}[d_{s_0}^\pi(s)]$ and thus $d_\rho^\pi(s) \geq (1 - \gamma)\rho(s)$ for any $\rho$ and $s$. Furthermore, for the reward function $r$, we introduce the state-action value function $Q_r^\pi : S \times A \to \mathbb{R}$ when the agent begins with a state-action pair $(s, a)$ and follows a policy $\pi$, and the associated advantage function $A_r^\pi : S \times A \to \mathbb{R}$,

$$Q_r^\pi(s, a) \;:=\; \mathbb{E}\left[\sum_{t=0}^{\infty} \gamma^t r(s_t, a_t) \mid s_0 = s, a_0 = a\right] \text{ and } A_r^\pi(s, a) \;:=\; Q_r^\pi(s, a) - V_r^\pi(s).$$

Similarly, we define $Q_u^\pi : S \times A \to \mathbb{R}$ and $A_u^\pi : S \times A \to \mathbb{R}$ for the utility function $u$.

In this work, we aim to find a policy solution $\pi^\star$ of a constrained policy optimization problem,

$$\underset{\pi \in \Pi}{\text{maximize}} \;\; V_r^\pi(\rho) \qquad \text{subject to} \;\; V_u^\pi(\rho) \;\geq\; b \tag{1}$$

where the objective is the reward value function $V_r^\pi(\rho)$ and the constraint requires that the utility value function $V_u^\pi(\rho)$ is above a given threshold $b$. For notational simplicity we assume a single constraint, but our algorithms are readily generalizable to the problems with multiple constraints, as well as our subsequent last-iterate convergence theory. Since $V_r^\pi(\rho)$ and $V_u^\pi(\rho) \in [0, 1/(1 - \gamma)]$, we assume $b \in (0, 1/(1 - \gamma)]$ to avoid trivial cases. Let $g : S \times A \to [-1, 1]$ be $g := u - (1 - \gamma)b$. We, equivalently, translate the constraint $V_u^\pi(\rho) \geq b$ into $V_g^\pi(\rho) \geq 0$ that is our focal constraint. The optimal constrained policy $\pi^\star$ depends on the initial state distribution $\rho$; see Appendix B.1.

By the method of Lagrange multipliers [45], we dualize the constraint in (1) and present a standard Lagrangian $L(\pi, \lambda) := V_r^\pi(\rho) + \lambda V_g^\pi(\rho)$, where $\pi \in \Pi$ is the primal variable and $\lambda \in [0, \infty]$ is the dual variable or the Lagrangian multiplier. Introduction of the Lagrangian $L(\pi, \lambda)$ interprets Problem (1) as a max-min problem: $\text{maximize}_{\pi \in \Pi} \, \text{minimize}_{\lambda \in [0, \infty]} \, L(\pi, \lambda)$, and thus we can view the Lagrangian $L(\pi, \lambda)$ as a value function with a composite function $r + \lambda g$,

$$\text{maximize}_{\pi \in \Pi} \, \text{minimize}_{\lambda \in [0, \infty]} \, V_{r + \lambda g}^\pi(\rho). \tag{2}$$

However, it's defective to view Problem (2) as a standard MDP problem by fixing a dual variable $\lambda$, even the optimal one; also see [9, 46, 47, 30]. This is often referred to as the *scalarization fallacy* [46]; see Appendix B.2 for the detail. From the perspective of game theory, we instead view $V_{r + \lambda g}^\pi(\rho) : \Pi \times [0, \infty] \to \mathbb{R}$ as a payoff function for a two-player zero-sum game in which max-player is the policy $\pi \in \Pi$ and min-player is the dual variable $\lambda \in [0, \infty]$, and study its saddle points. To proceed, we assume feasibility for Problem (1) throughout our analysis.

**Assumption 1** (Feasibility). *There exists a policy $\bar{\pi} \in \Pi$ and $\xi > 0$ such that $V_g^{\bar\pi}(\rho) \geq \xi$.*

Feasibility mirrors the Slater condition in the duality analysis of constrained optimization [45]. It can be verified by solving an unconstrained MDP problem with respect to $V_g^\pi(\rho)$.

A saddle point $(\pi', \lambda')$ satisfies $V_{r + \lambda' g}^\pi(\rho) \leq V_{r + \lambda' g}^{\pi'}(\rho) \leq V_{r + \lambda g}^{\pi'}(\rho)$ for all $\pi \in \Pi$, $\lambda \in [0, \infty]$, or equivalently, $\pi'$ is the max-min point, i.e., $\pi' \in \text{argmax}_{\pi \in \Pi} \, V_{r + \lambda' g}^\pi(\rho)$ and $\lambda'$ is the min-max point, i.e., $\lambda' \in \text{argmin}_{\lambda \in [0, \infty]} \, V_{r + \lambda g}^{\pi'}(\rho)$. To view Problem (2) as a saddle-point problem, we denote $V_P^\pi(\rho) := \inf_{\lambda \in [0, \infty]} V_{r + \lambda g}^\pi(\rho)$ as the primal function which takes $V_r^\pi(\rho)$ when $V_g^\pi(\rho) \geq 0$ and $-\infty$ otherwise, and $V_D^\lambda(\rho) := \max_{\pi \in \Pi} V_{r + \lambda g}^\pi(\rho)$ as the dual function. Let an optimal dual variable be $\lambda^\star \in \text{argmin}_{\lambda \in [0, \infty]} V_D^\lambda(\rho)$. For Problem (1) under Assumption 1, strong duality holds in policy space [12, Theorem 3] and optimal dual variables are bounded [48, Lemma 3].

**Lemma 1** (Strong duality/Saddle point existence and boundedness). *Let Assumption 1 hold. Then, (i) strong duality holds for Problem (1), i.e., $V_P^{\pi^\star}(\rho) = V_D^{\lambda^\star}(\rho)$; (ii) optimal dual variables are bounded, i.e., $\lambda^\star \in [0, (V_r^{\pi^\star} - V_r^{\bar\pi})/\xi]$.*

Let the set of max-min points be $\Pi^\star := \text{argmax}_{\pi \in \Pi} \min_{\lambda \in [0, \infty]} V_{r + \lambda g}^\pi(\rho)$ and the set of min-max points be $\Lambda^\star := \text{argmin}_{\lambda \in [0, \infty]} \max_{\pi \in \Pi} V_{r + \lambda g}^\pi(\rho)$. From Lemma 1 (ii), $\Lambda^\star$ is contained in a bounded interval $\Lambda := [0, 1/((1 - \gamma)\xi)]$. Lemma 1 (i) shows that any pair $(\pi^\star, \lambda^\star) \in \Pi^\star \times \Lambda^\star$ solves the following constrained saddle-point problem,

$$\text{maximize}_{\pi \in \Pi} \, \text{minimize}_{\lambda \in \Lambda} \, V_{r + \lambda g}^\pi(\rho) = \text{minimize}_{\lambda \in \Lambda} \, \text{maximize}_{\pi \in \Pi} \, V_{r + \lambda g}^\pi(\rho). \tag{3}$$

Any saddle points associated with the set $\Lambda^\star$ are captured by Problem (3) due to the invariance of saddle points, and searching for any pair $(\pi^\star, \lambda^\star) \in \Pi^\star \times \Lambda^\star$ is sufficient by the interchangeability of saddle points; see Lemmas 8-9 in Appendix B.3 for the properties of saddle points. Thus, we view the policy (primal) as max-player and the dual as min-player in a zero-sum game.

Three structural properties from constrained MDPs distinguish Problem (3) from recent last-iterate convergence for learning in zero-sum games (e.g., [37, 38, 43, 39, 44]): (i) Two players are *asymmetric*. One plays a stochastic policy that affects the transition dynamics and the other selects an action in a continuous interval that only changes the payoff; (ii) Problem (3) is a *non-convex* game, because of the non-concavity of the payoff $V_{r + \lambda g}^\pi(\rho)$ in policy $\pi$ (e.g., [49, Lemma 1]); (iii) A saddle-point policy for Problem (3) cannot be *uniformly* max-min optimal, i.e., being optimal *across all states*, since an optimal policy often depends on the initial state distribution $\rho$ in a constrained MDP; see Appendix B.1. Hence, known last-iterate results in zero-sum convex games or symmetric Markov games that admit uniformly optimal policies can't be applied and new techniques are required to address this non-standard saddle-point problem, which warrants our contributions in this work.

**Warm-up: Indirect policy search in occupancy-measure space**. Finding a saddle point of a non-convex game is hard in general [50]. Nevertheless, Problem (1) can be rewritten as a linear program regarding the occupancy measure [5], which permits *indirectly* searching for a saddle point of a bilinear Lagrangian [51, 42, 36]. These asymptotic or average-iterate convergence results can be easily strengthened by applying last-iterate convergence results for bilinear games (e.g., [38, 39]) to be

non-asymptotic and last-iterate. By doing so, we state an optimistic primal-dual (OPD) method (18) in Appendix B.4. Compared with a contemporaneous work [40], OPD is free of projection to an occupancy measure set, and enjoys strengthened linear convergence.

OPD is an *indirect* policy search method that iterates using occupancy measure-based gradients, not policy-based gradients. It is crucial to develop *direct* policy search methods that are widely-used in RL, which is our focus. We propose two such methods in Section 3 and Section 4, respectively.

## 3 Policy Last-Iterate Convergence: Regularized Method

Towards achieving policy last-iterate convergence, a practical strategy is using regularization [52] to "convexify" Problem (3). We present a regularized method – Regularized Policy Gradient Primal-Dual (RPG-PD) – that converges to a saddle point that yields an optimal constrained policy.

### 3.1 Regularized policy gradient primal-dual method

We introduce a regularized Lagrangian $L_\tau(\pi, \lambda) := V_{r+\lambda g}^\pi(\rho) + \tau(\mathcal{H}(\pi) + \frac{1}{2}\lambda^2)$ by adding a regularization term $\mathcal{H}(\pi) + \frac{1}{2}\lambda^2$ onto the original Lagrangian $V_{r+\lambda g}^\pi(\rho)$, where $\tau$ is a regularization parameter, and $\mathcal{H}(\pi) := \mathbb{E}[\sum_{t=0}^\infty -\gamma^t \log \pi(a_t \mid s_t)]$ is an entropy-like regularization term [52]. We now introduce a regularized constrained saddle-point problem,

$$\underset{\pi \in \Pi}{\text{maximize}} \; \underset{\lambda \in \Lambda}{\text{minimize}} \;\; L_\tau(\pi, \lambda) \;\; = \;\; \underset{\lambda \in \Lambda}{\text{minimize}} \; \underset{\pi \in \Pi}{\text{maximize}} \;\; L_\tau(\pi, \lambda). \tag{4}$$

Problem (4) is well-defined, since there exists a saddle point for $L_\tau(\pi, \lambda)$ over $\Pi \times \Lambda$ and it is unique; see Appendix C.1 for proof. A saddle point $(\pi_\tau^\star, \lambda_\tau^\star)$, i.e., $\pi_\tau^\star = \text{argmax}_{\pi \in \Pi} \min_{\lambda \in \Lambda} L_\tau(\pi, \lambda)$ and $\lambda_\tau^\star = \text{argmin}_{\lambda \in \Lambda} \max_{\pi \in \Pi} L_\tau(\pi, \lambda)$, satisfies a sandwich-like property,

$$V_{r+\lambda_\tau^\star g}^\pi(\rho) - \tau\mathcal{H}(\pi_\tau^\star) \;\leq\; V_{r+\lambda_\tau^\star g}^{\pi_\tau^\star}(\rho) \;\leq\; V_{r+\lambda g}^{\pi_\tau^\star}(\rho) + \frac{\tau}{2}\lambda^2 \;\text{ for all } (\pi, \lambda) \in \Pi \times \Lambda \tag{5}$$

that states that $(\pi_\tau^\star, \lambda_\tau^\star)$ is a saddle point of the original Lagrangian $V_{r+\lambda g}^\pi(\rho)$, up to two $\tau$-terms. We thus propose a regularized policy gradient primal-dual (RPG-PD) method by maintaining a sequence for policy and dual variables each: $\{\pi_t\}_{t \geq 0}$ for the policy-player, and $\{\lambda_t\}_{t \geq 0}$ for the dual-player,

$$\pi_{t+1}(\cdot \mid s) \;\; = \;\; \underset{\pi(\cdot \mid s) \in \hat{\Delta}(A)}{\text{argmax}} \left\{ \sum_a \pi(a \mid s) Q_{r+\lambda_t g + \tau\psi_t}^{\pi_t}(s, a) \;-\; \frac{1}{\eta} \text{KL}(\pi(\cdot \mid s), \pi_t(\cdot \mid s)) \right\} \tag{6a}$$

$$\lambda_{t+1} \;\; = \;\; \underset{\lambda \in \Lambda}{\text{argmin}} \left\{ \lambda\Big( V_g^{\pi_t}(\rho) + \tau\lambda_t \Big) \;+\; \frac{1}{2\eta}(\lambda - \lambda_t)^2 \right\}, \tag{6b}$$

where the gradient direction $Q_{r+\lambda_t g + \tau\psi_t}^{\pi_t}(s, a)$ is the state-action value function under a composite function $r + \lambda_t g + \tau\psi_t$ in which $\psi_t(s, a) := -\log \pi_t(a \mid s)$, $\text{KL}(p, p') := \sum_a p_a \log \frac{p_a}{p'_a}$ is the Kullback–Leibler (KL) divergence, $\hat{\Delta}(A) := \{\pi(\cdot \mid s) \in \Delta(A) \mid \pi(a \mid s) \geq \frac{\epsilon_0}{|A|}, a \in A\}$ is a restricted probability simplex set with parameter $\epsilon_0 \in (0, 1)$, $\eta$ is the stepsize, and $(\pi_0(\cdot \mid s), \lambda_0) \in \hat{\Delta}(A) \times \Lambda$ is an initial point. Projecting the policy iterate to the simplex set $\hat{\Delta}(A)$ ensures the boundedness of the gradient. Primal update (6a) works as the classical mirror descent with KL divergence [53] with a projection onto the set $\hat{\Delta}(A)$. Dual update (6b) performs typical projected gradient descent. Hence, RPG-PD is a single-time-scale method. RPG-PD simplifies the two-time-scale method [34] to be single-time-scale and generalize the single-time-scale methods [23, 54] with regularization.

### 3.2 Policy last-iterate convergence

In Theorem 2, we show that the primal-dual iterates of RPG-PD converge in the last iterate; see Appendix C.2 for proof. We characterize the convergence via a distance metric $\Phi_t = \text{KL}_t(\rho) + \frac{1}{2}(\lambda_\tau^\star - \lambda_t)^2$, where $\text{KL}_t(\rho) := (1/(1-\gamma)) \sum_s d_\rho^{\pi_\tau^\star}(s) \text{KL}_t(s)$ and $\text{KL}_t(s) := \text{KL}(\pi_\tau^\star(\cdot \mid s), \pi_t(\cdot \mid s))$.

**Theorem 2** (Linear convergence of RPG-PD). *Let Assumption 1 hold. If we set the stepsize $\eta \leq 1/C_{\tau,\xi,\epsilon_0}$, then the primal-dual iterates of RPG-PD (6) satisfy*

$$\Phi_{t+1} \;\; \leq \;\; e^{-\eta\tau t} \Phi_1 \;+\; \frac{\eta}{\tau} \max\left( (C_{\tau,\xi,\epsilon_0})^2, (C'_{\tau,\xi})^2 \right)$$

*where $C_{\tau,\xi,\epsilon_0} := (1 + 1/((1-\gamma)\xi) + \tau \log|A|)/(1-\gamma) - \tau \log(\epsilon_0/|A|)$, $C'_{\tau,\xi} := (1 + \tau/\xi)/(1-\gamma)$.*

Theorem 2 states that the primal-dual iterates of RPG-PD converge to a neighborhood of $(\pi_\tau^\star, \lambda_\tau^\star)$ in a linear rate. The size of neighborhood scales with $\eta/\tau + \eta\tau(1 + \log^2 \epsilon_0)$ and the convergence rate is $\eta\tau$. Even if $\epsilon_0$ is very small, the $\log \epsilon_0$-term is almost a constant. If we take $\eta = \min(\epsilon\tau, 1/C_{\tau,\xi,\epsilon_0})$ and $\epsilon_0 = \epsilon$, then after $O(1/\epsilon)$ iterations the RPG-PD's primal-dual iterate $(\pi_t, \lambda_t)$ is $\epsilon$-close to $(\pi_\tau^\star, \lambda_\tau^\star)$, i.e., $\Phi_t = O(\epsilon)$ for any $t \geq (1/(\epsilon\tau^2))\log(1/\epsilon)$. For small $\tau$, we can translate the policy convergence for the value functions in Corollary 3; see Appendix C.3 for proof.

**Corollary 3** (Nearly-optimal constrained policy). *Let Assumption 1 hold. For small $\epsilon > 0$, if we take $\eta = \Theta(\epsilon^4)$, $\tau = \Theta(\epsilon^2)$, and $\epsilon_0 = \epsilon$, then the policy iterates of RPG-PD (6) satisfy*

$$V_r^{\pi^\star}(\rho) - V_r^{\pi_t}(\rho) \ \leq \ \epsilon \ \ \text{and} \ \ -V_g^{\pi_t}(\rho) \ \leq \ \epsilon \ \ \text{for any } t = \Omega\left(\frac{1}{\epsilon^6}\log^2\frac{1}{\epsilon}\right)$$

*where $\Omega(\cdot)$ only has some problem-dependent constant.*

Corollary 3 states that the last policy iterate of RPG-PD is an $\epsilon$-optimal policy for Problem (1) after $\Omega(1/\epsilon^6)$ iterations. Compared with the single-time-scale methods [23, 54], RPG-PD improves the convergence from average-value (or regret-type) to *last policy iterate*. Not just being theoretically stronger, the last-iterate convergence is more appealing since it captures the stability of trajectories of an algorithm [29, 40]. Compared with the two-time-scale methods [28, 24, 34, 35], RPG-PD is free of nested loops, and uniform ergodicity and exploratory initial state distribution. We notice that the dual methods [28, 24] yield history-average policies and the dual methods [34, 35] return policies from a subroutine. In contrast, RPG-PD outputs a nearly-optimal policy in the last iterate, the first-of-its-kind in the constrained MDP literature, albeit the rate is worse than the average ones [23, 54].

To get zero constraint violation, i.e., $V_g^{\pi_t}(\rho) \geq 0$ at some $t$, it is straightforward to employ a conservative constraint $V_{g'}^\pi(\rho) \geq 0$ with $g' := g - (1-\gamma)\delta$ for some $\delta > 0$. When $\epsilon$ is small enough, there always exists some $\delta$ such that the policy iterates of RPG-PD (6) satisfy $V_r^{\pi^\star}(\rho) - V_r^{\pi_t}(\rho) \leq \epsilon$ and $V_g^{\pi_t}(\rho) \geq 0$ for large $t$; see Appendix C.4 for proof. Our zero constraint violation ensures the last policy iterate of RPG-PD to satisfy the constraint, which is not the zero average constraint violation in the episodic setting [55, 56]. Compared with the zero constraint violation of a policy induced by an average of past occupancy measures [42], RPG-PD's zero constraint violation directly settles the policy iterates down, which appears to be the first policy-based zero constraint violation.

Last but not least, the iteration complexity of RPG-PD is nearly-free of the MDP dimension, except for an $\log|A|$-term, which inherits the dimension-free property of the NPG methods [49, 57, 23]. Hence, it is ready to view RPG-PD as a variant of NPG methods and generalize RPG-PD to constrained MDPs with large state spaces in the function approximation setting.

### 3.3 Linear function approximation case

To deal with large state spaces, we use a parametrized policy $\pi_\theta$ with $\theta \in \mathbb{R}^d$ for RPG-PD (6) without restricting $\Delta(A)$, where $d$ is much smaller than the size of state/action spaces. To introduce function approximation, we begin with a tabular softmax policy $\pi_\theta(a\,|\,s) = \frac{\exp(\theta_{s,a})}{\sum_{a'}\exp(\theta_{s,a'})}$ for all $(s,a) \in S \times A$ and $\theta \in \mathbb{R}^{|S||A|}$. Connecting NPG to mirror descent [49, 58], we express RPG-PD (6) as a NPG method with the following update; see Appendix C.5 for proof,

$$\theta_{t+1} \ = \ \theta_t + \eta\,(1-\gamma)F_\rho(\theta_t)^\dagger \cdot \nabla_\theta L_\tau(\pi_{\theta_t}, \lambda_t) \tag{7a}$$

$$\lambda_{t+1} \ = \ \mathcal{P}_\Lambda\left((1-\eta\tau)\lambda_t - \eta V_g^{\pi_{\theta_t}}(\rho)\right) \tag{7b}$$

where $F_\rho(\theta_t)^\dagger \cdot \nabla_\theta L_\tau(\pi_{\theta_t}, \lambda_t)$ is a NPG direction, and $F_\rho(\theta)$ is the Fisher information matrix for a policy $\pi_\theta$, i.e., $F_\rho(\theta) := \mathbb{E}_{s \sim d_\rho^{\pi_\theta}}\mathbb{E}_{a \sim \pi_\theta(\cdot\,|\,s)}[\nabla_\theta \log \pi_\theta(a\,|\,s)(\nabla_\theta \log \pi_\theta(a\,|\,s))^\top]$. A useful property of (7a) is that NPG can be related to a linear regression. For any policy $\pi_\theta$ and a state-action value function $Q^{\pi_\theta}$, the associated compatible function approximation error is $E_Q(w, \theta, \nu) := \mathbb{E}_{(s,a) \sim \nu}[(w^\top \nabla \log \pi_\theta(a\,|\,s) - Q^{\pi_\theta}(s,a))^2]$, where $\nu(s,a) = d_\rho^{\pi_\theta}(s)\pi_\theta(a\,|\,s)$ is a state-action distribution. It is known that (7a) is equivalent to $\theta_{t+1} = \theta_t + \eta w_t^\star$, where $w_t^\star \in \operatorname{argmin}_{w \in \mathbb{R}^d} E_Q(w, \theta_t, \nu_t)$ in which $Q^{\pi_{\theta_t}}(s,a) = Q_{r+\lambda_t g + \tau\psi_t}^{\pi_{\theta_t}}(s,a)$ and $\nu_t(s,a) = d_\rho^{\pi_{\theta_t}}(s)\pi_{\theta_t}(a\,|\,s)$ (e.g., [59, Lemma 1]). In practice, only an approximate minimizer $w_t^\star$ is available if a sample-based algorithm is used, e.g., $w_t \approx \operatorname{argmin}_{\|w\| \leq W} E_Q(w, \theta_t, \nu_t)$, where $W > 0$.

A useful generalization of the softmax policy to large state spaces is the log-linear policy based on linear function approximation. Let $\phi_{s,a} \in \mathbb{R}^d$ be a feature map with $\|\phi_{s,a}\| \leq 1$ for each state-action pair $(s,a)$. A log-linear policy $\pi_\theta \colon S \to \Delta(A)$ is parametrized by a parameter $\theta \in \mathbb{R}^d$,

$$\pi_\theta(a \mid s) = \frac{\exp(\phi_{s,a}^\top \theta)}{\sum_{a'} \exp(\phi_{s,a'}^\top \theta)} \text{ for all } (s,a) \in S \times A$$

which takes the tabular softmax policy as a special case, i.e., $\phi_{s,a}$ is an indicator function. We notice that $\nabla_\theta \log \pi_\theta(a \mid s) = \phi_{s,a} - \mathbb{E}_{a' \sim \pi_\theta(\cdot \mid s)}[\phi_{s,a'}]$. Since the log-linear policy is invariant to any action-independent term, it is convenient to replace $\nabla_\theta \log \pi_\theta(a \mid s)$ by $\phi_{s,a}$ and we introduce a simplified compatible function approximation error, $\mathcal{E}_Q(w, \theta, \nu) := \mathbb{E}_{(s,a) \sim \nu}[(\phi_{s,a}^\top w - Q^{\pi_\theta}(s,a))^2]$. Thus, we can take $w_t^\star \in \operatorname{argmin}_{w \in \mathbb{R}^d} \mathcal{E}_Q(w, \theta_t, \nu_t)$ in which $Q^{\pi_{\theta_t}}(s,a) = Q_{r+\lambda_t g+\tau\psi_t}^{\pi_{\theta_t}}(s,a)$ and $\nu_t(s,a) = d_\rho^{\pi_{\theta_t}}(s)\pi_{\theta_t}(a \mid s)$ to update $\theta_{t+1} = \theta_t + \eta w_t^\star$. Using the log-linear policy class, we replace the primal gradient direction of RPG-PD (6) by its linear function approximation $\phi_{s,a}^\top w_t^\star$,

$$\pi_{\theta_{t+1}}(\cdot \mid s) = \operatorname*{argmax}_{\pi(\cdot \mid s) \in \hat{\Delta}(A)} \left\{ \sum_a \pi(a \mid s)\phi_{s,a}^\top w_t^\star - \frac{1}{\eta} \mathrm{KL}(\pi(\cdot \mid s), \pi_{\theta_t}(\cdot \mid s)) \right\} \tag{8}$$

which, together with Dual update (6b), leads to a general version of RPG-PD. The set $\hat{\Delta}(A)$ ensures bounded true gradient direction $Q_{r+\lambda_t g+\tau\psi_t}^{\pi_{\theta_t}}(s,a)$. When there is no function approximation error, (8) reduces to Primal update (6a). In practice, we can only compute $w_t^\star$ approximately via

$$w_t \approx \operatorname*{argmin}_{\|w\| \leq W} \mathcal{E}_Q(w, \theta_t, d_{t,\nu})$$

which leads to an inexact RPG-PD: Primal update (8) in which $w_t^\star$ is replaced by $w_t$ and Dual update (6b), where $d_{t,\nu} := (1-\gamma)\mathbb{E}_{(s_0,a_0)\sim\nu}\sum_{t=0}^\infty \gamma^t \mathrm{Pr}(s_t = s, a_t = a \mid \pi_{\theta_t}, s_0, a_0)$ is a state-action distribution starting from any distribution $\nu$. Noticeably, $d_{t,\nu}$ is more general than $\nu_t$. To control the function approximation error, we divide $\mathcal{E}_Q(w_t, \theta_t, d_{t,\nu})$ into a statistical error $\mathcal{E}_Q(w_t, \theta_t, d_{t,\nu}) - \mathcal{E}_Q(w_t^\star, \theta_t, d_{t,\nu})$ that is similar to the excess risk in supervised learning, and an approximation error $\mathcal{E}_Q(w_t^\star, \theta_t, d_{t,\nu})$ that captures how well a linear function $(w_t^\star)^\top \phi_{s,a}$ approximates the true value function under $d_{t,\nu}$. If the on-policy distribution $d_{t,\nu}$ in $\mathcal{E}_Q(w_t^\star, \theta_t, d_{t,\nu})$ is replaced by $\nu^\star(s,a) = d_\rho^{\pi_\tau^\star}(s)\mathrm{Unif}_A(a)$, we define a transfer error $\mathcal{E}_Q(w_t^\star, \theta_t, \nu^\star)$. Let the covariance matrix of $\phi_{s,a}$ in any state-action distribution $\nu$ be $\Sigma_\nu := \mathbb{E}_{(s,a)\sim\nu}[\phi_{s,a}\phi_{s,a}^\top]$, and the relative condition number between $\Sigma_\nu$ and $\Sigma_{\nu^\star}$ be $\kappa_\nu := \max_{w \in \mathbb{R}^d} \frac{w^\top \Sigma_{\nu^\star} w}{w^\top \Sigma_\nu w}$.

We make an assumption on the statistical error, the transfer error, and the relative condition number.

**Assumption 2.** *(i) There exist $\epsilon_{\mathrm{stat}}, \epsilon_{\mathrm{bias}} > 0$ such that $\mathbb{E}[\mathcal{E}_Q(w_t, \theta_t, d_{t,\nu}) - \mathcal{E}_Q(w_t^\star, \theta_t, d_{t,\nu})] \leq \epsilon_{\mathrm{stat}}$ and $\mathbb{E}[\mathcal{E}_Q(w_t^\star, \theta_t, \nu^\star)] \leq \epsilon_{\mathrm{bias}}$; (ii) The relative condition number is finite, i.e., $\kappa_\nu < \infty$.*

We assess the convergence of inexact RPG-PD via the distance metric $\mathbb{E}[\Phi_t] := \mathbb{E}[\mathrm{KL}_t(\rho)] + \frac{1}{2}\mathbb{E}[(\lambda_\tau^\star - \lambda_t)^2]$, where the expectation $\mathbb{E}$ is over the randomness of computing $w_t$ via a sample-based algorithm. We state the convergence in Theorem 4 and delay its proof to Appendix C.6.

**Theorem 4** (Linear convergence of inexact RPG-PD). *Let Assumptions 1–2 hold. If we take the stepsize $\eta \leq 1/C_W$, then the primal-dual iterates of inexact RPG-PD satisfy*

$$\mathbb{E}[\Phi_{t+1}] \leq e^{-\eta\tau t}\mathbb{E}[\Phi_1] + \frac{\eta}{\tau}\max\left((C_W)^2, (C_{\tau,\xi}')^2\right) + \frac{2}{\tau}\left(\sqrt{|A|\epsilon_{\mathrm{bias}}} + \sqrt{|A|\kappa_\nu\epsilon_{\mathrm{stat}}}\right)$$

*where $C_W := 2W/(1-\gamma)$ and $C_{\tau,\xi}' := (1+\tau/\xi)/(1-\gamma)$.*

Theorem 4 states that the primal-dual iterates of inexact RPG-PD converge to a neighborhood of $(\pi_\tau^\star, \lambda_\tau^\star)$ in a linear rate. The convergence rate is $\eta\tau$ and the size of neighborhood scales with a sum of an $\eta/\tau$-term and an $1/\tau$-term that amplifies the effect of function approximation ($\epsilon_{\mathrm{stat}}, \epsilon_{\mathrm{bias}}$). We note that, Theorem 4 does not require the strong duality in the parametrized policy class, and the function approximation error includes the duality gap caused by the inexpensiveness of function class and the policy representation error caused by the restricted policy set $\hat{\Delta}(A)$. When there is no function approximation error, Theorem 4 has a similar result as Theorem 2. It is important to control

$(\epsilon_{\text{stat}}, \epsilon_{\text{bias}})$ to be small: (i) Application of stochastic gradient methods to the linear regression leads to $\epsilon_{\text{stat}} = O(1/\sqrt{K})$ or $O(1/K)$, where $K$ is the number of gradient steps, and thus, it is easy to control $\epsilon_{\text{stat}}$; (ii) When $\epsilon_0$ is very small, the parametrized policy iterate can be contained in $\hat{\Delta}(A)$, and thus $\epsilon_{\text{bias}}$ becomes zero in some cases, e.g., tabular softmax case [49] or low-rank MDPs [60, 61] with $d \geq |A|$; it can be made very small if the function class is rich, e.g., wide neural networks [62]. When the errors are small, it is ready to establish Corollary 5; see Appendix C.7 for proof.

**Corollary 5** (Nearly-optimal constrained policy). *Let Assumptions 1–2 hold and $\epsilon_{\text{stat}}, \epsilon_{\text{bias}} = O(\epsilon^8)$ for small $\epsilon$, $\epsilon_0 > 0$. If we take the stepsize $\eta = \Theta(\epsilon^4)$ and $\tau = \Theta(\epsilon^2)$, then the policy iterates of inexact RPG-PD satisfy*

$$\mathbb{E}\left[ V_r^{\pi^\star}(\rho) - V_r^{\pi_{\theta_t}}(\rho) \right] \ \leq \ \epsilon \ \text{ and } \ \mathbb{E}\left[ -V_g^{\pi_{\theta_t}}(\rho) \right] \ \leq \ \epsilon \ \text{ for any } t = \Omega\left( \frac{1}{\epsilon^6} \log^2 \frac{1}{\epsilon} \right)$$

*where $\Omega(\cdot)$ only has some problem-dependent constant.*

Corollary 5 states that the iteration complexity in Corollary 3 holds in the function approximation case. When $\epsilon$ is small enough, we can design a conservative constraint such that the policy iterates of inexact RPG-PD satisfy $V_r^{\pi^\star}(\rho) - V_r^{\pi_{\theta_t}}(\rho) \leq \epsilon$ and $V_g^{\pi_{\theta_t}}(\rho) \geq 0$ for large $t$; see Appendix C.8 for proof. Compared with the zero average constraint violation [63], this appears to be the first policy-based zero constraint violation result in the function approximation setting. Moreover, we extend inexact RPG-PD to be a sample-based algorithm and provide its sample complexity in Appendix C.9.

## 4 Policy Last-Iterate Convergence: Optimistic Method

Having established sublinear policy last-iterate convergence via regularization, we turn to the optimistic gradient method [37] for a faster rate. We propose an optimistic method – Optimistic Policy Gradient Primal-Dual (OPG-PD) – that converges an optimal constrained policy at a linear rate.

### 4.1 Optimistic policy gradient primal-dual method

We propose an optimistic policy gradient primal-dual (OPG-PD) method by maintaining two sequences for policy and dual variables each: $\{\pi_t\}_{t \geq 1}$ and $\{\hat{\pi}_t\}_{t \geq 1}$ for the policy-player, and $\{\lambda_t\}_{t \geq 1}$ and $\{\hat{\lambda}_t\}_{t \geq 1}$ for the dual-player,

$$
\begin{aligned}
\pi_t(\cdot \,|\, s) &= \operatorname*{argmax}_{\pi(\cdot \,|\, s) \in \Delta(A)} \left\{ \sum_a \pi(a \,|\, s) Q_{r+\lambda_{t-1}g}^{\pi_{t-1}}(s,a) - \frac{1}{2\eta} \left\| \pi(\cdot \,|\, s) - \hat{\pi}_t(\cdot \,|\, s) \right\|^2 \right\} \\
\hat{\pi}_{t+1}(\cdot \,|\, s) &= \operatorname*{argmax}_{\pi(\cdot \,|\, s) \in \Delta(A)} \left\{ \sum_a \pi(a \,|\, s) Q_{r+\lambda_t g}^{\pi_t}(s,a) - \frac{1}{2\eta} \left\| \pi(\cdot \,|\, s) - \hat{\pi}_t(\cdot \,|\, s) \right\|^2 \right\}
\end{aligned}
\tag{9a}
$$

$$
\begin{aligned}
\lambda_t &= \operatorname*{argmin}_{\lambda \in \Lambda} \left\{ \lambda \, V_g^{\pi_{t-1}}(\rho) + \frac{1}{2\eta} (\lambda - \hat{\lambda}_t)^2 \right\} \\
\hat{\lambda}_{t+1} &= \operatorname*{argmin}_{\lambda \in \Lambda} \left\{ \lambda \, V_g^{\pi_t}(\rho) + \frac{1}{2\eta} (\lambda - \hat{\lambda}_t)^2 \right\}
\end{aligned}
\tag{9b}
$$

where $\eta$ is the stepsize and $(\hat{\pi}_0, \hat{\lambda}_0) = (\pi_0, \lambda_0) \in \Pi \times \Lambda$ is the initial point. OPG-PD concurrently works with two primal iterates and two dual iterates, and each two are updated consecutively to stabilize the algorithm dynamics. The "optimistic" in optimization, e.g., [64] views $(\hat{\pi}_{t+1}, \hat{\lambda}_{t+1})$-update as a real policy gradient step and $(\pi_t, \lambda_t)$-update as a prediction step that generates an intermediate iterate $(\pi_t, \lambda_t)$. Not policy gradient at $(\hat{\pi}_t, \hat{\lambda}_t)$, the real step uses a policy gradient at $(\pi_t, \lambda_t)$ from prediction, exhibiting the optimism towards the prediction. Specifically, Primal update (9a) works as the projected $Q$-ascent [58, 65], an application of the classical mirror descent with Euclidean distance [53], where the projection onto a probability simplex can be solved efficiently [66]. Dual update (9b) performs standard projected gradient descent. We note that OPG-PD is different from the one-step multiplicative weights update in the policy-based ReLOAD [40].

When there is no MDP transition dynamics, i.e., constrained bandit [40], last-iterate convergence of OPG-PD to a saddle point is known in the minimax optimization [67, 68, 38, 39], because Problem (3) reduces to a bilinear zero-sum game in this case. However, it is prohibitive to apply such

bilinear game results to the Lagrangian $V^\pi_{r+\lambda g}(s)$ in every state $s$, as has been done for zero-sum Markov games [43, 44]. The main reason for this is that there may not exist an optimal constrained policy that is uniformly optimal across all states; see Appendix B.2.

## 4.2 Policy last-iterate convergence

We define the distribution mismatch coefficient over $\rho$ as $\kappa_\rho := \sup_{\pi \in \Pi} \left\| d^\pi_\rho / \rho \right\|_\infty$, which is the maximum distribution mismatch of policy $\pi$ relative to $\rho$, where $d^\pi_\rho / \rho$ is divided per state. Hence, $\left\| d^\pi_\rho / d^{\pi^\star}_\rho \right\|_\infty \leq \kappa_\rho / (1 - \gamma)$ for any policy $\pi \in \Pi$ and $\kappa_\rho \leq 1 / \rho_{\min}$ where $\rho_{\min} := \min_s \rho(s)$. The projection operator $\mathcal{P}_X$ on a closed convex set $X$ defines $\mathcal{P}_X(x) := \text{argmin}_{x' \in X} \|x' - x\|$.

We state the policy last-iterate convergence of OPG-PD (9) in Theorem 6.

**Theorem 6** (Linear convergence of OPG-PD). *Let Assumption 1 hold. Assume the optimal state visitation distribution be unique, i.e., $d^\pi_\rho = d^{\pi^\star}_\rho$ for any $\pi \in \Pi^\star$, and $\rho_{\min} > 0$. If we set the stepsize $\eta \leq \min\left(1/(4\sqrt{\iota}), (1 - \gamma)^3/(4|A|), (1 - \gamma)^3/(2\kappa_\rho)\right)$, where $\iota > 0$ is defined in Appendix D.1, then the primal-dual iterates of OPG-PD (9) satisfy*

$$\frac{1}{2(1 - \gamma)} \sum_s d^{\pi^\star}_\rho(s) \|\mathcal{P}_{\Pi^\star}(\hat{\pi}_t(\cdot \mid s)) - \hat{\pi}_t(\cdot \mid s)\|^2 + \frac{1}{2}(\mathcal{P}_{\Lambda^\star}(\hat{\lambda}_t) - \hat{\lambda}_t)^2 \leq \left(\frac{1}{1+C}\right)^t$$

*where $C := \min(7(1 - \gamma)/8, 7\eta^2(1 - \gamma)^2(C_{\rho,\xi})^2 \rho_{\min}/(6\kappa_{\rho,\gamma})^2)$ in which $C_{\rho,\xi}$ and $\kappa_{\rho,\gamma}$ are given by $C_{\rho,\xi} := c\rho_{\min}/(2\sqrt{|S||A|})/(1 + 1/((1 - \gamma)\xi))$, $\kappa_{\rho,\gamma} := \max(\kappa_\rho/(1 - \gamma), 1)$, and $c > 0$ is a problem-dependent constant from Lemma 26.*

Theorem 6 states that the primal-dual iterates of OPG-PD converge to $\Pi^\star \times \Lambda^\star$ in a linear rate, or putting it differently, (9) is contracting to a set of optimal primal/dual variables. The rate is governed by a problem-dependent constant. Proof of Theorem 6 is provided in Appendix D. A key to our analysis is to bridge the per-state policy gradient update and the policy improvement for $V^\pi_{r+\lambda g}(\rho)$ that is non-convex in policy $\pi$, which departs from the convex last-iterate analysis [38, 39]. In addition, we address two technical difficulties. First, the lack of uniformly optimal policies prevents learning an optimal policy from per-state bilinear games in zero-sum Markov games [43, 44]. Instead, we characterize the proximity of primal-dual iterates to a saddle point supported by an optimal state visitation distribution $d^{\pi^\star}_\rho$. Second, Problem (3) is an asymmetric game since one plays a stochastic policy over a finite set of discrete actions and controls the transition dynamics, but the other selects an action in a continuous interval. Thus, our dual-player analysis handles the long-term effect of the policy-player, which did not appear in the symmetric game [43, 44].

A direct corollary of Theorem 6 is stated below; see Appendix D.3 for the proof.

**Corollary 7** (Nearly-optimal constrained policy). *Let Assumption 1 hold and the optimal state visitation distribution be unique, i.e., $d^\pi_\rho = d^{\pi^\star}_\rho$ for any $\pi \in \Pi^\star$, and $\rho_{\min} > 0$. If we use the stepsize $\eta$ from Theorem 6, then the policy iterates of OPG-PD (9) satisfy*

$$V^{\pi^\star}_r(\rho) - V^{\hat{\pi}_t}_r(\rho) \leq \epsilon \text{ and } -V^{\hat{\pi}_t}_g(\rho) \leq \epsilon \text{ for any } t = \Omega\left(\log^2 \frac{1}{\epsilon}\right)$$

*where $\Omega(\cdot)$ only has some problem-dependent constant.*

Corollary 7 states that the last policy iterate of OPG-PD is an $\epsilon$-optimal policy for Problem (1) after an almost constant number of iterations, which improves the sublinear rate in Corollary 3. OPG-PD also improves the average-value convergence of the single-time-scale methods [23, 54] and the two-time-scale methods [28, 24, 34, 35], and matches the last-iterate convergence rate of the two-time-scale methods [34, 35]. We stress that our last-iterate convergence indicates the stability of whole primal-dual iterates, which is not the last policy iterate from a subroutine [34, 35]. Again, when $\epsilon$ is small, we can design a conservative constraint such that the policy iterates of OPG-PD satisfy $V^{\pi^\star}_r(\rho) - V^{\hat{\pi}_t}_r(\rho) \leq \epsilon$ and $V^{\hat{\pi}_t}_g(\rho) \geq 0$ for large $t$; see Appendix D.4 for the proof.

## 5 Computational Experiment

We validate the effectiveness of RPG-PD (6) and OPG-PD (9) by comparing them with typical primal-dual methods in Figure 1. A few observations are in order. The initial oscillation of RPG-PD

(**– –**) is damped, and OPG-PD (**—**) is almost free of oscillation as PID Lagrangian (····). However, oscillation of NPG-PD (**–·–**) causes its last-iterate policy violating the constraint $V_g^\pi(\rho) \geq 0$. OPG-PD (**—**) reaches the maximum reward value in four methods and RPG-PD (**– –**) converges to a slightly smaller value due to regularization, while both meet the constraint at the end. However, PID Lagrangian (····) is highly sub-optimal. Hence, our methods OPG-PD and RPG-PD can overcome oscillation and approach a nearly-optimal constrained policy in the last-iterate fashion.

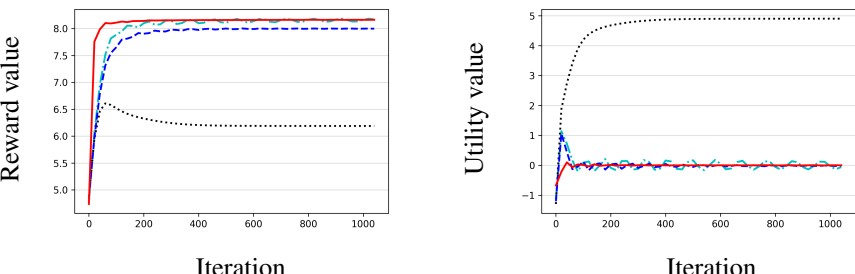

Figure 1: Convergence performance of RPG-PD, OPG-PD, and primal-dual methods. Learning curves of our RPG-PD (**– –**) and OPG-PD (**—**), and NPG-PD [23] (**–·–**) and PID Lagrangian [29] (····) methods. The horizontal axes mean the policy iterations $\{\pi_t\}_{t \geq 0}$ that are generated by each method and the vertical axes mean the value functions of the policy iterates $\{\pi_t\}_{t \geq 0}$: reward value $V_r^{\pi_t}(\rho)$ (Left) and utility value $V_g^{\pi_t}(\rho)$ (Right). In this experiment, we use the same stepsize $\eta = 0.1$ for all methods, the regularization parameter $\tau = 0.08$ for RPG-PD, and the uniform initial distribution $\rho$.

We showcase the linear convergence of OPG-PD (9) with three constant stepsizes in Figure 2. Three policy optimality gaps decrease linearly in the logarithmic scale plot, which verifies the linear last-iterate convergence of OPG-PD's policy iterates in Theorem 6. Noticeably, there is no oscillation behavior in OPG-PD's policy iterates, which perhaps is best for learning constraints [29, 10]. We also see that a large stepsize $\eta = 0.2$ improves the convergence, which is reflected by our rate.

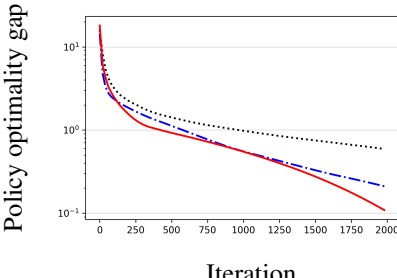

Figure 2: Convergence performance of OPG-PD with stepsize $\eta$: ($\eta = 0.05$, ····), ($\eta = 0.1$, **–·–**), ($\eta = 0.2$, **—**). The horizontal axis represents the policy iterations $\{\pi_t\}_{t \geq 0}$ that are generated by OPG-PD and the vertical axis means the policy optimality gap that measures the distance of the policy iterates $\{\pi_t\}_{t \geq 0}$ to an optimal policy $\pi^\star$: $\sum_s \|\pi_t(\cdot \mid s) - \pi^\star(\cdot \mid s)\|^2$. In this experiment, we take the initial distribution $\rho$ to be a uniform one.

Please see Appendix E for more details of this experiment, more baselines, and sensitivity analysis.

## 6 Concluding Remarks

We have presented two single-time-scale policy-based primal-dual methods for finding an optimal policy of a constrained MDP, with global non-asymptotic and last-iterate policy convergence guarantees. Our first regularized method enjoys a nearly dimension-free sublinear rate, while our second optimistic method possesses a linear rate that is problem-dependent. Our work stimulates a number of compelling future directions: (i) Our problem setting circumvents the exploration difficulty, which leaves online exploration open; (ii) Our convergence rates are not as sharp as solving convex-concave minimax optimization problems, regarding the order or instance-related constant; (iii) Last-iterate convergence is under-examined in constrained MDPs with other constraints, and unexplored for other gradient methods.

## Acknowledgments and Disclosure of Funding

D. Ding and A. Ribeiro were supported by THEORINET Simons-NSF MoDL, and DCIST CRA. K. Zhang acknowledges the support from Simons-Berkeley Research Fellowship and Northrop Grumman – Maryland Seed Grant Program.

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

# Supplementary Materials for
## "Last-Iterate Convergent Policy Gradient Primal-Dual Methods for Constrained MDPs"

## A  More Comparisons and Additional Related Works

In this section, we discuss more comparison details and other related works.

| Iterate Type | Method | Single-time-scale | Complexity |
|---|---|---|---|
| Occupancy measure | Saddle flow dynamics [36] | Yes | asymptotic |
| | ReLOAD [40] | Yes | asymptotic |
| | OPD (18) | Yes | $\Omega\left(\log^2\frac{1}{\epsilon}\right)$ |
| Policy | Dual descent [34] | No | $\Omega\left(\log^2\frac{1}{\epsilon}\right)$ |
| | Cutting-plane [35] | No | $\Omega\left(\log^3\frac{1}{\epsilon}\right)$ |
| | Policy-based ReLOAD [40] | Yes | — |
| | RPG-PD (6) | Yes | $\Omega\left(\frac{1}{\epsilon^6}\log^2\frac{1}{\epsilon}\right)$ |
| | OPG-PD (9) | Yes | $\Omega\left(\log^2\frac{1}{\epsilon}\right)$ |

Table 1: Iteration complexities of our methods and representative algorithms for solving a constrained MDP problem: $\text{maximize}_\pi V_r^\pi(\rho)$ subject to $V_g^\pi(\rho) \geq 0$ with reward/utility functions $r \in [0,1]$, $g \in [-1,1]$. The iteration complexity is the number of gradient-based updates for an algorithm to output the last policy-iterate $\pi_t$ that satisfies $V_r^\star(\rho) - V_r^{\pi_t}(\rho) \leq \epsilon$ and $-V_g^{\pi_t}(\rho) \leq \epsilon$.

**Last-iterate and value-average (or policy-mixture) performance in constrained MDPs**. There has been a flurry of research activities in studying convergence behaviors of direct policy search or policy gradient-based algorithms for constrained MDPs in the infinite-horizon discounted setting. There are two main streams: (i) Lagrangian-based policy search and (ii) Approximate constrained policy search.

(i) **Lagrangian-based policy search**. In the Lagrangian-based framework, last-iterate performance has been established as asymptotic convergence for several policy-based primal-dual algorithms, e.g., naive policy gradient-based primal-dual method [21] and actor-critic variants of policy gradient primal-dual methods [6, 22, 16, 9]. These studies rely on modeling primal-dual updates in two separate time scales and/or two nested loops as continuous-time gradient flow dynamics, and their asymptotic convergence is often restricted to some stationary points. We notice that recent global asymptotic convergence results [36, 40] are in terms of occupancy measure iterates instead of instantaneous policy iterates, as highlighted in Table 1. In contrast, our first method: OPD strengthens the asymptotic last-iterate convergence to non-asymptotic and last-iterate convergence with a linear rate.

To provide efficiency and optimality performances, it is crucial to develop algorithms with global non-asymptotic convergence guarantees. Several recent policy-based primal-dual algorithms have been proved to converge with non-asymptotic convergence rates, e.g., policy gradient primal-dual method [69], natural policy gradient-based or policy mirror-descent style primal-dual methods [23, 27, 54], accelerated natural policy gradient-based primal-dual methods [24, 25, 28], actor-critic version of natural policy gradient-based primal-dual

method [26], and anchor-changing natural policy gradient-based primal-dual method [70]. These studies characterize non-asymptotic convergence to an optimal constrained policy regarding the average of value functions, except for [27] in which the convergence is for a mixture of past policies and [28] in which the convergence is for a policy induced by a history-weighted occupancy measure. Similar non-asymptotic convergence can also be found in an occupancy measure-based primal-dual method [42], where the convergence is in terms of the average of occupancy measures. Besides, sublinear non-asymptotic convergence can be found in generative model-based methods [71, 41], regarding a mixture of past policies. In contrast, our two policy-based methods: RPG-PD and OPG-PD strengthen the sublinear non-asymptotic convergence of average value functions or a mixture of past policies to sublinear and linear non-asymptotic convergence of policy iterates. In particular, we exploit the regularization technique [52] and the optimistic gradient method [72, 37] to augment typical policy-based primal-dual methods with novel identification of decreasing distances of policy iterates to an optimal constrained policy, which allows for stronger convergence.

Instead of working with policy primal and dual variables both, Lagrangian-based dual method formulates a constrained MDP as a convex dual problem that enables classical dual ascent method [12]. Despite guaranteed convergence in dual space from convex optimization, it is challenging to compute an optimal constrained policy in primal policy space even if we use an optimal dual variable [9, 47, 30]. Recently, regularization [52] has been used in dual ascent methods [34, 35] in which policy last-iterates of natural policy gradient-based subroutines can be nearly-optimal. These dual-based algorithms [34, 35] work with two nested loops and their non-asymptotic last-iterate convergence relies on tuning loop parameters optimally. In contrast, our two policy-based methods: RPG-PD and OPG-PD remove the double loop requirement as listed in Table 1, which permits outputting the last-iterate policy as a nearly-optimal constrained policy. In Table 1, we see that our RPG-PD method has a worse rate while OPG-PD achieves a linear rate which is similar as the dual-based methods [34, 35]. Importantly, our non-asymptotic convergence characterizes the stability of primal-dual iterates generated by the algorithms, which is theoretically stronger and more appealing in practice.

(ii) **Approximate constrained policy search**. Approximation of constrained MDPs with surrogate functions has been shown to be effective, e.g., constrained policy optimization [8], successive convex relaxation [73], projection-based constrained policy optimization [74], first-order constrained optimization [75], and conservative policy update [76]. These studies have shown impressive empirical performance by iteratively solving an approximated constrained optimization problem and such performance is characterized in the worst-case policy improvement, except for [73] in which local asymptotic convergence is established, which leaves non-asymptotic convergence of policy iterates open. A related approach is the primal method [77] that treats a constrained MDP as an unconstrained one and corrects policy iterates whenever constraint violation happens. Non-asymptotic convergence of primal method has been established in terms of mixture of past policies. Another approximation method is the interior-point policy optimization [78] that solves an unconstrained MDP by adding a logarithm barrier function into the objective function, while convergence of this method is unknown. In contrast, we have supported our methods with non-asymptotic last-iterate convergence. Since Lagrangian-based methods are typically used to solve an approximated constrained optimization problem [8, 74, 75, 76], our methodologies can be applied to these methods for better convergence, which we leave as future work.

For constrained MDPs in the finite-horizon episodic setting and the total or average-reward settings, there is a rich line of works that have developed learning algorithms with non-asymptotic convergence guarantees in terms of the average of value functions [79, 80, 81, 82, 83, 84, 85, 55, 86, 87, 88, 89, 56, 90, 91, 92, 93, 94, 95], except for [96] in which global asymptotic convergence of a policy gradient method has been established in the finite-horizon constrained MDP setting. Although being not directly comparable, their non-asymptotic and last-iterate convergence are not established yet, to the best of our knowledge.

**Lagrangian-based policy gradient methods in constrained MDPs**. Our methods are closely pertinent to Lagrangian-based policy gradient methods for solving constrained MDPs in the infinite-horizon discounted setting, e.g., policy gradient-based primal-dual methods [21, 6, 22, 16, 9, 69],

natural policy gradient-based or mirror-descent style primal-dual methods [23, 27, 28, 24, 54, 25, 26], dual descent methods [97, 34, 35, 12]. Regarding algorithm implementation, primal-dual methods [21, 6, 22, 16, 9, 23, 27, 28, 24, 54, 25, 26, 69] work with primal-dual iterates simultaneously in a single loop, which is similar to the classical gradient-based primal-dual methods in constrained optimization [31, 32, 33], while diminishing stepsizes in different speeds is often required in many of them [21, 6, 16, 9]; dual descent methods [97, 34, 35, 12] intermittently operate the dual iterate only after a sufficient number of primal iterations, which often adds difficulty of tuning hyper-parameters of nested loops in practice. It is worth mentioning that, it is convenient to view such dual descent methods as primal-dual methods that update primal variable faster than iterating dual variable. With respect to convergence analysis, stochastic approximation has been widely used to establish asymptotic convergence of several primal-dual methods [6, 22, 16, 9] by analyzing the stability of limiting gradient flow dynamics, while recent methods [28, 24, 54, 25, 26, 69, 34, 35] exploit the connection between policy gradient and mirror-descent in convex optimization to prove non-asymptotic convergence. However, non-asymptotic convergence of primal-dual methods only characterizes the average of value functions [23, 54, 25, 26, 69] or a mixture of past policies [27, 28, 24] because of the dual update that results from regulating constraint violation. For the dual descent methods [34, 35], non-asymptotic convergence is characterized in terms of last-iterate policies that are computed by approximately solving unconstrained RL problems with fixed dual variables. Therefore, designing Lagrangian-based policy gradient methods that enjoy the single-loop simplicity and the non-asymptotic convergence of policy iterates is challenging, because of the oscillation and overshoot issues of updating primal-dual variables simultaneously [29, 40]. In this work, we have established non-asymptotic and last-iterate convergence of two single-loop Lagrangian-based policy gradient methods via the regularization and optimistic gradient techniques and our analysis builds on the mirror-descent analysis for policy gradient methods [49, 98, 65, 99] while focusing on the distance of policy iterates, which is stronger than the prior art. Compared with recent efforts [40, 36] as shown in Table 1, our algorithms are simpler and our theoretical guarantees are stronger.

**Gradient-based methods with last-iterate convergence for learning in games**. Since the Lagrangian-based approach for constrained MDPs can be viewed as solving a constrained saddle-point problem, another line of related work is gradient-based methods for solving saddle-point (or minimax optimization) problems with last-iterate convergence. Last-iterate convergence of gradient-based methods has been established in several scenarios, e.g., linear rates of extragradient methods for strongly convex problems [100, 101], asymptotic convergence of optimistic multiplicative weights updates for convex problems [67, 68], linear rates of optimistic gradient methods for convex problems [43], and lower bound-matching rates of extragradient and optimistic gradient methods for convex problems [39]. These studies focus on convex-concave saddle-point problems except for [102, 103, 104] in which non-asymptotic last-iterate convergence is achievable for saddle-point problems with special non-convexity structure. In contrast, our constrained saddle-point problem that result from constrained MDP is non-convex in policy primal variable, which prevents direct application of these last-iterate results. A slightly twisted exception is that our constrained saddle-point problem can be reformulated to be convex in occupancy measure instead of policy, which leads the second and third methods in Table 1. To solve our Lagrangian-based constrained saddle-point problem in policy space, our first policy-based method: RPG-PD relaxes the non-convexity by adding regularization into the objective function and we provide sublinear last-iterate policy convergence guarantee using the distance analysis for policy primal-dual iterates. To improve the convergence rate, we further develop another policy-based method: OPG-PD that extends the optimistic gradient method [37] for a class of non-convex constrained saddle-point problems. This extension departs from previous extensions for zero-sum Markov games [43, 44], because lacking of uniformly optimal policies prevents learning an optimal policy from per-state bilinear games. Instead, we provide a new distance analysis of policy primal-dual iterates of an optimistic gradient method for solving a new class of constrained non-convex saddle-point problems, with linear last-iterate policy convergence.

**Non-asymptotic last-iterate (or non-ergodic) convergence in constrained optimization**. Reduction of constrained optimization to saddle-point problems is a classical idea to solve constrained optimization problems by developing primal-dual algorithms, e.g., primal-dual interior-point methods [105], Uzawa and Arrow–Hurwicz algorithms [31, 106, 107], and Lagrange multiplier methods [108, 45]. Inspired by the Lagrange multiplier methods, many recent studies on constrained optimization have significantly advanced primal-dual algorithms with last-iterate convergence, e.g., accelerated augmented Lagrangian method [109], accelerated universal primal-

dual gradient method [110], Douglas-Rachford alternating direction method [111], inexact augmented Lagrangian method [112, 113], alternating proximal augmented Lagrangian algorithm [114], augmented Lagrangian-based decomposition method [115], faster Lagrangian method [116], and prediction-correction-based primal-dual method [117]. However, all these studies build on augmented Lagrangian methods to solve the classical convex optimization problems with linear constraints. In comparison, we have studied a class of *non-convex* constrained optimization problems that result from constrained MDPs using the standard Lagrange multiplier method. We notice that a Lagrangian-based two-player game has been used to study general non-convex constrained optimization problems with average performance analysis [118]. Our two policy-based primal-dual methods with sublinear and linear last-iterate convergence appear to be the first global non-asymptotic and last-iterate convergence result in non-convex constrained optimization.

# B Proofs in Section 2

In this section, we make some helpful observations and provide proofs of the claims in Section 2.

## B.1 Lack of uniformly optimal stationary policies in constrained MDPs

In unconstrained MDPs, there always exists an optimal policy that is optimal simultaneously for all states (e.g., see [119] and [120, Chapter 6]). In contrast, this is not true anymore for constrained MDPs. To see this, we adopt a counter-example from [46] and investigate it in the constrained MDP setting.

Figure 3: An example of a constrained MDP that has the objective function $V_r^\pi(\rho)$ and the constraint set $\{\pi \in \Pi \,|\, V_g^\pi(\rho) \geq 0\}$. The pair $(a, r, g)$ associated with a directed arrow represents (reward, utility) received when an action $a$ at a certain state is taken.

We introduce a constrained MDP with two states: Left ($L$) and Right ($R$), in Figure 3. In each state, there are two actions $\{L, R\}$. The MDP transition dynamics is deterministic. In state $L$, if the agent chooses action $L$, then the next state is $L$ and the reward/utility $(0, 0)$ is received; otherwise, action $R$ leads to next state $R$ and reward/utility $(1, -1)$. In state $R$, no matter which action the agent takes, the next state is $R$ and the reward/utility $(1, 1)$ is received.

Since the state $R$ is trivial, a stationary policy $\pi$ can be represented by the probability of taking action $L$ in state $L$ denoted by $p$. With a slight abuse of notation, we use notion $\rho$ to represent the probability of starting off from state $L$. Thus, we can compute the value functions as follows via the Bellman equations, i.e., $V(s) = \sum_a \pi(a \,|\, s)(r(s, a) + \gamma \sum_{s'} P(s' \,|\, s, a)V(s))$ for all $s$.

$$V_r(L) = \underbrace{p \times (0 + \gamma(V_r(L) \times 1 + V_r(R) \times 0))}_{\text{take action } L} + \underbrace{(1 - p)(1 + \gamma(V_r(L) \times 0 + V_r(R) \times 1))}_{\text{take action } R}$$

$$V_r(R) = \frac{1}{1 - \gamma}$$

and

$$V_g(L) = \underbrace{p \times (0 + \gamma(V_g(L) \times 1 + V_g(R) \times 0))}_{\text{take action } L} + \underbrace{(1 - p)(-1 + \gamma(V_g(L) \times 0 + V_g(R) \times 1))}_{\text{take action } R}$$

$$V_g(R) = \frac{1}{1 - \gamma}$$

or, equivalently,

$$V_r(L) = \frac{1 - p}{1 - \gamma p} \times \frac{1}{1 - \gamma}, \quad V_r(R) = \frac{1}{1 - \gamma}$$

$$\text{and} \quad V_g(L) = \frac{1-p}{1-\gamma p} \times \frac{2\gamma - 1}{1-\gamma}, \quad V_g(R) = \frac{1}{1-\gamma}.$$

(i) It is easy to check a basic case: $\gamma = 0$ [46]. We can compute the value functions as follows,

$$V_r^p(\rho) = (1-p)\rho + (1-\rho)$$

$$V_g^p(\rho) = -(1-p)\rho + (1-\rho).$$

Feasibility of the policy $p$ requires that $V_g^p(\rho) \geq 0$, i.e.,

$$p \geq \frac{2\rho - 1}{\rho} \quad \text{for any } \rho \in \left[\frac{1}{2}, 1\right]. \tag{10}$$

Hence, the maximum $V_r^p(\rho)$ within the feasible region can be reached at the optimal policy,

$$p^\star = \frac{2\rho - 1}{\rho}.$$

Therefore, the optimal policy $p^\star$ depends on the initial state distribution $\rho$. Moreover, except that $\rho = 1$ or $\frac{1}{2}$, the optimal policy $p^\star$ is a stochastic policy and is unique.

(ii) A slightly more general case is given by $\gamma = \frac{1}{4}$. Thus, we can compute the value functions as follows,

$$V_r^p(\rho) = \frac{4}{3} \times \frac{1-p}{1-p/4} \times \rho + \frac{4}{3} \times (1-\rho)$$

$$V_g^p(\rho) = -\frac{2}{3} \times \frac{1-p}{1-p/4} \times \rho + \frac{4}{3} \times (1-\rho).$$

Feasibility of the policy $p$ requires that $V_g^p(\rho) \geq 0$, i.e.,

$$p \geq 2 \times \frac{3\rho - 2}{3\rho - 1} \quad \text{for any } \rho \in \left(\frac{1}{3}, 1\right].$$

In particular, if we take $\rho = \frac{7}{9}$, then $p \geq \frac{1}{2}$. In this case, the maximization of $V_r^p(\rho)$ yields an optimal policy $p^\star = \frac{1}{2}$, which is a uniform policy and is unique.

## B.2   Scalarization fallacy in constrained MDPs

In constrained RL, scalarization is often used to reduce a constrained MDP problem to a standard unconstrained one, which might permit many unconstrained RL algorithms [8, 9]. Unfortunately, as pointed out in the literature (e.g., [46, Part 4] and [30]), such a reduction does not necessarily provide a solution to the original constrained MDP problem. It is easy to see this from the previous examples in Figure 3. In the basic case: $\gamma = 0$, if we take $\rho = \frac{3}{4}$, then from (10) the optimal policy is given by $p^\star = \frac{2}{3}$, which is a stochastic policy; we see a uniform optimal policy when $\gamma = \frac{1}{4}$. By shaping a composite function $r + \lambda g$ with some fixed $\lambda \in [0, \infty]$, the scalarization method aims to solve the following unconstrained MDP problem,

$$\operatorname*{maximize}_{\pi \in \Pi} \quad V_{r+\lambda g}^\pi(\rho). \tag{11}$$

However, by the optimality of dynamic programming [120, Chapter 6], an optimal policy is given in a deterministic form which has been widely used in theory and practice. Therefore, solving the above scalarized version of a constrained MDP problem does not necessarily provide an optimal solution for the original constrained MDP problem. We also notice that this phenomenon is reported in recent empirical studies [9, 30] and a more formal statement [47, Lemma 1]. Hence, it can be infeasible for dual descent methods [12, 30] to find an optimal constrained policy, because solving Problem (11) often yields a deterministic policy, which can be sub-optimal for a constrained MDP with a unique stochastic optimal policy, e.g., constrained MDP examples in Appendix B.1.

## B.3 Properties of saddle points

First, we state the invariance property of saddle points [121] for our constrained saddle-point problem (3). By the invariance property of saddle points, we can restrict the problem domain without changing the saddle-point property when the original saddle points are contained in the restricted domain. Let the set of max-min points be $\Pi^\star := \operatorname{argmax}_{\pi \in \Pi} \min_{\lambda \in [0,\infty]} V^\pi_{r+\lambda g}(\rho)$ and the set of min-max points be $\Lambda^\star := \operatorname{argmin}_{\lambda \in [0,\infty]} \max_{\pi \in \Pi} V^\pi_{r+\lambda g}(\rho)$.

**Lemma 8** (Invariance of saddle points)**.** *Let $(\pi^\star, \lambda^\star) \in \Pi^\star \times \Lambda^\star$ be a saddle point of $V^\pi_{r+\lambda g}(\rho)$ over $\Pi \times [0, +\infty]$. For any subset $\Lambda' \subset [0, +\infty]$, if $(\pi^\star, \lambda^\star) \in \Pi \times \Lambda'$, then $(\pi^\star, \lambda^\star)$ is a saddle point of $V^\pi_{r+\lambda g}(\rho)$ over $\Pi \times \Lambda'$.*

*Proof.* From the saddle-point property of $(\pi^\star, \lambda^\star)$, we have

$$\pi^\star \in \operatorname*{argmax}_{\pi \in \Pi} V^\pi_{r+\lambda^\star g}(\rho) \text{ and } \lambda^\star \in \operatorname*{argmin}_{\lambda \in [0,+\infty]} V^{\pi^\star}_{r+\lambda g}(\rho)$$

It is straightforward to see that

$$V^\pi_{r+\lambda^\star g}(\rho) \leq V^{\pi^\star}_{r+\lambda^\star g}(\rho) \text{ for any } \pi \in \Pi. \tag{12}$$

Since $\Lambda' \subset [0, +\infty]$ and $\lambda^\star \in \Lambda'$, $V^{\pi^\star}_{r+\lambda^\star g}(\rho) = \min_{\lambda \in \Lambda'} V^{\pi^\star}_{r+\lambda g}(\rho)$. Hence,

$$V^{\pi^\star}_{r+\lambda^\star g}(\rho) \leq V^{\pi^\star}_{r+\lambda g}(\rho) \text{ for any } \lambda \in \Lambda'. \tag{13}$$

Finally, combining (12) and (13) defines $(\pi^\star, \lambda^\star)$ as a saddle point of of $V^\pi_{r+\lambda g}(\rho)$ over $\Pi \times \Lambda'$. □

We next show the interchangeability of saddle points in two-player zero-sum games [122] for our *non-convex* game.

**Lemma 9** (Interchangeability of saddle points)**.** *Let $(\pi^\star, \lambda^\star)$, $(\bar{\pi}^\star, \bar{\lambda}^\star) \in \Pi^\star \times \Lambda^\star$ be two saddle points of $V^\pi_{r+\lambda g}(\rho)$ over $\Pi \times [0, +\infty]$. Then, both $(\pi^\star, \bar{\lambda}^\star)$ and $(\bar{\pi}^\star, \lambda^\star)$ are saddle points of $V^\pi_{r+\lambda g}(\rho)$ over $\Pi \times [0, +\infty]$.*

*Proof.* By the definition of saddle points $(\pi^\star, \lambda^\star)$ and $(\bar{\pi}^\star, \bar{\lambda}^\star)$,

$$V^\pi_{r+\lambda^\star g}(\rho) \leq V^{\pi^\star}_{r+\lambda^\star g}(\rho) \leq V^{\pi^\star}_{r+\lambda g}(\rho) \text{ for all } \pi \in \Pi \text{ and } \lambda \in [0, \infty]$$

$$V^\pi_{r+\bar{\lambda}^\star g}(\rho) \leq V^{\bar{\pi}^\star}_{r+\bar{\lambda}^\star g}(\rho) \leq V^{\bar{\pi}^\star}_{r+\lambda g}(\rho) \text{ for all } \pi \in \Pi \text{ and } \lambda \in [0, \infty].$$

Then,

$$V^{\bar{\pi}^\star}_{r+\bar{\lambda}^\star g}(\rho) \leq V^{\bar{\pi}^\star}_{r+\lambda^\star g}(\rho) \leq V^{\pi^\star}_{r+\lambda^\star g}(\rho)$$

$$V^{\pi^\star}_{r+\lambda^\star g}(\rho) \leq V^{\pi^\star}_{r+\bar{\lambda}^\star g}(\rho) \leq V^{\bar{\pi}^\star}_{r+\bar{\lambda}^\star g}(\rho).$$

Therefore,

$$V^\pi_{r+\bar{\lambda}^\star g}(\rho) \leq V^{\bar{\pi}^\star}_{r+\bar{\lambda}^\star g}(\rho) \leq V^{\pi^\star}_{r+\lambda^\star g}(\rho) \leq V^{\pi^\star}_{r+\bar{\lambda}^\star g}(\rho) \leq V^{\bar{\pi}^\star}_{r+\bar{\lambda}^\star g}(\rho) \leq V^{\pi^\star}_{r+\lambda^\star g}(\rho) \leq V^{\pi^\star}_{r+\lambda g}(\rho)$$

for all $\pi \in \Pi$ and $\lambda \in [0, \infty]$, which shows that $(\pi^\star, \bar{\lambda}^\star)$ is a saddle point of $V^\pi_{r+\lambda g}(\rho)$ over $\Pi \times [0, +\infty]$.

Similarly, we can prove it for $(\bar{\pi}^\star, \lambda^\star)$. □

## B.4 Constrained MDPs in occupancy-measure space

In the analytic approach [123, 5], the value functions in Problem (1) are bilinear in the occupancy measure $V^\pi_\diamond(\rho) = \langle \diamond, q \rangle$ for $\diamond = r$ or $u$, where $q^\pi : S \times A \to \mathbb{R}$ is an (un-normalized) occupancy measure over the state-action space,

$$q^\pi(s, a) = \sum_{t=0}^\infty \gamma^t \mathrm{Pr}(s_t = s, a_t = a \mid \pi, s_0 \sim \rho) \tag{14}$$

which adds up discounted probabilities of visiting $(s, a)$ in the execution of $\pi$. Furthermore, we let the operator $P^\top \colon \mathbb{R}^{|S||A|} \to \mathbb{R}^{|S|}$ be $(P^\top q)(s) := \sum_{s',a'} P(s \,|\, s', a') q(s', a')$, and the operator $E^\top \colon \mathbb{R}^{|S||A|} \to \mathbb{R}^{|S|}$ be $(E^\top q)(s) := \sum_a q(s, a)$ or simply $q(s)$. With a slight abuse of notation, we denote $\rho = [\,\rho(s_1), \ldots, \rho(s_{|S|})\,]^\top$. A valid occupancy measure $q^\pi \in \mathbb{R}^{|S||A|}$ satisfies the the Bellman flow equations,

$$\mathcal{Q} := \big\{\, q \in \mathbb{R}^{|S||A|} \,|\, E^\top q = \gamma P^\top q + \rho \text{ and } q \geq 0 \,\big\}. \tag{15}$$

It is known that the Bellman flow constraint is necessary and sufficient for any $q \in \mathbb{R}^{|S||A|}$ to be a valid occupancy measure (e.g., [124, Lemma 1]).

Let the concatenation of $r(s, a)$, $u(s, a)$ for all $(s, a)$ be $r$, $u \in \mathbb{R}^{|S||A|}$, respectively. In the occupancy measure space, the goal of a constrained MDP is to find a solution $q^\star$ of a linear program,

$$\underset{q \,\in\, \mathcal{Q}}{\text{maximize}} \ \langle r, q \rangle \quad \text{subject to} \ \langle u, q \rangle \geq b. \tag{16}$$

Denoting $g \colon S \times A \to [-1, 1]$ as $g := u - (1 - \gamma)b$, we simplify the constraint $\langle u, q \rangle \geq b$ as $\langle g, q \rangle \geq 0$. By the method of Lagrange multipliers, we dualize two constraints in (16) and introduce a standard Lagrangian,

$$L(q, \lambda, \mu) \ := \ \langle r + \lambda g, q \rangle \ + \ \mu^\top \big(\rho - (E - \gamma P)^\top q\big)$$

where $\pi \in \Pi$ is the primal variable, $\lambda \in [0, \infty]$ is the dual variable for the constraint $\langle g, q \rangle \geq 0$, and $\mu$ is the dual variable for the equality constraint in $\mathcal{Q}$. Since the strong duality holds for any feasible linear program, the boundedness of $\lambda^\star$ in Lemma 1 holds for $L(q, \lambda, \mu)$. Thus, any saddle point $(q^\star, \lambda^\star, \mu^\star)$ is also a max-min and min-max point, i.e., $q^\star$ is the occupancy measure associated with the optimal policy $\pi^\star$, and $(\lambda^\star, \mu^\star) \in \text{argmin}_{\lambda \geq 0, \mu} \max_{q \geq 0} L(q, \lambda, \mu)$. Boundedness of $q^\star$ is straightforward from (14),

$$q^\star \ \in \ Q \ := \ \left\{ q \in \mathbb{R}^{|S||A|} \,\big|\, 0 \leq q(s, a) \leq \frac{1}{1 - \gamma}, \forall (s, a) \in S \times A \right\}$$

which allows us further restrict $q \in Q \subset \mathcal{Q}$. We next show boundedness of $(\lambda^\star, \mu^\star)$ in Lemma 10.

**Lemma 10** (Boundedness). *Let Assumption 1 hold. Then, $\lambda^\star \in \Lambda$ and $\mu^\star \in M$, where $\Lambda$ is stated below (3) and $M := \{\mu \,|\, |\mu(s)| \leq \mu_{\max}, \forall s \in S\}$ where $\mu_{\max} := \frac{1 - \gamma + 1/\xi}{(1 - \gamma)^2}$.*

*Proof.* From the saddle-point property of $(q^\star, \lambda^\star, \mu^\star)$, we have

$$q^\star \ \in \ \underset{q \,\in\, Q}{\text{argmax}} \ L(q, \lambda^\star, \mu^\star)$$

$$(\lambda^\star, \mu^\star) \ \in \ \underset{\lambda \geq 0, \mu}{\text{argmin}} \ L(q^\star, \lambda, \mu)$$

equivalently, for any $q \in Q$, $\partial_q L(q^\star, \lambda^\star, \mu^\star)^\top (q - q^\star) \leq 0$, and for any $\mu$ and $\lambda \geq 0$, $\partial_\mu L(q^\star, \lambda^\star, \mu^\star)^\top (\mu - \mu^\star) + \partial_\lambda L(q^\star, \lambda^\star, \mu^\star)(\lambda - \lambda^\star) \geq 0$. Hence, for any $q \in Q$,

$$\langle r + (\gamma P - E)\mu^\star + \lambda^\star g, q - q^\star \rangle \ \leq \ 0.$$

Arbitrary $q \in Q$ demands the inequality $r + (\gamma P - E)\mu^\star + \lambda^\star g \leq 0$. By the definition of occupancy measure, we know that for any $s \in S$ there exists an action $a \in A$ such that $q^\star(s, a) > 0$. Thus, the equality $r + (\gamma P - E)\mu^\star + \lambda^\star g = 0$ must hold at such state-action pairs in which we represent the associated reward and transition by $(\bar{r}, \bar{P})$. Hence,

$$\big\|\bar{r} + \lambda^\star g\big\|_\infty \ = \ \big\|(\gamma \bar{P} - \bar{E})\mu^\star\big\|_\infty$$

$$\geq \ (1 - \gamma) \big\|\mu^\star\big\|_\infty.$$

Thus, $(1 - \gamma) \big\|\mu^\star\big\|_\infty \leq 1 + \lambda^\star$. However, by Assumption 1, $L(\bar{q}, \lambda^\star, \mu^\star) \leq L(q^\star, \lambda^\star, \mu^\star)$. Hence,

$$\langle r, q^\star - \bar{q} \rangle \ + \ \lambda^\star \langle g, q^\star - \bar{q} \rangle \ \geq \ 0$$

which, together with the feasibility of $\bar{q}$ and the optimality of $q^\star$, imply that $0 \leq \lambda^\star \xi \leq \langle r, q^\star - \bar{q} \rangle$. Hence, $0 \leq \lambda^\star \leq \frac{1}{(1 - \gamma)\xi}$, which further yields a bound on $\big\|\mu^\star\big\|_\infty$. $\qquad\square$

We now obtain a constrained saddle-point problem in terms of the $q$-based Lagrangian,

$$\underset{q \in Q}{\text{maximize}} \; \underset{\lambda \in \Lambda, \, \mu \in M}{\text{minimize}} \; L(q, \lambda, \mu) \; = \; \underset{\lambda \in \Lambda, \, \mu \in M}{\text{minimize}} \; \underset{q \in Q}{\text{maximize}} \; L(q, \lambda, \mu) \tag{17}$$

where we take bounded polytopes $Q$ and $\Lambda \times M$ such that they contain $q^\star$ and $(\lambda^\star, \mu^\star)$, respectively.

For notational brevity, we introduce $z = (q, \lambda, \mu)$, $Z := Q \times \Lambda \times M$, and $Z^\star := Q^\star \times \Lambda^\star \times M^\star$ which contains all saddle points $z^\star := (q^\star, \lambda^\star, \mu^\star)$. Let the gradient of $L(q, \lambda, \mu)$ be

$$F(q, \lambda, \mu) \; := \; \begin{bmatrix} -\nabla_q L(q, \lambda, \mu) \\ \\ \nabla_{(\lambda, \mu)} L(q, \lambda, \mu) \end{bmatrix}.$$

Due to the bilinearity of $L(q, \lambda, \mu)$ over a compact domain, $L(q, \lambda, \mu)$ has a gradient Lipschitz constant $L_f$. Let $\mathcal{P}_X$ be the projection operator onto a set $X$, i.e., $\mathcal{P}_X(x) := \operatorname{argmin}_{x' \in X} \|x' - x\|$. Since the $q$-based Lagrangian $L(q, \lambda, \mu)$ is bilinear and the domains are polytopes, Problem (17) satisfies the metric subregularity condition [38, Theorem 5].

**Lemma 11** (Metric subregularity condition). *Let Assumption 1 hold. Then, the gradient function $F(z)$ satisfies that for any $z \in Z/Z^\star$ with $z^\star = \mathcal{P}_{Z^\star}(z)$,*

$$\sup_{z' \in Z} \frac{F(z)^\top (z - z')}{\|z - z'\|} \; \geq \; C \, \|z - z^\star\|$$

*where $C > 0$ is a problem-dependent constant.*

Hence, application of the optimistic gradient method [38] to Problem (17) yields an Optimistic Primal-Dual (OPD) algorithm that begins with two initial tuples of primal/dual variables $(q_0, \lambda_0, \mu_0) = (\hat{q}_1, \hat{\lambda}_1, \hat{\mu}_1) \in Z$, and performs two gradient steps for each primal/dual variable at time $t \geq 1$,

$$\begin{aligned} z_t &= \mathcal{P}_Z \left( \hat{z}_t - \eta F(z_{t-1}) \right) \\ \hat{z}_{t+1} &= \mathcal{P}_Z \left( \hat{z}_t - \eta F(z_t) \right) \end{aligned} \tag{18}$$

where $\eta$ is the stepsize. Let the squared distance of a point $z \in Z$ to the set $Z^\star$ be $\text{dist}^2(z, Z^\star) := \|z - \mathcal{P}_{Z^\star}(z)\|^2$. It is straightforward to employ [38, Theorem 8] to claim last-iterate convergence guarantee of OPD (18) below.

**Theorem 12** (Linear convergence of OPD). *Let Assumption 1 hold. If the stepsize $\eta$ in OPD (18) satisfies $\eta < \frac{1}{8L_f}$, then the iterates $\{z_t\}_{t \geq 0}$ converge to the set of saddle points $Z^\star$ linearly,*

$$\text{dist}^2(z_t, Z^\star) \; \leq \; C_1 \left( \frac{1}{1 + C_2} \right)^t$$

*where $C_1 = 64 \, \text{dist}^2(\hat{z}_1, Z^\star)$ and $C_2 = \min(\frac{16\eta^2 C^2}{81}, \frac{1}{2})$ for a problem-dependent constant $C > 0$.*

Theorem 12 shows that the primal-dual iterates of OPD converge to the saddle point set $Z^\star$ in linear rate. Compared with a contemporaneous work [40], OPD is free of projection to a occupancy measure set, and enjoys non-asymptotic and last-iterate linear convergence. If the underlying policy is recovered via $\pi_t(a \,|\, s) = \frac{q_t(s,a)}{\sum_{a'} q_t(s,a')}$ for all $(s, a)$, these policy iterates $\pi_t$ associated with the occupancy measure iterates $q_t$ also converge to an optimal constrained policy $\pi^\star$.

**Corollary 13** (Nearly-optimal constrained policy). *Let Assumption 1 hold. Assume $\rho_{\min} := \min_s \rho(s) > 0$ and re-define $Q := \{q \in \mathbb{R}^{|S||A|} \,|\, 0 \leq q(s,a) \leq 1/(1-\gamma), q(s) \geq \rho_{\min}/(1 - \gamma), \forall (s, a) \in S \times A\}$. If we set the stepsize $\eta$ in a similar way as in Theorem 12, then the recovered policy iterates $\{\pi_t\}_{t \geq 0}$ of OPD (18) satisfy*

$$V_r^{\pi^\star}(\rho) - V_r^{\pi_t}(\rho) \; \leq \; \epsilon \quad \text{and} \quad -V_g^{\pi_t}(\rho) \; \leq \; \epsilon \quad \text{for any } t = \Omega \left( \log^2 \frac{1}{\epsilon} \right)$$

*where $\pi_t(a \,|\, s) = \frac{q_t(s,a)}{\sum_{a'} q_t(s,a')}$ for all $(s, a)$, and $\Omega(\cdot)$ only has some problem-dependent constant.*

*Proof.* The key to our proof is to connect the occupancy measure iterates $q_t$ with associated policy iterates $\pi_t$. It is straightforward that Theorem 12 continues to hold, with a further restricted the domain $Q$. If we set the stepsize $\eta$ in a similar way as in Theorem 12, then for any $t = \Omega(\log \frac{1}{\epsilon})$,

$$\max\left\{ \left\|\mathcal{P}_{Q^\star}(q_t) - q_t\right\|^2, \ \left\|\mathcal{P}_{\Lambda^\star}(\lambda_t) - \lambda_t\right\|^2, \ \left\|\mathcal{P}_{M^\star}(\mu_t) - \mu_t\right\|^2 \right\} = O(\epsilon).$$

Let $\pi_t^\star$ be a policy that is associated with the occupancy measure $q^{\pi_t^\star} = \mathcal{P}_{Q^\star}(q_t)$ and $\pi_t$ be a policy that is associated with the occupancy measure iterate $q_t$, i.e., $\pi_t(a \,|\, s) = \frac{q_t(s,a)}{\sum_{a'} q_t(s,a')}$ for all $(s,a)$. We denote $(\lambda_t^\star, \mu_t^\star) := (\mathcal{P}_{\Lambda^\star}(\lambda_t), \mathcal{P}_{M^\star}(\mu_t))$. Thus,

$$
\begin{aligned}
&\sum_s d_\rho^{\pi_t^\star}(s) \left\|\pi_t(\cdot \,|\, s) - \pi_t^\star(\cdot \,|\, s)\right\| \\
&= \sum_s d_\rho^{\pi_t^\star}(s) \left\|\frac{q_t(s,\cdot)}{q_t(s)} - \frac{q^{\pi_t^\star}(s,\cdot)}{q^{\pi_t^\star}(s)}\right\| \\
&\leq \sum_s d_\rho^{\pi_t^\star}(s) \frac{\left\|q_t(s,\cdot) - q^{\pi_t^\star}(s,\cdot)\right\| q_t(s)}{q_t(s) q^{\pi_t^\star}(s)} + \sum_s d_\rho^{\pi_t^\star}(s) \frac{\left\|q_t(s,\cdot)\right\| |q_t(s) - q^{\pi_t^\star}(s)|}{q_t(s) q^{\pi_t^\star}(s)} \\
&\leq \sum_s \frac{\left\|q_t(s,\cdot) - q^{\pi_t^\star}(s,\cdot)\right\|}{q^{\pi_t^\star}(s)} + \sqrt{|A|} \sum_s \frac{|q_t(s) - q^{\pi_t^\star}(s)|}{q_t(s)} \\
&\leq \frac{\sqrt{|A|}}{\rho_{\min}} \sum_s \left(\left\|q_t(s,\cdot) - q^{\pi_t^\star}(s,\cdot)\right\| + |q_t(s) - q^{\pi_t^\star}(s)|\right) \\
&\leq \frac{\sqrt{|A|}}{\rho_{\min}} \left(\sqrt{|S|}\sqrt{\sum_s \|q_t(s,\cdot) - q^{\pi_t^\star}(s,\cdot)\|^2} + \sqrt{|S||A|}\sqrt{\sum_{s,a} |q_t(s,a) - q^{\pi_t^\star}(s,a)|^2}\right) \\
&\leq \frac{2\sqrt{|S||A|}}{\rho_{\min}} \left\|q_t - q^{\pi_t^\star}\right\| \\
&= \frac{2\sqrt{|S||A|}}{\rho_{\min}} \left\|q_t - \mathcal{P}_{Q^\star}(q_t)\right\|
\end{aligned}
$$

where the first inequality is due to triangle inequality, we use the fact: $(1-\gamma)q^{\pi_t^\star}(s) = d_\rho^{\pi_t^\star}(s)$, $d_\rho^{\pi_t^\star}(s) \leq 1$, and $\|q_t(s,\cdot)\| \leq \frac{\sqrt{|A|}}{1-\gamma}$ in the second inequality, the third inequality is due to that $q^{\pi_t^\star}(s), q_t(s) \geq \rho_{\min}$, and $\rho_{\min} > 0$, we apply Cauchy-Schwarz inequality in the fourth inequality, and finally we combine two square root terms by relaxing the first one in the last inequality.

First, we have

$$
\begin{aligned}
V_r^{\pi_t^\star}(\rho) - V_r^{\pi_t}(\rho) &= \frac{1}{1-\gamma} \sum_{s,a} d_\rho^{\pi_t^\star}(s) \left(\pi_t^\star(a \,|\, s) - \pi_t(a \,|\, s)\right) Q_r^{\pi_t}(s,a) \\
&\leq \frac{1}{(1-\gamma)^2} \sum_s d_\rho^{\pi_t^\star}(s) \left\|\pi_t^\star(\cdot \,|\, s) - \pi_t(\cdot \,|\, s)\right\|_1 \\
&\leq \frac{\sqrt{|A|}}{(1-\gamma)^2} \sum_s d_\rho^{\pi_t^\star}(s) \left\|\pi_t^\star(\cdot \,|\, s) - \pi_t(\cdot \,|\, s)\right\|
\end{aligned}
$$

where the equality is due to performance difference lemma in Lemma 28, we use Cauchy–Schwarz inequality in the first inequality, and the second inequality is due to $\|x\|_1 \leq \sqrt{d}\|x\|_2$ for $x \in \mathbb{R}^d$, which shows $V_r^{\pi_t^\star}(\rho) - V_r^{\pi_t}(\rho) \leq O(\sqrt{\epsilon})$. By the optimality of $\pi_t^\star$, $V_r^{\pi_t^\star}(\rho) = V_r^{\pi^\star}(\rho)$. Therefore, $V_r^{\pi^\star}(\rho) - V_r^{\pi_t}(\rho) \leq O(\sqrt{\epsilon})$.

Second, we have

$$-V_g^{\pi_t}(\rho) = \underbrace{-V_g^{\hat{\pi}_t^\star}(\rho)}_{(i)} + \underbrace{V_g^{\hat{\pi}_t^\star}(\rho) - V_g^{\pi_t}(\rho)}_{(ii)}.$$

Similar to bounding $V_r^{\pi_t^\star}(\rho) - V_r^{\pi_t}(\rho)$, we can show that (ii) $\leq O(\sqrt{\epsilon})$. By the optimality of $\pi_t^\star$, we have $V_g^{\pi_t^\star}(\rho) \geq 0$. Therefore, $-V_g^{\pi_t}(\rho) \leq O(\sqrt{\epsilon})$.

Finally, we replace the accuracy $\sqrt{\epsilon}$ by $\epsilon$ and combine all big O notation to conclude the proof. $\qquad\square$

Corollary 13 states that after an almost constant number of iterations a policy induced by the last occupancy measure iterate of OPD is an $\epsilon$-optimal constrained policy for Problem (1). We notice that recent last-iterate convergence result for convex-concave saddle-point problems [39] is also applicable to Problem (17), which provides the optimal rate without problem-dependent constants. It is worth mentioning that direct application of such last-iterate convergence results in convex minimax optimization to constrained MDPs with general utilities [125, 126] and convex MDPs [124, 47] in occupancy-measure space is also straightforward. We omit these exercises in this paper, and focus on the design and analysis of algorithms in policy space.

## C  Proofs in Section 3

In this section, we provide proofs of the claims in Section 3.

### C.1  Existence and uniqueness of regularized saddle points

**Lemma 14** (Existence and uniqueness). *There exists a unique primal-dual pair* $(\bar{\pi}, \bar{\lambda}) \in \Pi \times \Lambda$ *such that* $L_\tau(\bar{\pi}, \lambda) \geq L_\tau(\bar{\pi}, \bar{\lambda}) \geq L_\tau(\pi, \bar{\lambda})$ *for any* $\pi \in \Pi$ *and* $\lambda \in \Lambda$.

*Proof.* We first re-write the regularized Lagrangian $L_\tau(\pi, \lambda)$ in terms of occupancy measure $q \in \mathcal{Q}$,

$$L_\tau(\pi, \lambda) \;=\; \langle r + \lambda g, q \rangle \;+\; \tau \left( \mathcal{H}(\pi) + \frac{1}{2}\lambda^2 \right)$$

$$\mathcal{H}(\pi) \;=\; -\sum_{s,a} q(s,a) \log \frac{q(s,a)}{\sum_{a'} q(s,a')} \;:=\; \mathcal{H}(q).$$

We next use notation $L_\pi(q, \lambda)$ to represent $L_\tau(\pi, \lambda)$. We first check that $\mathcal{H}(q)$ is a concave function,

$$
\begin{aligned}
&\mathcal{H}(\alpha q_1 + (1-\alpha)q_2) \\
={} &-\sum_{s,a} (\alpha q_1(s,a) + (1-\alpha)q_2(s,a)) \log \frac{\alpha q_1(s,a) + (1-\alpha)q_2(s,a)}{\alpha \sum_{a'} q_1(s,a') + (1-\alpha)\sum_{a'} q_2(s,a')} \\
\geq{} &-\sum_{s,a} \alpha q_1(s,a) \log \frac{\alpha q_1(s,a)}{\alpha \sum_{a'} q_1(s,a')} - \sum_{s,a} (1-\alpha)q_2(s,a) \log \frac{(1-\alpha)q_2(s,a)}{(1-\alpha)\sum_{a'} q_2(s,a')} \\
={} &\alpha \mathcal{H}(q_1) + (1-\alpha)\mathcal{H}(q_2)
\end{aligned}
$$

for any $q_1,\, q_2 \in \mathcal{Q}$ and $\alpha \in [0,1]$, where the inequality is because of the log sum inequality $(\sum_i a_i) \ln \frac{\sum_i a_i}{\sum_i b_i} \leq \sum_i a_i \ln \frac{a_i}{b_i}$ for non-negative $a_i$ and $b_i$, and the equality holds if and only if

$$\frac{q_1(s,a)}{\sum_{a'} q_1(s,a')} \;=\; \frac{q_2(s,a)}{\sum_{a'} q_2(s,a')} \quad \text{for all } s, a.$$

Therefore, $L_\tau(q, \lambda)$ is concave in $q \in \mathcal{Q}$ and strongly convex in $\lambda \in \Lambda$. We notice that $\mathcal{Q}$ is a polytope and $\Lambda$ is a bounded interval. By Sion's minimax theorem [127], $L_\tau(q, \lambda)$ has a saddle point $(\bar{q}, \bar{\lambda}) \in \mathcal{Q} \times \Lambda$. From the one-to-one correspondence between policy and occupancy measure, $\bar{q}$ induces a policy $\bar{\pi}$ and $(\bar{\pi}, \bar{\lambda})$ serves as a saddle point of $L_\tau(\pi, \lambda)$, which proves the existence of saddle points.

To show the uniqueness of $(\bar{\pi}, \bar{\lambda})$ (or $(\bar{q}, \bar{\lambda})$), it is sufficient to show that $L_\tau(q, \lambda)$ is strictly concave in $q$ and strictly convex in $\lambda$. The second argument is straightforward from the strong convexity of $L_\tau(q, \lambda)$ in $\lambda$. We next show the first argument that $L_\tau(q, \lambda)$ is strictly concave in $q$ for any $\lambda \in \Lambda$. Assume that there are two (different) policies $\pi_1$ and $\pi_2$ and the associated two occupancy measures are $q_1$ and $q_2$. From the concavity of $H(q)$, there is another occupancy measure $\alpha q_1 + (1-\alpha)q_2$

that satisfies

$$L_\tau(\alpha q_1 + (1-\alpha)q_2, \lambda)$$

$$= \langle r + \lambda g, \alpha q_1 + (1-\alpha)q_2\rangle + \tau\left(\mathcal{H}(\alpha q_1 + (1-\alpha)q_2) + \frac{1}{2}\lambda^2\right)$$

$$\geq \alpha\left(\langle r + \lambda g, q_1\rangle + \tau\mathcal{H}(q_1) + \frac{\tau}{2}\lambda^2\right) + (1-\alpha)\left(\langle r + \lambda g, q_2\rangle + \tau\mathcal{H}(q_2) + \frac{\tau}{2}\lambda^2\right)$$

$$= \alpha L_\tau(q_1, \lambda) + (1-\alpha)L_\tau(q_2, \lambda)$$

where the inequality $\geq$ has to be strict since $\pi_1$ and $\pi_2$ differentiate by at least one state-action pair $(\bar{s}, \bar{a})$,

$$\pi_1(\bar{a}\,|\,\bar{s}) = \frac{q_1(\bar{s}, \bar{a})}{\sum_{a'} q_1(\bar{s}, a')} \neq \frac{q_2(\bar{s}, \bar{a})}{\sum_{a'} q_2(\bar{s}, a')} = \pi_2(\bar{a}\,|\,\bar{s}).$$

Hence, a new policy associated with $\alpha q_1 + (1-\alpha)q_2$ can achieve a strictly higher value of the convex combination of two objective functions, unless $q_1 = q_2$ or $\alpha$ equals either 0 or 1. Therefore, $L_\tau(q, \lambda)$ is strictly concave in $q \in \mathcal{Q}$ for any $\lambda \in \Lambda$. $\qquad\square$

Let a saddle point of $L_\tau(\pi, \lambda)$ be $\pi_\tau^\star = \text{argmax}_{\pi \in \Pi} \min_{\lambda \in \Lambda} L_\tau(\pi, \lambda)$ and $\lambda_\tau^\star = \text{argmin}_{\lambda \in \Lambda} \max_{\pi \in \Pi} L_\tau(\pi, \lambda)$. Existence of saddle points in Lemma 14 ensures that,

$$L_\tau(\pi, \lambda_\tau^\star) \leq L_\tau(\pi_\tau^\star, \lambda_\tau^\star) \leq L_\tau(\pi_\tau^\star, \lambda) \text{ for all } (\pi, \lambda) \in \Pi \times \Lambda$$

in which the first inequality implies that for any $\pi \in \Pi$,

$$V_{r+\lambda_\tau^\star g}^{\pi_\tau^\star}(\rho) + \tau\left(\mathcal{H}(\pi_\tau^\star) + \frac{1}{2}(\lambda_\tau^\star)^2\right) \geq V_{r+\lambda_\tau^\star g}^{\pi}(\rho) + \tau\left(\mathcal{H}(\pi) + \frac{1}{2}(\lambda_\tau^\star)^2\right)$$

$$\geq V_{r+\lambda_\tau^\star g}^{\pi}(\rho) + \tau\frac{1}{2}(\lambda_\tau^\star)^2$$

and the second inequality implies that for any $\lambda \in \Lambda$,

$$V_{r+\lambda g}^{\pi_\tau^\star}(\rho) + \tau\left(\mathcal{H}(\pi_\tau^\star) + \frac{1}{2}\lambda^2\right) \geq V_{r+\lambda_\tau^\star g}^{\pi_\tau^\star}(\rho) + \tau\left(\mathcal{H}(\pi_\tau^\star) + \frac{1}{2}(\lambda_\tau^\star)^2\right)$$

$$\geq V_{r+\lambda_\tau^\star g}^{\pi_\tau^\star}(\rho) + \tau\mathcal{H}(\pi_\tau^\star)$$

Combination of two implied inequalities above leads to (5), i.e., $(\pi_\tau^\star, \lambda_\tau^\star)$ is a saddle point of the original Lagrangian $V_{r+\lambda g}^{\pi}(\rho)$, up to two $\tau$-terms.

### C.2 Proof of Theorem 2

*Proof.* We begin with the standard decomposition of the primal-dual gap,

$$L_\tau(\pi_\tau^\star, \lambda_t) - L_\tau(\pi_t, \lambda_\tau^\star) = \underbrace{L_\tau(\pi_\tau^\star, \lambda_t) - L_\tau(\pi_t, \lambda_t)}_{\text{(i)}} + \underbrace{L_\tau(\pi_t, \lambda_t) - L_\tau(\pi_t, \lambda_\tau^\star)}_{\text{(ii)}} \qquad (19)$$

and we next deal with (i) and (ii), separately. We notice that

$$\mathcal{H}(\pi) := \mathbb{E}\left[\sum_{t=0}^{\infty} -\gamma^t \log \pi(a_t\,|\,s_t)\right] = -\frac{1}{1-\gamma}\sum_{s,a} d_\rho^\pi(s)\pi(a\,|\,s)\log\pi(a\,|\,s).$$

We note that $Q_{r+\lambda_t g+\tau\psi_t}^{\pi_t}(s, a)$ is a sum of a soft $Q$ value function associated with a composite function $r + \lambda_t g - \tau\log\pi_t$, and $-\tau\log\pi_t$. Because of the boundedness of the soft $Q$ value function and the $\epsilon_0$-restriction on the policy domain $\hat{\Delta}(A)$, $|Q_{r+\lambda_t g+\tau\psi_t}^{\pi_t}(s, a)| \leq \frac{1}{1-\gamma}(1 + \frac{1}{(1-\gamma)\xi} + \tau\log|A|) -$

$\tau \log \frac{\epsilon_0}{|A|} := C_{\tau,\xi,\epsilon_0}$. For the term (i),

$$L_\tau(\pi_\tau^\star, \lambda_t) - L_\tau(\pi_t, \lambda_t)$$

$$= V_{r+\lambda_t g}^{\pi_\tau^\star}(\rho) - V_{r+\lambda_t g}^{\pi_t}(\rho)$$
$$- \frac{\tau}{1-\gamma} \sum_{s,a} d_\rho^{\pi_\tau^\star}(s)\pi_\tau^\star(a\,|\,s) \log \pi_\tau^\star(a\,|\,s) + \frac{\tau}{1-\gamma} \sum_{s,a} d_\rho^{\pi_t}(s)\pi_t(a\,|\,s) \log \pi_t(a\,|\,s)$$

$$= V_{r+\lambda_t g+\tau\psi_t}^{\pi_\tau^\star}(\rho) - V_{r+\lambda_t g+\tau\psi_t}^{\pi_t}(\rho) - \tau V_{\psi_t}^{\pi_\tau^\star}(\rho) + \tau V_{\psi_t}^{\pi_t}(\rho)$$
$$- \frac{\tau}{1-\gamma} \sum_{s,a} d_\rho^{\pi_\tau^\star}(s)\pi_\tau^\star(a\,|\,s) \log \pi_\tau^\star(a\,|\,s) + \frac{\tau}{1-\gamma} \sum_{s,a} d_\rho^{\pi_t}(s)\pi_t(a\,|\,s) \log \pi_t(a\,|\,s)$$

$$= \frac{1}{1-\gamma} \sum_{s,a} d_\rho^{\pi_\tau^\star}(s) \left(\pi_\tau^\star(a\,|\,s) - \pi_t(a\,|\,s)\right) Q_{r+\lambda_t g+\tau\psi_t}^{\pi_t}(s,a)$$
$$+ \frac{\tau}{1-\gamma} \sum_{s,a} d_\rho^{\pi_\tau^\star}(s)\pi_\tau^\star(a\,|\,s) \log \pi_t(a\,|\,s) - \frac{\tau}{1-\gamma} \sum_{s,a} d_\rho^{\pi_t}(s)\pi_t(a\,|\,s) \log \pi_t(a\,|\,s)$$
$$- \frac{\tau}{1-\gamma} \sum_{s,a} d_\rho^{\pi_\tau^\star}(s)\pi_\tau^\star(a\,|\,s) \log \pi_\tau^\star(a\,|\,s) + \frac{\tau}{1-\gamma} \sum_{s,a} d_\rho^{\pi_t}(s)\pi_t(a\,|\,s) \log \pi_t(a\,|\,s)$$

$$= \frac{1}{1-\gamma} \sum_{s,a} d_\rho^{\pi_\tau^\star}(s) \left(\pi_\tau^\star(a\,|\,s) - \pi_t(a\,|\,s)\right) Q_{r+\lambda_t g+\tau\psi_t}^{\pi_t}(s,a) - \frac{\tau}{1-\gamma} \sum_s d_\rho^{\pi_\tau^\star}(s)\mathrm{KL}_t(s)$$

$$\leq \sum_s d_\rho^{\pi_\tau^\star}(s) \left(\frac{\mathrm{KL}_t(s) - \mathrm{KL}_{t+1}(s)}{\eta(1-\gamma)}\right) + \eta(C_{\tau,\xi,\epsilon_0})^2 - \frac{\tau}{1-\gamma} \sum_s d_\rho^{\pi_\tau^\star}(s)\mathrm{KL}_t(s)$$

$$= \sum_s d_\rho^{\pi_\tau^\star}(s) \left(\frac{(1-\eta\tau)\mathrm{KL}_t(s) - \mathrm{KL}_{t+1}(s)}{\eta(1-\gamma)}\right) + \eta(C_{\tau,\xi,\epsilon_0})^2$$

$$= \frac{(1-\eta\tau)\mathrm{KL}_t(\rho) - \mathrm{KL}_{t+1}(\rho)}{\eta(1-\gamma)} + \eta(C_{\tau,\xi,\epsilon_0})^2.$$

where the first two equalities are because of the entropy regularization $\mathcal{H}(\pi)$, we apply the performance difference lemma in Lemma 28 and $\psi_t(s,a) = -\log \pi_t(a\,|\,s)$ to the third equality, the first inequality is due to an application of Lemma 27 with $x^\star = \pi_\tau^\star(\cdot\,|\,s)$, $x = \pi_t(\cdot\,|\,s)$, $g = -Q_{r+\lambda_t g+\tau\psi_t}^{\pi_t}(s,a)/(1-\gamma)$, and $\eta \leq 1/C_{\tau,\xi,\epsilon_0}$.

Similarly, for the term (ii),

$$L_\tau(\pi_t, \lambda_t) - L_\tau(\pi_t, \lambda_\tau^\star)$$

$$= V_{r+\lambda_t g}^{\pi_t}(\rho) - V_{r+\lambda_\tau^\star g}^{\pi_t}(\rho) + \frac{1}{2}\tau(\lambda_t)^2 - \frac{1}{2}\tau(\lambda_\tau^\star)^2$$

$$= (\lambda_t - \lambda_\tau^\star)V_g^{\pi_t}(\rho) + \frac{1}{2}\tau(\lambda_t)^2 - \frac{1}{2}\tau(\lambda_\tau^\star)^2$$

$$= (\lambda_t - \lambda_\tau^\star)\left(V_g^{\pi_t}(\rho) + \tau\lambda_t\right) - \frac{1}{2}\tau(\lambda_t - \lambda_\tau^\star)^2$$

$$\leq \frac{(\lambda_\tau^\star - \lambda_t)^2 - (\lambda_\tau^\star - \lambda_{t+1})^2}{2\eta} + \frac{1}{2}\eta(C_{\tau,\xi}')^2 - \frac{1}{2}\tau(\lambda_t - \lambda_\tau^\star)^2$$

$$= \frac{(1-\eta\tau)(\lambda_\tau^\star - \lambda_t)^2 - (\lambda_\tau^\star - \lambda_{t+1})^2}{2\eta} + \frac{1}{2}\eta(C_{\tau,\xi}')^2$$

where the inequality is due to the standard descent lemma [128] and $V_g^{\pi_t}(\rho) + \tau\lambda_t \leq \frac{1}{1-\gamma}(1+\frac{\tau}{\xi}) := C_{\tau,\xi}'$.

Using the definition $\Phi_t := \mathrm{KL}_t(\rho) + \frac{1}{2}(\lambda_\tau^\star - \lambda_t)^2$, we combine the two inequalities above to show that,

$$\Phi_{t+1} \leq (1-\eta\tau)\Phi_t - \eta(L_\tau(\pi_\tau^\star, \lambda_t) - L_\tau(\pi_t, \lambda_\tau^\star)) + \eta^2 \max\left((C_{\tau,\xi,\epsilon_0})^2, (C_{\tau,\xi}')^2\right)$$

$$\leq (1-\eta\tau)\Phi_t + \eta^2 \max\left((C_{\tau,\xi,\epsilon_0})^2, (C_{\tau,\xi}')^2\right).$$

where the second inequality is due to Lemma 14. If we expand the inequality above recursively, then,

$$
\begin{aligned}
\Phi_{t+1} & \leq (1-\eta\tau)\Phi_t + \eta^2 \max\left((C_{\tau,\xi,\epsilon_0})^2, (C'_{\tau,\xi})^2\right) \\
& \leq (1-\eta\tau)^2\Phi_{t-1} + (\eta^2 + \eta^2(1-\eta\tau))\max\left((C_{\tau,\xi,\epsilon_0})^2, (C'_{\tau,\xi})^2\right) \\
& \leq \cdots \\
& \leq (1-\eta\tau)^t\Phi_1 + \left(\eta^2\left(1 + (1-\eta\tau) + (1-\eta\tau)^2 + \cdots\right)\right)\max\left((C_{\tau,\xi,\epsilon_0})^2, (C'_{\tau,\xi})^2\right) \\
& \leq (1-\eta\tau)^t\Phi_1 + \frac{\eta}{\tau}\max\left((C_{\tau,\xi,\epsilon_0})^2, (C'_{\tau,\xi})^2\right) \\
& \leq e^{-\eta\tau t}\Phi_1 + \frac{\eta}{\tau}\max\left((C_{\tau,\xi,\epsilon_0})^2, (C'_{\tau,\xi})^2\right)
\end{aligned}
$$

which completes the proof. $\qquad\square$

### C.3  Proof of Corollary 3

*Proof.* According to Theorem 2, if we take $\tau = \Theta(\epsilon)$, $\eta = \Theta(\epsilon^2)$, and $\epsilon_0 = \epsilon$, then $\Phi_{t+1} = O(\epsilon)$ for any $t = \Omega(\frac{1}{\epsilon^3}\log\frac{1}{\epsilon})$, where $\Omega(\cdot)$ hides some problem-dependent constant. We next consider a primal-dual iterate $(\pi_t, \lambda_t)$ for some $t = \Omega(\frac{1}{\epsilon^3}\log\frac{1}{\epsilon})$. It is straightforward to check that

$$
\mathrm{KL}_t(\rho) = O(\epsilon) \text{ and } \frac{1}{2}(\lambda^\star_\tau - \lambda_t)^2 = O(\epsilon).
$$

First, we have

$$
V_r^{\pi^\star}(\rho) - V_r^{\pi_t}(\rho) = \underbrace{V_r^{\pi^\star}(\rho) - V_r^{\pi^\star_\tau}(\rho)}_{\text{(i)}} + \underbrace{V_r^{\pi^\star_\tau}(\rho) - V_r^{\pi_t}(\rho)}_{\text{(ii)}}. \tag{20}
$$

For the term (ii), because $\mathrm{KL}_t(\rho) = O(\epsilon)$, we have

$$
\begin{aligned}
\text{(ii)} & = \frac{1}{1-\gamma}\sum_{s,a} d_\rho^{\pi^\star_\tau}(s)\left(\pi^\star_\tau(a\,|\,s) - \pi_t(a\,|\,s)\right)Q_r^{\pi_t}(s,a) \\
& \leq \frac{1}{(1-\gamma)^2}\sum_s d_\rho^{\pi^\star_\tau}(s)\|\pi^\star_\tau(\cdot\,|\,s) - \pi_t(\cdot\,|\,s)\|_1 \\
& \leq \frac{1}{(1-\gamma)^2}\sum_s d_\rho^{\pi^\star_\tau}(s)\sqrt{2\,\mathrm{KL}_t(s)} \\
& \leq \frac{1}{(1-\gamma)^2}\sqrt{2\sum_s d_\rho^{\pi^\star_\tau}(s)\mathrm{KL}_t(s)} \\
& = \frac{1}{(1-\gamma)^2}\sqrt{2\,\mathrm{KL}_t(\rho)}
\end{aligned}
$$

where we use Cauchy–Schwarz inequality in the first and third inequalities, and the second inequality is due to Pinsker's inequality, which shows $V_r^{\pi^\star_\tau}(\rho) - V_r^{\pi_t}(\rho) \leq O(\sqrt{\epsilon})$. For the term (i), if we take $\pi = \pi^\star$ in (5), then,

$$
V_r^{\pi^\star}(\rho) - \tau\mathcal{H}(\pi^\star_\tau) \leq V_r^{\pi^\star_\tau}(\rho) + \lambda^\star_\tau\left(V_g^{\pi^\star_\tau}(\rho) - V_g^{\pi^\star}(\rho)\right).
$$

Meanwhile, if we take $\lambda = 0$ in (5), then $\lambda^\star_\tau V_g^{\pi^\star_\tau}(\rho) \leq 0$. We notice that the feasibility of $\pi^\star$ yields $V_g^{\pi^\star}(\rho) \geq 0$. Hence,

$$
\text{(i)} = V_r^{\pi^\star}(\rho) - V_r^{\pi^\star_\tau}(\rho) \leq \tau\mathcal{H}(\pi^\star_\tau).
$$

We now substitute the upper bounds of (i) and (ii) above into (20) to obtain $V_r^{\pi^\star}(\rho) - V_r^{\pi_t}(\rho) \leq O(\sqrt{\epsilon})$, where we take $\tau = \Theta(\epsilon)$.

Second, we have

$$-V_g^{\pi_t}(\rho) \;=\; \underbrace{-V_g^{\pi_\tau^\star}(\rho)}_{\text{(iii)}} + \underbrace{V_g^{\pi_\tau^\star}(\rho) - V_g^{\pi_t}(\rho)}_{\text{(iv)}}. \tag{21}$$

Similar to bounding $V_r^{\pi_\tau^\star}(\rho) - V_r^{\pi_t}(\rho)$, we can show that (iv) $\leq O(\sqrt{\epsilon})$. Let $\lambda_{\max} := \frac{1}{(1-\gamma)\xi}$. For the term (iii), if we take $\lambda = \lambda_{\max}$ in (5), then,

$$-(\lambda_{\max} - \lambda_\tau^\star)V_g^{\pi_\tau^\star}(\rho) \;\leq\; \frac{\tau}{2}(\lambda_{\max})^2.$$

By the definition $\lambda_\tau^\star := \mathrm{argmin}_{\lambda \in \Lambda}\left\{\lambda V_g^{\pi_\tau^\star}(\rho) + \frac{\tau}{2}\lambda^2\right\}$, there are three values for $\lambda_\tau^\star$ to take: $-\frac{V_g^{\pi_\tau^\star}(\rho)}{\tau}$, or 0, or $\lambda_{\max}$ as follows

(1) When $0 < -\frac{V_g^{\pi_\tau^\star}(\rho)}{\tau} < \lambda_{\max}$, $\lambda_\tau^\star = -\frac{V_g^{\pi_\tau^\star}(\rho)}{\tau}$ which shows that $\lambda_{\max} - \lambda_\tau^\star > 0$;

(2) When $-\frac{V_g^{\pi_\tau^\star}(\rho)}{\tau} \leq 0$, $\lambda_\tau^\star = 0$ which shows that $\lambda_{\max} - \lambda_\tau^\star = \lambda_{\max} > 0$;

(3) When $-\frac{V_g^{\pi_\tau^\star}(\rho)}{\tau} \geq \lambda_{\max}$, $\lambda_\tau^\star = \lambda_{\max}$. In this case, using (5) with $\pi = \pi^\star$ leads to

$$\lambda_{\max}(V_g^{\pi^\star}(\rho) - V_g^{\pi_\tau^\star}(\rho)) \;\leq\; V_r^{\pi_\tau^\star}(\rho) - V_r^{\pi^\star}(\rho) + \tau\mathcal{H}(\pi_\tau^\star). \tag{22}$$

Meanwhile, for any saddle point $(\pi^\star, \lambda^\star) \in \Pi^\star \times \Lambda^\star$, $V_r^{\pi_\tau^\star}(\rho) - V_r^{\pi^\star}(\rho) \leq \lambda^\star(V_g^{\pi^\star}(\rho) - V_g^{\pi_\tau^\star}(\rho))$, which in conjunction with (22) and $V_g^{\pi^\star}(\rho) \geq 0$ yields,

$$-(\lambda_{\max} - \lambda^\star)V_g^{\pi_\tau^\star}(\rho) \;\leq\; \tau\mathcal{H}(\pi_\tau^\star).$$

It is easy to see that we can always take $\lambda^\star < \lambda_{\max}$. In fact, except for $\lambda^\star = 0$, we know that $\lambda^\star V_g^{\pi^\star}(\rho) \leq 0$ leads to $V_g^{\pi^\star}(\rho) = 0$. By the definition $\lambda^\star \in \mathrm{argmin}_{\lambda \in \Lambda} V_{r+\lambda g}^{\pi^\star}(\rho)$, any $\lambda^\star < \lambda_{\max}$ is a min-max point. Therefore,

$$\text{(iii)} \;\leq\; \frac{\tau}{(\lambda_{\max} - \lambda^\star)}\mathcal{H}(\pi_\tau^\star).$$

By combining three cases above, we conclude that (iii) $\leq O(\tau) = O(\epsilon)$.

We now substitute the upper bounds of (iii) and (iv) above into (21) to obtain $-V_g^{\pi_t}(\rho) \leq O(\sqrt{\epsilon})$.

Finally, we replace the accuracy $\sqrt{\epsilon}$ by $\epsilon$ and combine all big O notation to conclude the proof. □

### C.4 Zero constraint violation of RPG-PD (6)

**Corollary 15** (Zero constraint violation). *Let Assumption 1 hold. For small $\epsilon$, there exists $\delta > 0$ such that if we instead use the conservative constraint $V_{g'}^\pi(\rho) \geq 0$ for $g' = g - (1-\gamma)\delta$, and take $\eta = \Theta(\epsilon^4)$, $\tau = \Theta(\epsilon^2)$, and $\epsilon_0 = \epsilon$, then the policy iterates of RPG-PD (6) satisfy*

$$V_r^{\pi^\star}(\rho) - V_r^{\pi_t}(\rho) \;\leq\; \epsilon \;\; \text{and} \;\; -V_g^{\pi_t}(\rho) \;\leq\; 0 \;\; \text{for any } t = \Omega\left(\frac{1}{\epsilon^6}\log^2\frac{1}{\epsilon}\right)$$

*where $\Omega(\cdot)$ only has some problem-dependent constant.*

*Proof.* We apply the conservatism to the translated constraint $V_g^\pi(\rho) \geq 0$ in Problem (1). Specifically, for any $\delta < \min(\xi, 1)$, we let $g' := g - (1-\gamma)\delta$ and define a conservative constraint,

$$V_{g'}^\pi \;:=\; V_g^\pi(\rho) - \delta \;\geq\; 0.$$

It is straightforward to see that Assumption 1 ensures that $V_{g'}^{\pi_t}(\rho) \geq 0$ is strictly feasible for a new slack variable $\xi' := \xi - \delta$. We now can apply RPG-PD (6) to a new regularized Lagrangian,

$$L_\tau'(\pi, \lambda) \;:=\; V_{r+\lambda g'}^\pi(\rho) + \tau\left(\mathcal{H}(\pi) + \frac{1}{2}\lambda^2\right)$$

and Corollary 3 holds if we replace $g$ in RPG-PD by $g'$. Thus,

$$V_r^{\pi_\delta^\star}(\rho) - V_r^{\pi_t}(\rho) \;\leq\; \epsilon \;\text{ and }\; -V_{g'}^{\pi_t}(\rho) \;\leq\; \epsilon \;\text{ for any } t = \Omega\left(\frac{1}{\epsilon^6}\log^2\frac{1}{\epsilon}\right)$$

where $\Omega(\cdot)$ hides some problem-dependent constant, and $\pi_\delta^\star$ is an optimal policy to the $\delta$-perturbed constrained policy optimization problem,

$$\underset{\pi \in \Pi}{\text{maximize }} V_r^\pi(\rho) \qquad \text{subject to } V_g^\pi(\rho) - \delta \;\geq\; 0. \tag{23}$$

We notice that the above $\Omega(\cdot)$ has $\xi'$-dependence and we denote it by $\Xi(\xi')$, where $\Xi\colon \mathbb{R}_+ \to \mathbb{R}_+$ is a positive function. Thus, we select $\delta$ such that $\delta \geq \epsilon\Xi(\xi')$, which is always possible for small enough $\epsilon$, for instance, $\delta = \frac{\xi}{2}$ and $\xi' = \frac{\xi}{2}$. Hence, if we take $\delta = \frac{\xi}{2}$ and such small $\epsilon$, then,

$$-V_{g'}^{\pi_t}(\rho) \;=\; -V_g^{\pi_t}(\rho) + \delta \;\leq\; \epsilon\Xi(\xi') \;\text{ for any } t = \Omega\left(\frac{1}{\epsilon^6}\log^2\frac{1}{\epsilon}\right)$$

which shows that $V_g^{\pi_t}(\rho) \geq 0$ for some large $t$.

The rest is to show that $V_r^{\pi^\star}(\rho) - V_r^{\pi_t}(\rho) \leq O(\epsilon)$. We notice that $\pi^\star$ is an optimal policy to Problem (23) when $\delta = 0$. Let $q^\star$ and $q_\delta^\star$ be associated occupancy measures of policies $\pi^\star$ and $\pi_\delta^\star$. In the occupancy measure space, Problem (23) becomes a linear program and it has a solution $q_\delta^\star$. Thus, we can view $q_\delta^\star$ as a $\delta$-perturbed solution of a convex optimization problem in which all functions are continuously differentiable and the domain is convex and compact. It is known from [129, Theorem 3.1] that the optimal solution $q_\delta^\star$ is continuous in $\delta$, which implies that for any $\epsilon > 0$, there exists $\delta' > 0$ such that $|\langle r, q^\star\rangle - \langle r, q_\delta^\star\rangle| \leq O(\epsilon)$ for any $\delta < \delta'$. Thus, $|V_r^{\pi^\star}(\rho) - V_r^{\pi_\delta^\star}(\rho)| \leq O(\epsilon)$ for small enough $\epsilon$. Therefore,

$$V_r^{\pi^\star}(\rho) - V_r^{\pi_t}(\rho) \;\leq\; V_r^{\pi_\delta^\star}(\rho) - V_r^{\pi_t}(\rho) + |V_r^{\pi^\star}(\rho) - V_r^{\pi_\delta^\star}(\rho)| \;\leq\; O(\epsilon)$$

for some large $t$. Collecting all conditions on $\delta$ leads to our final choice of $\delta = \min(\frac{\xi}{2}, 1, \delta')$. Finally, we combine all big O notation to complete the proof. $\qquad\square$

## C.5   Reduction of RPG-PD (6) as a NPG variant (7)

We introduce some useful notation in the regularized MDP [52]. Let $V_\tau^\pi(\rho) := V_{r+\lambda g}^\pi(\rho) + \tau\mathcal{H}(\pi)$. We introduce the soft-$Q$ value function $Q_\tau^\pi\colon S \times A \to \mathbb{R}$ and $V_\tau^\pi\colon S \to \mathbb{R}$ via Bellman equations,

$$Q_\tau^\pi(s,a) \;=\; r(s,a) + \lambda g(s,a) + \gamma\mathbb{E}_{s' \sim P(\cdot\,|\,s,a)}\left[V_\tau^\pi(s')\right]$$

$$V_\tau^\pi(s) \;=\; \mathbb{E}_{a \sim \pi(\cdot\,|\,s)}\left[-\tau\log\pi(a\,|\,s) + Q_\tau^\pi(s,a)\right].$$

We also define $A_\tau^\pi(s,a) := Q_\tau^\pi(s,a) - \tau\log\pi(a\,|\,s) - V_\tau^\pi(s)$. Hence,

$$Q_{r+\lambda g+\tau\psi}^\pi(s,a) \;=\; Q_\tau^\pi(s,a) - \tau\log\pi(a\,|\,s)$$

$$A_{r+\lambda g+\tau\psi}^\pi(s,a) \;=\; Q_{r+\lambda g+\tau\psi}^\pi(s,a) - V_{r+\lambda g+\tau\psi}^\pi(s)$$

where $\psi(s,a) := -\log\pi(a\,|\,s)$ for all $(s,a) \in S \times A$.

Setting $\epsilon_0 = 0$, it is easy to show that RPG-PD (6) is a case of (7) in the tabular case by introducing the softmax policy that is widely used in policy optimization. A softmax policy $\pi_\theta\colon S \to \Delta(A)$ is parametrized by a parameter $\theta \in \mathbb{R}^{|S||A|}$ via a softmax function,

$$\pi_\theta(a\,|\,s) \;=\; \frac{\exp(\theta_{s,a})}{\sum_{a'}\exp(\theta_{s,a'})} \;\text{ for all } (s,a) \in S \times A.$$

With a slight abuse of notation, we also use notation $\pi_\theta$ as a vector in $\mathbb{R}^{|S||A|}$.

**Lemma 16.** *Set $\epsilon_0 = 0$. Under the softmax policy parametrization, RPG-PD (6) is equivalent to (7).*

*Proof.* Dual update (7b) is straightforward. We next show the equivalence for the primal update by applying the softmax function to both sides of Primal update (7a).

We first notice that $F_\rho(\theta_t)^\dagger \cdot \nabla_\theta L_\tau(\pi_{\theta_t}, \lambda_t) = F_\rho(\theta_t)^\dagger \cdot \nabla_\theta V_\tau^{\pi_\theta}(\rho)$. Thus,

$$
\begin{aligned}
\exp(\theta_{t+1,s,a}) &= \exp(\theta_{t,s,a}) \exp\left(\eta(1-\gamma)(F_\rho(\theta_t)^\dagger \cdot \nabla_\theta L_\tau(\pi_{\theta_t}, \lambda_t))_{s,a}\right) \\
&= \exp(\theta_{t,s,a}) \exp\left(\eta A_\tau^{\pi_{\theta_t}}(s,a) + \eta c(s)\right)
\end{aligned}
$$

where $c(s)$ is an action-independent constant, and the second equality is the property of natural policy gradient (e.g., Lemma 29). Hence, after normalization over actions and some re-arrangement, we have

$$
\begin{aligned}
\pi_{\theta_{t+1}}(a \,|\, s) &= \pi_{\theta_t}(a \,|\, s) \frac{\exp\left(\eta A_\tau^{\pi_{\theta_t}}(s,a) + \eta c(s)\right)}{\sum_{a'} \pi_{\theta_t}(a' \,|\, s) \exp\left(\eta A_\tau^{\pi_{\theta_t}}(s,a') + \eta c(s)\right)} \\
&= \pi_{\theta_t}(a \,|\, s) \frac{\exp\left(\eta Q_\tau^{\pi_{\theta_t}}(s,a) - \eta\tau \log \pi_{\theta_t}(a \,|\, s)\right)}{\sum_{a'} \pi_{\theta_t}(a' \,|\, s) \exp\left(\eta Q_\tau^{\pi_{\theta_t}}(s,a') - \eta\tau \log \pi_{\theta_t}(a' \,|\, s)\right)} \\
&= \pi_{\theta_t}(a \,|\, s) \frac{\exp\left(\eta Q_{r+\lambda_t g + \tau \psi_t}^{\pi_{\theta_t}}(s,a)\right)}{\sum_{a'} \pi_{\theta_t}(a' \,|\, s) \exp\left(\eta Q_{r+\lambda_t g + \tau \psi_t}^{\pi_{\theta_t}}(s,a')\right)}
\end{aligned}
$$

which is an explicit form of the policy update in (6). Since the above derivation holds in both directions, the proof is complete. □

## C.6 Proof of Theorem 4

*Proof.* We utilize the decomposition of the primal-dual gap as in (19) and analyze the term (i) and the term (ii), separately.

For the term (i), we have

$$
L_\tau(\pi_\tau^\star, \lambda_t) - L_\tau(\pi_{\theta_t}, \lambda_t)
$$

$$
= V_{r+\lambda_t g}^{\pi_\tau^\star}(\rho) - V_{r+\lambda_t g}^{\pi_{\theta_t}}(\rho)
$$
$$
- \frac{\tau}{1-\gamma} \sum_{s,a} d_\rho^{\pi_\tau^\star}(s) \pi_\tau^\star(a \mid s) \log \pi_\tau^\star(a \mid s) + \frac{\tau}{1-\gamma} \sum_{s,a} d_\rho^{\pi_{\theta_t}}(s) \pi_{\theta_t}(a \mid s) \log \pi_{\theta_t}(a \mid s)
$$

$$
= V_{r+\lambda_t g+\tau\psi_t}^{\pi_\tau^\star}(\rho) - V_{r+\lambda_t g+\tau\psi_t}^{\pi_{\theta_t}}(\rho) - \tau V_{\psi_t}^{\pi_\tau^\star}(\rho) + \tau V_{\psi_t}^{\pi_{\theta_t}}(\rho)
$$
$$
- \frac{\tau}{1-\gamma} \sum_{s,a} d_\rho^{\pi_\tau^\star}(s) \pi_\tau^\star(a \mid s) \log \pi_\tau^\star(a \mid s) + \frac{\tau}{1-\gamma} \sum_{s,a} d_\rho^{\pi_{\theta_t}}(s) \pi_{\theta_t}(a \mid s) \log \pi_{\theta_t}(a \mid s)
$$

$$
= \frac{1}{1-\gamma} \sum_{s,a} d_\rho^{\pi_\tau^\star}(s) \left( \pi_\tau^\star(a \mid s) - \pi_{\theta_t}(a \mid s) \right) Q_{r+\lambda_t g+\tau\psi_t}^{\pi_{\theta_t}}(s,a)
$$
$$
+ \frac{\tau}{1-\gamma} \sum_{s,a} d_\rho^{\pi_\tau^\star}(s) \pi_\tau^\star(a \mid s) \log \pi_{\theta_t}(a \mid s) - \frac{\tau}{1-\gamma} \sum_{s,a} d_\rho^{\pi_{\theta_t}}(s) \pi_{\theta_t}(a \mid s) \log \pi_{\theta_t}(a \mid s)
$$
$$
- \frac{\tau}{1-\gamma} \sum_{s,a} d_\rho^{\pi_\tau^\star}(s) \pi_\tau^\star(a \mid s) \log \pi_\tau^\star(a \mid s) + \frac{\tau}{1-\gamma} \sum_{s,a} d_\rho^{\pi_{\theta_t}}(s) \pi_{\theta_t}(a \mid s) \log \pi_{\theta_t}(a \mid s)
$$

$$
= \frac{1}{1-\gamma} \sum_{s,a} d_\rho^{\pi_\tau^\star}(s) \left( \pi_\tau^\star(a \mid s) - \pi_{\theta_t}(a \mid s) \right) \phi_{s,a}^\top w_t - \frac{\tau}{1-\gamma} \sum_s d_\rho^{\pi_\tau^\star}(s) \mathrm{KL}_t(s)
$$
$$
+ \frac{1}{1-\gamma} \sum_{s,a} d_\rho^{\pi_\tau^\star}(s) \left( \pi_\tau^\star(a \mid s) - \pi_{\theta_t}(a \mid s) \right) \left( Q_{r+\lambda_t g+\tau\psi_t}^{\pi_{\theta_t}}(s,a) - \phi_{s,a}^\top w_t \right)
$$

$$
\leq \sum_s d_\rho^{\pi_\tau^\star}(s) \left( \frac{\mathrm{KL}_t(s) - \mathrm{KL}_{t+1}(s)}{\eta(1-\gamma)} \right) + \eta(C_W)^2 - \frac{\tau}{1-\gamma} \sum_s d_\rho^{\pi_\tau^\star}(s) \mathrm{KL}_t(s)
$$
$$
+ \frac{1}{1-\gamma} \sum_{s,a} d_\rho^{\pi_\tau^\star}(s) \left( \pi_\tau^\star(a \mid s) - \pi_{\theta_t}(a \mid s) \right) \left( Q_{r+\lambda_t g+\tau\psi_t}^{\pi_{\theta_t}}(s,a) - \phi_{s,a}^\top w_t \right)
$$

$$
= \sum_s d_\rho^{\pi_\tau^\star}(s) \left( \frac{(1-\eta\tau)\mathrm{KL}_t(s) - \mathrm{KL}_{t+1}(s)}{\eta(1-\gamma)} \right) + \eta(C_W)^2
$$
$$
+ \frac{1}{1-\gamma} \sum_{s,a} d_\rho^{\pi_\tau^\star}(s) \left( \pi_\tau^\star(a \mid s) - \pi_{\theta_t}(a \mid s) \right) \left( Q_{r+\lambda_t g+\tau\psi_t}^{\pi_{\theta_t}}(s,a) - \phi_{s,a}^\top w_t \right)
$$

$$
= \frac{(1-\eta\tau)\mathrm{KL}_t(\rho) - \mathrm{KL}_{t+1}(\rho)}{\eta(1-\gamma)} + \eta(C_W)^2
$$
$$
+ \frac{1}{1-\gamma} \sum_{s,a} d_\rho^{\pi_\tau^\star}(s) \left( \pi_\tau^\star(a \mid s) - \pi_{\theta_t}(a \mid s) \right) \left( Q_{r+\lambda_t g+\tau\psi_t}^{\pi_{\theta_t}}(s,a) - \phi_{s,a}^\top w_t \right),
$$

where the first two equalities are because of the entropy regularization $\mathcal{H}(\pi)$, we apply performance difference lemma in Lemma 28 and $\psi_t(s,a) = -\log \pi_{\theta_t}(a \mid s)$ to the third equality, the inequality is due to an application of Lemma 27 with $x^\star = \pi_\tau^\star(\cdot \mid s)$, $x = \pi_{\theta_t}(\cdot \mid s)$, $g = -\phi_{s,a}^\top w_t$, and $\eta \leq 1/C_W$, where $|\phi_{s,a}^\top w_t/(1-\gamma)| \leq 2W/(1-\gamma) := C_W$. Moreover, the cross term has the following decomposition,

$$
\sum_{s,a} d_\rho^{\pi_\tau^\star}(s) \left( \pi_\tau^\star(a \mid s) - \pi_{\theta_t}(a \mid s) \right) \left( Q_{r+\lambda_t g+\tau\psi_t}^{\pi_{\theta_t}}(s,a) - \phi_{s,a}^\top w_t \right)
$$
$$
= \underbrace{\sum_{s,a} d_\rho^{\pi_\tau^\star}(s) \pi_\tau^\star(a \mid s) \left( Q_{r+\lambda_t g+\tau\psi_t}^{\pi_{\theta_t}}(s,a) - \phi_{s,a}^\top w_t^\star \right)}_{(a)} + \underbrace{\sum_{s,a} d_\rho^{\pi_\tau^\star}(s) \pi_{\theta_t}(a \mid s) \phi_{s,a}^\top (w_t - w_t^\star)}_{(b)}
$$
$$
+ \underbrace{\sum_{s,a} d_\rho^{\pi_\tau^\star}(s) \pi_\tau^\star(a \mid s) \phi_{s,a}^\top (w_t^\star - w_t)}_{(c)} + \underbrace{\sum_{s,a} d_\rho^{\pi_\tau^\star}(s) \pi_{\theta_t}(a \mid s) \left( \phi_{s,a}^\top w_t^\star - Q_{r+\lambda_t g+\tau\psi_t}^{\pi_{\theta_t}}(s,a) \right)}_{(d)}.
$$

We now deal with the four terms (a), (b), (c), and (d), separately. For the term (a),

$$
\begin{aligned}
|(a)| &\leq \sum_{s,a} d_\rho^{\pi_\tau^\star}(s)\pi_\tau^\star(a\mid s)\left|Q_{r+\lambda_t g+\tau\psi_t}^{\pi_{\theta_t}}(s,a)-\phi_{s,a}^\top w_t^\star\right| \\
&\leq \sqrt{\sum_{s,a}\frac{(d_\rho^{\pi_\tau^\star}(s)\pi_\tau^\star(a\mid s))^2}{d_\rho^{\pi_\tau^\star}(s)\mathrm{Unif}_A(a)}\sum_{s,a}d_\rho^{\pi_\tau^\star}(s)\mathrm{Unif}_A(a)\left(Q_{r+\lambda_t g+\tau\psi_t}^{\pi_{\theta_t}}(s,a)-\phi_{s,a}^\top w_t^\star\right)^2} \\
&= \sqrt{\sum_{s,a}\frac{(d_\rho^{\pi_\tau^\star}(s)\pi_\tau^\star(a\mid s))^2}{d_\rho^{\pi_\tau^\star}(s)\mathrm{Unif}_A(a)}\mathcal{E}_Q(w_t^\star,\theta_t,\nu^\star)} \\
&\leq \sqrt{\sum_{s,a}\frac{d_\rho^{\pi_\tau^\star}(s)\pi_\tau^\star(a\mid s)}{\mathrm{Unif}_A(a)}\mathcal{E}_Q(w_t^\star,\theta_t,\nu^\star)} \\
&= \sqrt{|A|\mathcal{E}_Q(w_t^\star,\theta_t,\nu^\star)},
\end{aligned}
$$

where we recall the definition of $\mathcal{E}_Q(w_t^\star,\theta_t,\nu^\star)$. Similarly, we can bound the term (d) by $|(d)|\leq\sqrt{|A|\mathcal{E}_Q(w_t^\star,\theta_t,\nu^\star)}$. For the term (b),

$$
\begin{aligned}
|(b)| &\leq \sum_{s,a} d_\rho^{\pi_\tau^\star}(s)\pi_{\theta_t}(a\mid s)|\phi_{s,a}^\top(w_t-w_t^\star)| \\
&\leq \sqrt{\sum_{s,a}\frac{(d_\rho^{\pi_\tau^\star}(s)\pi_{\theta_t}(a\mid s))^2}{d_\rho^{\pi_\tau^\star}(s)\mathrm{Unif}_A(a)}\sum_{s,a}d_\rho^{\pi^\star}(s)\mathrm{Unif}_A(a)\left(\phi_{s,a}^\top(w_t-w_t^\star)\right)^2} \\
&= \sqrt{\sum_{s,a}\frac{(d_\rho^{\pi_\tau^\star}(s)\pi_{\theta_t}(a\mid s))^2}{d_\rho^{\pi_\tau^\star}(s)\mathrm{Unif}_A(a)}\|w_t-w_t^\star\|_{\Sigma_{\nu^\star}}^2} \\
&\leq \sqrt{\sum_{s,a}\frac{d_\rho^{\pi_\tau^\star}(s)\pi_{\theta_t}(a\mid s)}{\mathrm{Unif}_A(a)}\|w_t-w_t^\star\|_{\Sigma_{\nu^\star}}^2} \\
&\leq \sqrt{|A|\,\|w_t-w_t^\star\|_{\Sigma_{\nu^\star}}^2} \\
&\leq \sqrt{|A|\kappa_\nu\,\|w_t-w_t^\star\|_{\Sigma_{d_{t,\nu}}}^2},
\end{aligned}
$$

where we recall the definition of $\kappa_\nu$ to obtain the last line. Similarly, we can bound the term (c) by $|(c)|\leq\sqrt{|A|\,\|w_t-w_t^\star\|_{\Sigma_{\nu^\star}}^2}$. Moreover, the optimality of $w_t^\star\in\arg\min_{w\in\mathbb{R}^d}\mathcal{E}_Q(w,\theta_t,d_{t,\nu})$ yields,

$$
(w-w_t^\star)^\top\nabla_w\mathcal{E}_Q(w_t^\star,\theta_t,d_{t,\nu})\geq 0,\quad\text{for any }\|w\|\leq W
$$

which further implies that for any $\|w\|\leq W$,

$$
\begin{aligned}
&\mathcal{E}_Q(w,\theta_t,d_{t,\nu})-\mathcal{E}_Q(w_t^\star,\theta_t,d_{t,\nu}) \\
&= \mathbb{E}_{(s,a)\sim d_{t,\nu}}\left[\left(\phi_{s,a}^\top w-\phi_{s,a}^\top w_t^\star+\phi_{s,a}^\top w_t^\star-Q_{r+\lambda_t g+\tau\psi_t}^{\pi_{\theta_t}}(s,a)\right)^2\right]-\mathcal{E}_Q(w_t^\star,\theta_t,d_{t,\nu}) \\
&= \mathbb{E}_{(s,a)\sim d_{t,\nu}}\left[\left(\phi_{s,a}^\top w-\phi_{s,a}^\top w_t^\star\right)^2\right]+2(w-w_t^\star)^\top\mathbb{E}_{(s,a)\sim d_{t,\nu}}\left[\phi_{s,a}\left(\phi_{s,a}^\top w_t^\star-Q_{r+\lambda_t g+\tau\psi_t}^{\pi_{\theta_t}}(s,a)\right)\right] \\
&= \mathbb{E}_{(s,a)\sim d_{t,\nu}}\left[\left(\phi_{s,a}^\top w-\phi_{s,a}^\top w_t^\star\right)^2\right]+2(w-w_t^\star)^\top\nabla_w\mathcal{E}_Q(w_t^\star,\theta_t,d_{t,\nu}) \\
&\geq \mathbb{E}_{(s,a)\sim d_{t,\nu}}\left[\left(\phi_{s,a}^\top w-\phi_{s,a}^\top w_t^\star\right)^2\right] \\
&= \|w-w_t^\star\|_{\Sigma_{d_{t,\nu}}}^2.
\end{aligned}
$$

Therefore,

$$
|(b)|\leq\sqrt{|A|\kappa_\nu\,\|w_t-w_t^\star\|_{\Sigma_{d_{t,\nu}}}^2}\leq\sqrt{|A|\kappa_\nu\left(\mathcal{E}_Q(w_t,\theta_t,d_{t,\nu})-\mathcal{E}_Q(w_t^\star,\theta_t,d_{t,\nu})\right)}.
$$

With a similar reasoning, we can bound the term (c) by

$$|(c)| \leq \sqrt{|A|\kappa_\nu \left(\mathcal{E}_Q(w_t, \theta_t, d_{t,\nu}) - \mathcal{E}_Q(w_t^\star, \theta_t, d_{t,\nu})\right)}.$$

By applying the upper bounds above to the cross term, and then taking expectation over the randomness in $w_t$, we have

$$
\begin{aligned}
&\mathbb{E}\left[L_\tau(\pi_\tau^\star, \lambda_t) - L_\tau(\pi_{\theta_t}, \lambda_t)\right] \\
\leq\ & \frac{(1-\eta\tau)\mathbb{E}[\mathrm{KL}_t(\rho)] - \mathbb{E}[\mathrm{KL}_{t+1}(\rho)]}{\eta} + \eta(C_W)^2 + 2\mathbb{E}\left[\sqrt{|A|\mathcal{E}_Q(w_t^\star, \theta_t, \nu^\star)}\right] \\
&+ 2\mathbb{E}\left[\sqrt{|A|\kappa_\nu \left(\mathcal{E}_Q(w_t, \theta_t, d_{t,\nu}) - \mathcal{E}_Q(w_t^\star, \theta_t, d_{t,\nu})\right)}\right] \\
\leq\ & \frac{(1-\eta\tau)\mathbb{E}[\mathrm{KL}_t(\rho)] - \mathbb{E}[\mathrm{KL}_{t+1}(\rho)]}{\eta} + \eta(C_W)^2 + 2\sqrt{|A|\mathbb{E}\left[\mathcal{E}_Q(w_t^\star, \theta_t, \nu^\star)\right]} \\
&+ 2\sqrt{|A|\kappa_\nu \mathbb{E}\left[\mathcal{E}_Q(w_t, \theta_t, d_{t,\nu}) - \mathcal{E}_Q(w_t^\star, \theta_t, d_{t,\nu})\right]} \\
\leq\ & \frac{(1-\eta\tau)\mathbb{E}[\mathrm{KL}_t(\rho)] - \mathbb{E}[\mathrm{KL}_{t+1}(\rho)]}{\eta} + \eta(C_W)^2 + 2\sqrt{|A|\epsilon_{\mathrm{bias}}} + 2\sqrt{|A|\kappa_\nu \epsilon_{\mathrm{stat}}}.
\end{aligned}
$$

Similarly, for the term (ii),

$$
\begin{aligned}
&L_\tau(\pi_{\theta_t}, \lambda_t) - L_\tau(\pi_{\theta_t}, \lambda_\tau^\star) \\
=\ & V_{r+\lambda_t g}^{\pi_{\theta_t}}(\rho) - V_{r+\lambda_\tau^\star g}^{\pi_{\theta_t}}(\rho) + \frac{1}{2}\tau(\lambda_t)^2 - \frac{1}{2}\tau(\lambda_\tau^\star)^2 \\
=\ & (\lambda_t - \lambda_\tau^\star)V_g^{\pi_{\theta_t}}(\rho) + \frac{1}{2}\tau(\lambda_t)^2 - \frac{1}{2}\tau(\lambda_\tau^\star)^2 \\
=\ & (\lambda_t - \lambda_\tau^\star)\left(V_g^{\pi_{\theta_t}}(\rho) + \tau\lambda_t\right) - \frac{1}{2}\tau(\lambda_t - \lambda_\tau^\star)^2 \\
\leq\ & \frac{(\lambda_\tau^\star - \lambda_t)^2 - (\lambda_\tau^\star - \lambda_{t+1})^2}{2\eta} + \frac{1}{2}\eta(C_{\tau,\xi}')^2 - \frac{1}{2}\tau(\lambda_t - \lambda_\tau^\star)^2 \\
=\ & \frac{(1-\eta\tau)(\lambda_\tau^\star - \lambda_t)^2 - (\lambda_\tau^\star - \lambda_{t+1})^2}{2\eta} + \frac{1}{2}\eta(C_{\tau,\xi}')^2
\end{aligned}
$$

where the inequality is due to the standard descent lemma [128] and $V_g^{\pi_{\theta_t}}(\rho) + \tau\lambda_t \leq \frac{1}{1-\gamma}(1 + \frac{\tau}{\xi}) := C_{\tau,\xi}'$. By taking expectation over the randomness in $w_t$,

$$\mathbb{E}\left[L_\tau(\pi_{\theta_t}, \lambda_t) - L_\tau(\pi_{\theta_t}, \lambda_\tau^\star)\right] \leq \frac{(1-\eta\tau)\mathbb{E}\left[(\lambda_\tau^\star - \lambda_t)^2\right] - \mathbb{E}\left[(\lambda_\tau^\star - \lambda_{t+1})^2\right]}{2\eta} + \frac{1}{2}\eta(C_{\tau,\xi}')^2.$$

Using the definition $\mathbb{E}\left[\Phi_t\right] := \mathbb{E}\left[\mathrm{KL}_t(\rho)\right] + \frac{1}{2}\mathbb{E}\left[(\lambda_\tau^\star - \lambda_t)^2\right]$, we combine the two inequalities above to show that,

$$
\begin{aligned}
\mathbb{E}\left[\Phi_{t+1}\right] \leq\ & (1-\eta\tau)\mathbb{E}\left[\Phi_t\right] - \eta\mathbb{E}\left[L_\tau(\pi_\tau^\star, \lambda_t) - L_\tau(\pi_{\theta_t}, \lambda_\tau^\star)\right] + \eta^2 \max\left((C_W)^2, (C_{\tau,\xi}')^2\right) \\
& + 2\eta\sqrt{|A|\epsilon_{\mathrm{bias}}} + 2\eta\sqrt{|A|\kappa_\nu \epsilon_{\mathrm{stat}}} \\
\leq\ & (1-\eta\tau)\mathbb{E}\left[\Phi_t\right] + \eta^2 \max\left((C_W)^2, (C_{\tau,\xi}')^2\right) + 2\eta\sqrt{|A|\epsilon_{\mathrm{bias}}} + 2\eta\sqrt{|A|\kappa_\nu \epsilon_{\mathrm{stat}}},
\end{aligned}
$$

where the second inequality is due to Lemma 14. If we expand the inequality above recursively, then,

$$
\begin{aligned}
\mathbb{E}\left[\Phi_{t+1}\right] \;\leq\; & (1-\eta\tau)\mathbb{E}\left[\Phi_t\right] + \eta^2 \max\left((C_W)^2, (C'_{\tau,\xi})^2\right) + 2\eta\sqrt{|A|\epsilon_{\text{bias}}} + 2\eta\sqrt{|A|\kappa_\nu\epsilon_{\text{stat}}} \\
\leq\; & (1-\eta\tau)^2 \mathbb{E}\left[\Phi_{t-1}\right] + \left(\eta^2 + \eta^2(1-\eta\tau)\right)\max\left((C_W)^2, (C'_{\tau,\xi})^2\right) \\
& + 2\eta\left(1+(1-\eta\tau)\right)\left(\sqrt{|A|\epsilon_{\text{bias}}} + 2\sqrt{|A|\kappa_\nu\epsilon_{\text{stat}}}\right) \\
\leq\; & \cdots \\
\leq\; & (1-\eta\tau)^t \mathbb{E}\left[\Phi_1\right] + \left(\eta^2\left(1+(1-\eta\tau)+(1-\eta\tau)^2+\ldots\right)\right)\max\left((C_W)^2, (C'_{\tau,\xi})^2\right) \\
& + 2\eta\left(1+(1-\eta\tau)+(1-\eta\tau)^2+\ldots\right)\left(\sqrt{|A|\epsilon_{\text{bias}}} + \sqrt{|A|\kappa_\nu\epsilon_{\text{stat}}}\right) \\
\leq\; & (1-\eta\tau)^t \mathbb{E}\left[\Phi_1\right] + \frac{\eta}{\tau}\max\left((C_W)^2, (C'_{\tau,\xi})^2\right) + \frac{2}{\tau}\left(\sqrt{|A|\epsilon_{\text{bias}}} + \sqrt{|A|\kappa_\nu\epsilon_{\text{stat}}}\right) \\
\leq\; & \mathrm{e}^{-\eta\tau t}\mathbb{E}\left[\Phi_1\right] + \frac{\eta}{\tau}\max\left((C_W)^2, (C'_{\tau,\xi})^2\right) + \frac{2}{\tau}\left(\sqrt{|A|\epsilon_{\text{bias}}} + \sqrt{|A|\kappa_\nu\epsilon_{\text{stat}}}\right)
\end{aligned}
$$

which completes the proof. $\qquad\square$

## C.7 Proof of Corollary 5

*Proof.* The proof is similar to the proof of Corollary 3, except that we take the expectation over the randomness of computing $w_t$ via a sample-based algorithm.

According to Theorem 4 and $\epsilon_{\text{stat}} = O(\epsilon^4)$, $\epsilon_{\text{stat}} = O(\epsilon^4)$, if we take $\tau = \Theta(\epsilon)$ and $\eta = \Theta(\epsilon^2)$, then $\mathbb{E}[\Phi_{t+1}] = O(\epsilon)$ for any $t = \Omega\left(\frac{1}{\epsilon^3}\log\frac{1}{\epsilon}\right)$. We next consider a primal-dual iterate $(\pi_{\theta_t}, \lambda_t)$ for some $t = \Omega\left(\frac{1}{\epsilon^3}\log\frac{1}{\epsilon}\right)$. It is straightforward to check that

$$
\mathbb{E}\left[\text{KL}_t(\rho)\right] \;=\; O(\epsilon) \quad \text{and} \quad \frac{1}{2}\mathbb{E}\left[(\lambda_\tau^\star - \lambda_t)^2\right] \;=\; O(\epsilon).
$$

First, we have

$$
\mathbb{E}\left[V_r^{\pi^\star}(\rho) - V_r^{\pi_{\theta_t}}(\rho)\right] \;=\; \underbrace{V_r^{\pi^\star}(\rho) - V_r^{\pi_\tau^\star}(\rho)}_{\text{(i)}} + \underbrace{\mathbb{E}\left[V_r^{\pi_\tau^\star}(\rho) - V_r^{\pi_{\theta_t}}(\rho)\right]}_{\text{(ii)}}. \tag{24}
$$

For the term (ii), because of $\text{KL}_t(\rho) = O(\epsilon)$, we have

$$
\begin{aligned}
\text{(ii)} \;=\; & \mathbb{E}\left[\frac{1}{1-\gamma}\sum_{s,a} d_\rho^{\pi_\tau^\star}(s)\left(\pi_\tau^\star(a\,|\,s) - \pi_{\theta_t}(a\,|\,s)\right) Q_r^{\pi_{\theta_t}}(s,a)\right] \\
\leq\; & \frac{1}{(1-\gamma)^2}\sum_s d_\rho^{\pi_\tau^\star}(s)\mathbb{E}\left[\|\pi_\tau^\star(\cdot\,|\,s) - \pi_{\theta_t}(\cdot\,|\,s)\|_1\right] \\
\leq\; & \frac{1}{(1-\gamma)^2}\sum_s d_\rho^{\pi_\tau^\star}(s)\sqrt{2\,\mathbb{E}\left[\text{KL}_t(s)\right]} \\
\leq\; & \frac{1}{(1-\gamma)^2}\sqrt{2\sum_s d_\rho^{\pi_\tau^\star}(s)\mathbb{E}\left[\text{KL}_t(s)\right]} \\
=\; & \frac{1}{(1-\gamma)^2}\sqrt{2\,\mathbb{E}\left[\text{KL}_t(\rho)\right]}
\end{aligned}
$$

where we use Cauchy–Schwarz inequality in the first and third inequalities, and the second inequality is due to Pinsker's inequality and Jensen's inequality, which shows $\mathbb{E}\left[V_r^{\pi_\tau^\star}(\rho) - V_r^{\pi_{\theta_t}}(\rho)\right] \leq O(\sqrt{\epsilon})$. For the term (i), if we take $\pi = \pi^\star$ in (5), then,

$$
V_r^{\pi^\star}(\rho) - \tau\mathcal{H}(\pi_\tau^\star) \;\leq\; V_r^{\pi_\tau^\star}(\rho) + \lambda_\tau^\star\left(V_g^{\pi_\tau^\star}(\rho) - V_g^{\pi^\star}(\rho)\right).
$$

Meanwhile, if we take $\lambda = 0$ in (5), then $\lambda_\tau^\star V_g^{\pi_\tau^\star}(\rho) \leq 0$. We notice that the feasibility of $\pi^\star$ yields $V_g^{\pi^\star}(\rho) \geq 0$. Hence,

$$(\text{i}) \quad = \quad V_r^{\pi^\star}(\rho) - V_r^{\pi_\tau^\star}(\rho) \quad \leq \quad \tau\mathcal{H}(\pi_\tau^\star).$$

We now substitute the upper bounds of (i) and (ii) into (24) to obtain $\mathbb{E}\left[V_r^{\pi^\star}(\rho) - V_r^{\pi_{\theta_t}}(\rho)\right] \leq O(\sqrt{\epsilon})$, where we take $\tau = \Theta(\epsilon)$.

Second, we have

$$\mathbb{E}\left[-V_g^{\pi_{\theta_t}}(\rho)\right] \quad = \quad \underbrace{-V_g^{\pi_\tau^\star}(\rho)}_{(\text{iii})} + \underbrace{\mathbb{E}\left[V_g^{\pi_\tau^\star}(\rho) - V_g^{\pi_{\theta_t}}(\rho)\right]}_{(\text{iv})}. \tag{25}$$

Similar to bounding $\mathbb{E}\left[V_r^{\pi_\tau^\star}(\rho) - V_r^{\pi_{\theta_t}}(\rho)\right]$, we can show that $\mathbb{E}\left[(\text{iv})\right] \leq O(\sqrt{\epsilon})$. For the term (iii), we can show that $(\text{iii}) \leq O(\tau) = O(\epsilon)$ in a similar way as dealing with (iii) in (21).

We now substitute the upper bounds of (iii) and (iv) above into (25) to obtain $\mathbb{E}\left[-V_g^{\pi_{\theta_t}}(\rho)\right] \leq O(\sqrt{\epsilon})$.

Finally, we replace the accuracy $\sqrt{\epsilon}$ by $\epsilon$ and combine all big O notation to conclude the proof. $\quad\square$

## C.8 Zero constraint violation of inexact RPG-PD

**Corollary 17** (Zero constraint violation). *Let Assumptions 1–2 hold and $\epsilon_{\text{stat}}, \epsilon_{\text{bias}} = O(\epsilon^8)$ for small $\epsilon$, $\epsilon_0 > 0$. For small $\epsilon$, there exists $\delta > 0$ such that if we instead use the conservative constraint $V_{g'}^\pi(\rho) \geq 0$ for $g' = g - (1-\gamma)\delta$, and take the stepsize $\eta = \Theta(\epsilon^4)$ and $\tau = \Theta(\epsilon^2)$, then the policy iterates of inexact RPG-PD satisfy*

$$\mathbb{E}\left[V_r^{\pi^\star}(\rho) - V_r^{\pi_{\theta_t}}(\rho)\right] \leq \epsilon \text{ and } \mathbb{E}\left[-V_g^{\pi_{\theta_t}}(\rho)\right] \leq 0 \text{ for any } t = \Omega\left(\frac{1}{\epsilon^6}\log^2\frac{1}{\epsilon}\right)$$

*where $\Omega(\cdot)$ only has some problem-dependent constant.*

*Proof.* We apply the conservatism to the translated constraint $V_g^\pi(\rho) \geq 0$ in Problem (1). Specifically, for any $\delta < \min(\xi, 1)$, we let $g' := g - (1-\gamma)\delta$ and define a conservative constraint,

$$V_{g'}^\pi \quad := \quad V_g^\pi(\rho) - \delta \quad \geq \quad 0.$$

It is straightforward to see that Assumption 1 ensures that $V_{g'}^{\pi_t}(\rho) \geq 0$ is strictly feasible for a new slack variable $\xi' := \xi - \delta$. We now can apply inexact RPG-PD (8) to a new regularized Lagrangian,

$$L_\tau'(\pi, \lambda) \quad := \quad V_{r+\lambda g'}^\pi(\rho) + \tau\left(\mathcal{H}(\pi) + \frac{1}{2}\lambda^2\right)$$

and Corollary 5 holds if we replace $g$ in inexact RPG-PD by $g'$ and $\epsilon_{\text{stat}}, \epsilon_{\text{bias}} = O(\epsilon^8)$ for small $\epsilon > 0$. Thus,

$$\mathbb{E}\left[V_r^{\pi_\delta^\star}(\rho) - V_r^{\pi_{\theta_t}}(\rho)\right] \leq \epsilon \text{ and } \mathbb{E}\left[-V_{g'}^{\pi_{\theta_t}}(\rho)\right] \leq \epsilon \text{ for any } t = \Omega\left(\frac{1}{\epsilon^6}\log^2\frac{1}{\epsilon}\right)$$

where $\Omega(\cdot)$ has some problem-dependent constant, and $\pi_\delta^\star$ is an optimal policy to the $\delta$-perturbed constrained policy optimization problem,

$$\underset{\pi \in \Pi}{\text{maximize}} \ V_r^\pi(\rho) \qquad \text{subject to } V_g^\pi(\rho) - \delta \geq 0. \tag{26}$$

We notice that the above $\Omega(\cdot)$ has $\xi'$-dependence and we denote it by $\Xi(\xi')$, where $\Xi\colon \mathbb{R}_+ \to \mathbb{R}_+$ is a positive function. Thus, we select $\delta$ such that $\delta \geq \epsilon\Xi(\xi')$, which is always possible for small enough $\epsilon$, for instance, $\delta = \frac{\xi}{2}$ and $\xi' = \frac{\xi}{2}$. Hence, if we take $\delta = \frac{\xi}{2}$ and such small $\epsilon$, then

$$\mathbb{E}\left[-V_{g'}^{\pi_{\theta_t}}(\rho)\right] \quad = \quad \mathbb{E}\left[-V_g^{\pi_{\theta_t}}(\rho)\right] + \delta \quad \leq \quad \epsilon\Xi(\xi') \text{ for any } t = \Omega\left(\frac{1}{\epsilon^6}\log^2\frac{1}{\epsilon}\right)$$

which shows that $\mathbb{E}\left[V_g^{\pi_{\theta_t}}(\rho)\right] \geq 0$ for some large $t$.

The rest is to show that $\mathbb{E}\left[V_r^{\pi^\star}(\rho) - V_r^{\pi_{\theta_t}}(\rho)\right] \le O(\epsilon)$. We notice that $\pi^\star$ is an optimal policy to Problem (26) when $\delta = 0$. Let $q^\star$ and $q_\delta^\star$ be associated occupancy measures of policies $\pi^\star$ and $\pi_\delta^\star$. In the occupancy measure space, Problem (26) becomes a linear program and it has a solution $q_\delta^\star$. Thus, we can view $q_\delta^\star$ as a $\delta$-perturbed solution of a convex optimization problem in which all functions are continuous differentiable and the domain is convex and compact. It is known from [129, Theorem 3.1] that the optimal solution $q_\delta^\star$ is continuous in $\delta$, which implies that for any $\epsilon > 0$, there exists $\delta'$ such that $|\langle r, q^\star \rangle - \langle r, q_\delta^\star \rangle| \le O(\epsilon)$ for any $\delta < \delta'$. Thus, $|V_r^{\pi^\star}(\rho) - V_r^{\pi_\delta^\star}(\rho)| \le O(\epsilon)$ for small enough $\epsilon$. Therefore,

$$\mathbb{E}\left[V_r^{\pi^\star}(\rho) - V_r^{\pi_{\theta_t}}(\rho)\right] \;\le\; \mathbb{E}\left[V_r^{\pi_\delta^\star}(\rho) - V_r^{\pi_{\theta_t}}(\rho)\right] + |V_r^{\pi^\star}(\rho) - V_r^{\pi_\delta^\star}(\rho)| \;\le\; O(\epsilon)$$

for some large $t$. Collecting all conditions on $\delta$ leads to our final choice of $\delta = \min(\frac{\xi}{2}, 1, \delta')$. Finally, we combine all big O notation to complete the proof. $\square$

### C.9  Sample-based inexact RPG-PD algorithm

We generalize the inexact RPG-PD to be a sample-based algorithm that only takes sample-based estimates. We propose a sample-based RPG-PD with linear function approximation as follows,

$$\pi_{\theta_{t+1}}(\cdot \,|\, s) \;=\; \underset{\pi(\cdot \,|\, s) \in \hat{\Delta}(A)}{\operatorname{argmax}} \left\{ \sum_a \pi(a \,|\, s)\phi_{s,a}^\top \hat{w}_t - \frac{1}{\eta}\,\mathrm{KL}(\pi(\cdot \,|\, s), \pi_{\theta_t}(\cdot \,|\, s)) \right\} \tag{27a}$$

$$\lambda_{t+1} \;=\; \underset{\lambda \in \Lambda}{\operatorname{argmin}} \left\{ \lambda\left(\widehat{V}_g^{\pi_{\theta_t}}(\rho) + \tau\lambda_t\right) + \frac{1}{2\eta}(\lambda - \lambda_t)^2 \right\}, \tag{27b}$$

where $\hat{w}_t$ and $\hat{V}_g^{\pi_t}(\rho)$ are the sample-based estimates of NPG directions and value functions. It is standard to assume that there exists a policy simulator that generates policy rollouts for any given policies [49]. At time $t$, we can estimate $\hat{w}_t$ by solving the regression problem $\mathcal{E}_Q(w, \theta_t, d_{t,\nu}) = \mathbb{E}_{(s,a) \sim d_{t,\nu}}[(\phi_{s,a}^\top w - Q^{\pi_{\theta_t}}(s,a))^2]$ with $Q^{\pi_{\theta_t}}(s,a) = Q_{r+\lambda_t g + \tau\psi_t}^{\pi_{\theta_t}}(s,a)$ via a projected stochastic gradient descent (SGD) method,

$$w_t^{k+1} \;=\; \mathcal{P}_{\|w\| \le W}\left(w_t^k - \alpha\, G_t^k\right)$$

where $k \ge 0$ counts the number of SGD iterations, and $G_t^k$ is a sample-based estimate of the population gradient $\nabla_w \mathcal{E}_Q(w, \theta_t, d_{t,\nu})$,

$$G_t^k \;=\; 2\left(\phi_{s,a}^\top w_t^k - \hat{Q}_{r+\lambda_t g + \tau\psi_t}^{\pi_{\theta_t}}(s,a)\right)\phi_{s,a}.$$

From the projected SGD result [130], we use a weighted average $\frac{2}{K(K+1)}\sum_{k=0}^{K-1}(k+1)w_t^k$ as our $\hat{w}_t$. We note that $Q_{r+\lambda_t g + \tau\psi_t}^{\pi_{\theta_t}}(s,a)$ is a sum of a soft-$Q$ value function associated with a composite function $r + \lambda_t g - \tau\log\pi_{\theta_t}$, and $-\tau\log\pi_{\theta_t}$. Thus, we can estimate the value function $\hat{Q}_{r+\lambda_t g + \tau\psi_t}^{\pi_{\theta_t}}(s,a)$ using policy rollouts in Algorithm 2, which provides an unbiased estimate and it has bounded variance [131],

$$\mathbb{E}\left[\hat{Q}_{r+\lambda_t g + \tau\psi_t}^{\pi_{\theta_t}}(s,a) \,\big|\, s,a\right] \;=\; Q_{r+\lambda_t g + \tau\psi_t}^{\pi_{\theta_t}}(s,a) \;\text{ and }\; \mathbb{E}\left[G_t^k\right] \;=\; \nabla_w \mathcal{E}_Q(w_t^k, \theta_t, d_{t,\nu})$$

where the expectation $\mathbb{E}$ is taken over the randomness of drawing $(s,a) \sim d_{t,\nu}$. Another value function $\hat{V}_g^{\pi_{\theta_t}}(\rho)$ can be estimated using policy rollouts in Algorithm 3, $\mathbb{E}\left[\hat{V}_g^{\pi_{\theta_t}}(\rho)\right] = V_g^{\pi_{\theta_t}}(\rho)$, which is also unbiased and has bounded variance [23]. Hence, simply replacing all population quantities in inexact RPG-PD by their sample-based estimates leads to a sample-based inexact RPG-PD that is detailed in Algorithm 1.

We are now ready to establish the sample complexity of Algorithm 1 by exploiting the projected SGD result [130].

**Corollary 18** (Sample complexity of inexact RPG-PD)**.** *Let Assumptions 1–2 hold. Assume $\Sigma_\nu = \mathbb{E}_{(s,a) \sim \nu}\left[\phi_{s,a}\phi_{s,a}^\top\right] \ge \kappa_0 I$ for any state-action distribution $\nu$ and some $\kappa_0 > 0$. If we take the*

**Algorithm 1** Sample-based inexact RPG-PD algorithm with log-linear policy parametrization

1: **Input**: Learning rate $\eta$, number of SGD iterations $K$, SGD learning rate $\alpha$.
2: Initialize $\theta_0 = 0$, $\lambda_0 = 0$,
3: **for** $t = 0, \ldots, T - 1$ **do**
4:     Initialize $w_t^0 = 0$.
5:     **for** $k = 0, 1, \ldots, K - 1$ **do**
6:         Estimate $\hat{Q}_{r+\lambda_t g+\tau\psi_t}^{\pi_{\theta_t}}(s, a)$ for some $(s, a) \sim d_{t,\nu}$, using Algorithm 2 with policy $\pi_{\theta_t}$.
7:         Perform projected SGD step with $\alpha_k = \frac{2}{\kappa_0(k+2)}$,

$$w_t^{k+1} \;=\; \mathcal{P}_{\|w\| \leq W}\left(w_t^k - 2\alpha_k\left(\phi_{s,a}^\top w_t^k - \hat{Q}_{r+\lambda_t g+\tau\psi_t}^{\pi_{\theta_t}}(s, a)\right)\phi_{s,a}\right).$$

8:     **end for**
9:     Set $\hat{w}_t = \frac{2}{K(K+1)}\sum_{k=0}^{K-1}(k+1)w_t^k$.
10:    Estimate $\hat{V}_g^{\pi_{\theta_t}}(\rho)$ using Algorithm 3 with policy $\pi_{\theta_t}$.
11:    Perform inexact RPG-PD update,

$$\begin{aligned}
\pi_{\theta_{t+1}}(\cdot \mid s) &= \underset{\pi(\cdot \mid s) \in \hat{\Delta}(A)}{\operatorname{argmax}}\left\{\sum_a \pi(a \mid s)\phi_{s,a}^\top \hat{w}_t - \frac{1}{\eta}\operatorname{KL}(\pi(\cdot \mid s), \pi_{\theta_t}(\cdot \mid s))\right\} \\
\lambda_{t+1} &= \mathcal{P}_\Lambda\left((1 - \eta\tau)\lambda_t - \eta\hat{V}_g^{\pi_{\theta_t}}(\rho)\right).
\end{aligned}$$

12: **end for**

---

**Algorithm 2** Unbiased estimate $Q$

1: **Input**: Initial state-action distribution $\nu$, policy $\pi$, dual variable $\lambda$, regularization parameter $\tau$, discount factor $\gamma$.
2: Sample $(s_0, a_0) \sim \nu$, execute the policy $\pi$ with probability $\gamma$ at each step $h$; otherwise, accept $(s_h, a_h)$ as the sample.
3: Start with $(s_h, a_h)$, execute the policy $\pi$ with the termination probability $1 - \sqrt{\gamma}$. Once terminated, add all composite values $\gamma^{(k-h)/2}(r + \lambda g + \tau\psi)$ from step $k = h + 1$ onwards and $-\tau\log\pi(a_h \mid s_h)$ as $\hat{Q}_{r+\lambda g+\tau\psi}^\pi(s_h, a_h)$.
4: **Output**: $(s_h, a_h)$ and $\hat{Q}_{r+\lambda g+\tau\psi}^\pi(s_h, a_h)$.

---

**Algorithm 3** Unbiased estimate $V$

1: **Input**: Initial state distribution $\rho$, policy $\pi$, discount factor $\gamma$.
2: Sample $s_0 \sim \rho$, execute the policy $\pi$ with the termination probability $1 - \gamma$. Once terminated, add all utilities up as $\hat{V}_g^\pi(\rho)$.
3: **Output**: $\hat{V}_g^\pi(\rho)$.

*stepsize $\eta \le 1/C_W$, then the primal-dual iterates of sample-based inexact RPG-PD in Algorithm 1 satisfy*

$$\mathbb{E}[\,\Phi_{t+1}\,] \quad \le \quad \mathrm{e}^{-\eta\tau t}\mathbb{E}[\,\Phi_1\,] + \frac{\eta}{\tau}\max\big((C_W)^2, (C'_{\tau,\xi})^2\big) + \frac{C_{W,\xi,\tau,\epsilon_0}}{\tau}\sqrt{\frac{|A|\kappa_\nu}{K+1}} + \frac{2}{\tau}\sqrt{|A|\epsilon_{\mathrm{bias}}}$$

*where $C_W := 2W/(1-\gamma)$, $C_{W,\xi,\tau,\epsilon_0} := 8(W + 2/(\xi(1-\sqrt{\gamma})^2) + \tau(2\log|A| + |\log\epsilon_0|)/((1-\sqrt{\gamma})^2\xi))/\sqrt{\kappa_0}$, and $C'_{\tau,\xi} := (1 + \tau/\xi)/(1-\gamma)$.*

*Proof.* The proof is based on the proof of Theorem 4. Additionally, we have to take the randomness of sample-based estimates into account and bound the statistical error $\epsilon_{\mathrm{stat}}$ using the projected SGD result [130]. We first check all conditions of the projected SGD [130] are indeed satisfied by the SGD step in Algorithm 1: (i) The domain $\|w\| \le W$ is convex and bounded; (ii) The gradient $G_t^k$ is an unbiased estimate of the population gradient; (iii) The minimizer of $\mathcal{E}_Q(w, \theta_t, d_{t,\nu})$ is unique, since $\Sigma_{d_{t,\nu}} \ge \kappa_0 I$ for some $\kappa_0 > 0$; (iv) The squared norm of the estimated gradient $G_t^k$ is bounded or the gradient has bounded variance. To show (iv), it is sufficient to check that

$$\mathbb{E}\left[\,\big\|\phi_{s,a}^\top\phi_{s,a}w_t^k\big\|^2\,\right] \quad \le \quad \mathbb{E}\left[\,\big\|\phi_{s,a}^\top\phi_{s,a}\big\|^2\,\big\|w_t^k\big\|^2\,\right] \quad \le \quad W^2$$

$$\mathbb{E}\left[\,\left\|\hat{Q}_{r+\lambda_t g+\tau\psi_t}^{\pi_{\theta_t}}(s,a)\phi_{s,a}\right\|^2\,\right] \quad \le \quad \mathbb{E}\left[\,\left(\hat{Q}_{r+\lambda_t g+\tau\psi_t}^{\pi_{\theta_t}}(s,a)\right)^2\|\phi_{s,a}\|^2\,\right]$$

$$\le \quad \mathbb{E}\left[\,\left(\hat{Q}_{r+\lambda_t g+\tau\psi_t}^{\pi_{\theta_t}}(s,a)\right)^2\,\right]$$

where we use the boundedness of $\|w_t^k\| \le W$ and $\|\phi_{s,a}\| \le 1$. We notice that $|r + \lambda_t g| \le \frac{2}{(1-\gamma)\xi}$. By [131, Lemma 3.5], we know that

$$\mathbb{E}\left[\,\left(\hat{Q}_{r+\lambda_t g+\tau\psi_t}^{\pi_{\theta_t}}(s,a)\right)^2\,\right] \quad \le \quad 4\left(\frac{2/((1-\gamma)\xi)}{1-\sqrt{\gamma}}\right)^2 + 4\left(\frac{\tau\log|A|}{1-\sqrt{\gamma}}\right)^2 + 2\left(\tau\log\frac{\epsilon_0}{|A|}\right)^2$$

where the first two terms in the upper bound is due to the variance of soft-$Q$ value function, and the last term is due to $|\log\pi_{\theta_t}| \le |\log\frac{\epsilon_0}{|A|}|$ for $\pi_{\theta_t} \in \hat{\Delta}(A)$. Hence, the estimated gradient $G_t^k$ has bounded second-order moment (or variance), which verifies (iv). From the projected SGD result [130], if the SGD stepsize $\alpha_k = \frac{2}{k+2}$, then

$$\mathbb{E}\left[\,\mathcal{E}_Q(\hat{w}_t, \theta_t, d_{t,\nu}) - \mathcal{E}_Q(w_t^\star, \theta_t, d_{t,\nu})\,\right] \quad \le \quad \frac{16\left(W + \frac{2}{\xi(1-\sqrt{\gamma})^2} + \frac{\tau(2\log|A|+|\log\epsilon_0|)}{(1-\sqrt{\gamma})^2\xi}\right)^2}{\kappa_0(K+1)}$$

which leads to $\sqrt{\epsilon_{\mathrm{stat}}} \le \frac{4(W+2/(\xi(1-\sqrt{\gamma})^2)+\tau(2\log|A|+|\log\epsilon_0|)/((1-\sqrt{\gamma})^2\xi))}{\sqrt{\kappa_0(K+1)}}$. Substituting the bound of $\sqrt{\epsilon_{\mathrm{stat}}}$ into the proof of Theorem 4 yields our desired result. $\square$

Corollary 18 states a similar result as Theorem 4. The effect of using sample-based estimates to update inexact RPG-PD appears as the number $K$ of SGD steps at each time $t$. Thus, we can interpret the iteration complexity in Corollary 5 in terms of the number of SGD steps by taking $\epsilon_{\mathrm{stat}} = O(\epsilon^8)$, i.e., $K = \Omega(\frac{1}{\epsilon^8})$. Thus, whenever $\epsilon_{\mathrm{bias}} = O(\epsilon^8)$ for small $\epsilon > 0$, if we take $(\eta, \tau)$ in Corollary 5 and $K = \Omega(\frac{1}{\epsilon^8})$ for Algorithm 1, then,

$$\mathbb{E}\left[\,V_r^{\pi^\star}(\rho) - V_r^{\pi_{\theta_t}}(\rho)\,\right] \le \epsilon \quad \text{and} \quad \mathbb{E}\left[\,-V_g^{\pi_{\theta_t}}(\rho)\,\right] \le \epsilon \quad \text{for any } t = \Omega\left(\frac{1}{\epsilon^6}\log^2\frac{1}{\epsilon}\right)$$

where $\Omega(\cdot)$ only has some problem-dependent constant. In other words, the total number of policy rollouts or sampled trajectories $tK = \Omega\left(\frac{1}{\epsilon^{14}}\right)$ is required for Algorithm 1 to output an $\epsilon$-optimal constrained policy. Furthermore, the zero constraint violation in Corollary 17 can be interpreted similarly. When $\epsilon$ is small enough, we can design a conservative constraint such that the policy iterates of Algorithm 1 satisfy $V_r^{\pi^\star}(\rho) - V_r^{\pi_t}(\rho) \le \epsilon$ and $V_g^{\pi_{\theta_t}}(\rho) \ge 0$ for some $t$, $K$ that satisfy $tK = \Omega\left(\frac{1}{\epsilon^{14}}\right)$. This appears to be the first sample-based zero constraint violation result for constrained MDPs in the function approximation setting. We leave achieving the optimal sample complexity as future work.

# D  Proofs in Section 4

In this section, we provide proofs for the claims in Section 4.

## D.1  Preliminary last-iterate analysis of OPG-PD (9)

To measure the proximity of the primal-dual iterates of OPG-PD (9) to the optimal pair $(\pi^\star, \lambda^\star)$, we introduce the following two distance metrics $\Theta_t$ and $\zeta_t$ at time $t \geq 1$,

$$
\begin{aligned}
\Theta_t &:= \frac{1}{2(1-\gamma)} \sum_s d_\rho^{\pi^\star}(s) \|\hat{\pi}_t(\cdot\,|\,s) - \mathcal{P}_{\Pi^\star}(\hat{\pi}_t(\cdot\,|\,s))\|^2 + \frac{1}{2}(\hat{\lambda}_t - \mathcal{P}_{\Pi^\star}(\hat{\lambda}_t))^2 \\
&\quad + \frac{1}{16(1-\gamma)} \sum_s d_\rho^{\pi^\star}(s) \|\hat{\pi}_t(\cdot\,|\,s) - \pi_{t-1}(\cdot\,|\,s)\|^2 + \frac{1}{16}(\hat{\lambda}_t - \lambda_{t-1})^2
\end{aligned}
$$

$$
\begin{aligned}
\zeta_t &:= \frac{1}{2(1-\gamma)} \sum_s d_\rho^{\pi^\star}(s) \|\hat{\pi}_{t+1}(\cdot|s) - \pi_t(\cdot|s)\|^2 + \frac{1}{2}(\hat{\lambda}_{t+1} - \lambda_t)^2 \\
&\quad + \frac{1}{2(1-\gamma)} \sum_s d_\rho^{\pi^\star}(s) \|\pi_t(\cdot|s) - \hat{\pi}_t(\cdot|s)\|^2 + \frac{1}{2}(\lambda_t - \hat{\lambda}_t)^2
\end{aligned}
$$

and a problem-dependent constant $\iota$,

$$
\iota := \max\left( \frac{2\kappa_\rho^2 |A|}{(1-\gamma)^6}, \frac{8\gamma^2\sqrt{|A|}\kappa_\rho}{(1-\gamma)^6}\left(1 + \frac{1}{(1-\gamma)^2\xi^2}\right), \frac{4}{(1-\gamma)^3} \right). \tag{28}
$$

**Lemma 19.** *Let the optimal state visitation distribution be unique, i.e., $d_\rho^\pi = d_\rho^{\pi^\star}$ for any $\pi \in \Pi^\star$. If we set the stepsize $\eta \leq 1/(4\sqrt{\iota})$, then the primal-dual iterates of OPG-PD (9) satisfy*

$$
\Theta_{t+1} \leq \Theta_t - \frac{7}{16}\zeta_t \quad \text{for all } t \geq 1. \tag{29}
$$

*Proof.* We begin with the standard decomposition of the primal-dual gap at time $t \geq 1$,

$$
V_{r+\lambda_t g}^{\pi^\star}(\rho) - V_{r+\lambda^\star g}^{\pi_t}(\rho) = \underbrace{V_{r+\lambda_t g}^{\pi^\star}(\rho) - V_{r+\lambda_t g}^{\pi_t}(\rho)}_{(i)} + \underbrace{V_{r+\lambda_t g}^{\pi_t}(\rho) - V_{r+\lambda^\star g}^{\pi_t}(\rho)}_{(ii)} \tag{30}
$$

and we next deal with (i) and (ii), separately.

For the first term (i),

$$
\begin{aligned}
&V_{r+\lambda_t g}^{\pi^\star}(\rho) - V_{r+\lambda_t g}^{\pi_t}(\rho) \\
&= \frac{1}{1-\gamma} \sum_{s,a} d_\rho^{\pi^\star}(s)(\pi^\star(a\,|\,s) - \pi_t(a\,|\,s))Q_{r+\lambda_t g}^{\pi_t}(s,a) \\
&= \frac{1}{1-\gamma} \sum_{s,a} d_\rho^{\pi^\star}(s)(\pi^\star(a\,|\,s) - \hat{\pi}_{t+1}(a\,|\,s))Q_{r+\lambda_t g}^{\pi_t}(s,a) \\
&\quad + \frac{1}{1-\gamma} \sum_{s,a} d_\rho^{\pi^\star}(s)(\hat{\pi}_{t+1}(a\,|\,s) - \pi_t(a\,|\,s))Q_{r+\lambda_{t-1}g}^{\pi_{t-1}}(s,a) \\
&\quad + \frac{1}{1-\gamma} \sum_{s,a} d_\rho^{\pi^\star}(s)(\hat{\pi}_{t+1}(a\,|\,s) - \pi_t(a\,|\,s))\left(Q_{r+\lambda_t g}^{\pi_t}(s,a) - Q_{r+\lambda_{t-1}g}^{\pi_{t-1}}(s,a)\right) \\
&\leq \frac{1}{2\eta(1-\gamma)} \sum_s d_\rho^{\pi^\star}(s)\left(\|\pi^\star(\cdot\,|\,s) - \hat{\pi}_t(\cdot\,|\,s)\|^2 - \|\pi^\star(\cdot\,|\,s) - \hat{\pi}_{t+1}(\cdot\,|\,s)\|^2 - \|\hat{\pi}_{t+1}(\cdot\,|\,s) - \hat{\pi}_t(\cdot\,|\,s)\|^2\right) \\
&\quad + \frac{1}{2\eta(1-\gamma)} \sum_s d_\rho^{\pi^\star}(s)\left(\|\hat{\pi}_{t+1}(\cdot\,|\,s) - \hat{\pi}_t(\cdot\,|\,s)\|^2 - \|\hat{\pi}_{t+1}(\cdot\,|\,s) - \pi_t(\cdot\,|\,s)\|^2 - \|\pi_t(\cdot|s) - \hat{\pi}_t(\cdot|s)\|^2\right) \\
&\quad + \frac{1}{1-\gamma} \sum_s d_\rho^{\pi^\star}(s)(\hat{\pi}_{t+1}(a\,|\,s) - \pi_t(a\,|\,s))\left(Q_{r+\lambda_t g}^{\pi_t}(s,a) - Q_{r+\lambda_{t-1}g}^{\pi_{t-1}}(s,a)\right)
\end{aligned}
$$

$$\tag{31}$$

where the first equality is due to performance difference lemma in Lemma 28, the second equality is because of some re-arrangement, the inequality is from an application of Lemma 24 to $\hat{\pi}_{t+1}(\cdot \mid s)$ and $\pi_t(\cdot \mid s)$:

$$
\sum_a (\pi^\star(a \mid s) - \hat{\pi}_{t+1}(a \mid s)) Q^{\pi_t}_{r+\lambda_t g}(s, a)
$$
$$
\leq \quad \frac{1}{2\eta} \left( \|\pi^\star(\cdot|s) - \hat{\pi}_t(\cdot|s)\|^2 - \|\pi^\star(\cdot|s) - \hat{\pi}_{t+1}(\cdot|s)\|^2 - \|\hat{\pi}_{t+1}(\cdot|s) - \hat{\pi}_t(\cdot|s)\|^2 \right)
$$

$$
\sum_a (\hat{\pi}_{t+1}(a \mid s) - \pi_t(a \mid s)) Q^{\pi_{t-1}}_{r+\lambda_{t-1} g}(s, a)
$$
$$
\leq \quad \frac{1}{2\eta} \left( \|\hat{\pi}_{t+1}(\cdot|s) - \hat{\pi}_t(\cdot|s)\|^2 - \|\hat{\pi}_{t+1}(\cdot|s) - \pi_t(\cdot|s)\|^2 - \|\pi_t(\cdot|s) - \hat{\pi}_t(\cdot|s)\|^2 \right).
$$

Similarly, we deal with the second term (ii) with some re-arrangement, and apply Lemma 24 to $\hat{\lambda}_{t+1}$ and $\lambda_t$,

$$
\begin{aligned}
V^{\pi_t}_{r+\lambda_t g}(\rho) - V^{\pi_t}_{r+\lambda^\star g}(\rho) &= (\lambda_t - \lambda^\star) V^{\pi_t}_g(\rho) \\
&= (\lambda_t - \hat{\lambda}_{t+1}) V^{\pi_{t-1}}_g(\rho) + (\lambda_t - \hat{\lambda}_{t+1}) \left( V^{\pi_t}_g(\rho) - V^{\pi_{t-1}}_g(\rho) \right) \\
&\quad + (\hat{\lambda}_{t+1} - \lambda^\star) V^{\pi_t}_g(\rho) \\
&\leq \frac{1}{2\eta} \left( (\hat{\lambda}_{t+1} - \hat{\lambda}_t)^2 - (\hat{\lambda}_{t+1} - \lambda_t)^2 - (\lambda_t - \hat{\lambda}_t)^2 \right) \\
&\quad + (\lambda_t - \hat{\lambda}_{t+1}) \left( V^{\pi_t}_g(\rho) - V^{\pi_{t-1}}_g(\rho) \right) \\
&\quad + \frac{1}{2\eta} \left( (\lambda^\star - \hat{\lambda}_t)^2 - (\lambda^\star - \hat{\lambda}_{t+1})^2 - (\hat{\lambda}_{t+1} - \hat{\lambda}_t)^2 \right).
\end{aligned}
\tag{32}
$$

On the other hand, from Lemma 25, we have

$$
\|\hat{\pi}_{t+1}(\cdot \mid s) - \pi_t(\cdot \mid s)\|_1 \quad \leq \quad \eta \left\| Q^{\pi_t}_{r+\lambda_t g}(s, \cdot) - Q^{\pi_{t-1}}_{r+\lambda_{t-1} g}(s, \cdot) \right\|_\infty
\tag{33}
$$

and

$$
|\lambda_t - \hat{\lambda}_{t+1}| \quad \leq \quad \eta |V^{\pi_t}_g(\rho) - V^{\pi_{t-1}}_g(\rho)|.
\tag{34}
$$

Hence,

$$
\sum_{s,a} d^{\pi^\star}_\rho(s) (\hat{\pi}_{t+1}(a \mid s) - \pi_t(a \mid s)) \left( Q^{\pi_t}_{r+\lambda_t g}(s, a) - Q^{\pi_{t-1}}_{r+\lambda_{t-1} g}(s, a) \right)
$$
$$
\leq \quad \eta \sum_s d^{\pi^\star}_\rho(s) \left\| Q^{\pi_t}_{r+\lambda_t g}(s, \cdot) - Q^{\pi_{t-1}}_{r+\lambda_{t-1} g}(s, \cdot) \right\|^2_\infty
$$
$$
\leq \quad 2\eta \sum_s d^{\pi^\star}_\rho(s) \left( \left\| (\lambda_t - \lambda_{t-1}) Q^{\pi_t}_g(s, \cdot) \right\|^2_\infty + \left\| Q^{\pi_t}_{r+\lambda_{t-1} g}(s, \cdot) - Q^{\pi_{t-1}}_{r+\lambda_{t-1} g}(s, \cdot) \right\|^2_\infty \right)
$$
$$
\leq \quad \frac{2\eta}{(1-\gamma)^2} (\lambda_t - \lambda_{t-1})^2 + \frac{4\eta\gamma^2}{(1-\gamma)^4} \left( 1 + \frac{1}{(1-\gamma)^2 \xi^2} \right) \max_s \|\pi_t(\cdot \mid s) - \pi_{t-1}(\cdot \mid s)\|^2_1
$$
$$
\leq \quad \frac{2\eta}{(1-\gamma)^2} (\lambda_t - \lambda_{t-1})^2 + \frac{4\eta\gamma^2 \kappa_\rho}{(1-\gamma)^5} \left( 1 + \frac{1}{(1-\gamma)^2 \xi^2} \right) \sum_s d^{\pi^\star}_\rho(s) \|\pi_t(\cdot \mid s) - \pi_{t-1}(\cdot \mid s)\|^2_1
$$
$$
\leq \quad \frac{4\eta}{(1-\gamma)^2} \left( (\lambda_t - \hat{\lambda}_t)^2 + (\hat{\lambda}_t - \lambda_{t-1})^2 \right)
$$
$$
+ \frac{8\eta\gamma^2 \sqrt{|A|} \kappa_\rho}{(1-\gamma)^5} \left( 1 + \frac{1}{(1-\gamma)^2 \xi^2} \right) \sum_s d^{\pi^\star}_\rho(s) \left( \|\pi_t(\cdot \mid s) - \hat{\pi}_t(\cdot \mid s)\|^2 + \|\hat{\pi}_t(\cdot \mid s) - \pi_{t-1}(\cdot \mid s)\|^2 \right)
$$

where the first inequality is due to Cauchy–Schwarz inequality and (33), we subtract and add $Q^{\pi_t}_{r+\lambda_{t-1} g}$ and apply the inequality $(x+y)^2 \leq 2x^2 + 2y^2$ in the second inequality, the third inequality

is due to Lemma 22 and the boundedness of value functions and dual variables, the fourth inequality comes from the property of $\kappa_\rho$, and the last inequality is due to the inequality $(x+y)^2 \leq 2x^2 + 2y^2$ and the inequality $\|p - p'\|_1 \leq \sqrt{|A|}\|p - p'\|_2$ for two probability distributions $p$ and $p'$. Meanwhile, using a similar reasoning, we can derive that

$$
\begin{aligned}
&(\lambda_t - \hat{\lambda}_{t+1})\left(V_g^{\pi_t}(\rho) - V_g^{\pi_{t-1}}(\rho)\right) \\
&\leq \quad \eta\left(V_g^{\pi_t}(\rho) - V_g^{\pi_{t-1}}(\rho)\right)^2 \\
&\leq \quad \frac{\eta\kappa_\rho^2}{(1-\gamma)^6}\left(\sum_s d_\rho^{\pi^\star}(s)\|\pi_t(\cdot\,|\,s) - \pi_{t-1}(\cdot\,|\,s)\|_1\right)^2 \\
&\leq \quad \frac{\eta\kappa_\rho^2}{(1-\gamma)^6}\sum_s\left(\sqrt{d_\rho^{\pi^\star}(s)}\|\pi_t(\cdot\,|\,s) - \pi_{t-1}(\cdot\,|\,s)\|_1\right)^2 \\
&\leq \quad \frac{2\eta\kappa_\rho^2|A|}{(1-\gamma)^6}\sum_s d_\rho^{\pi^\star}(s)\left(\|\pi_t(\cdot\,|\,s) - \hat{\pi}_t(\cdot\,|\,s)\|^2 + \|\hat{\pi}_t(\cdot\,|\,s) - \pi_{t-1}(\cdot\,|\,s)\|^2\right)
\end{aligned}
$$

where the first inequality is due to Cauchy–Schwarz inequality and (34), the second inequality is due to Lemma 22, the third inequality is an application of Cauchy-Schwarz inequality, and the last inequality is due to the inequality $(x + y)^2 \leq 2x^2 + 2y^2$ and the inequality $\|p - p'\|_1 \leq \sqrt{|A|}\|p - p'\|_2$ for two probability distributions $p$ and $p'$.

We set notation,

$$
\iota \;:=\; \max\left(\frac{2\kappa_\rho^2|A|}{(1-\gamma)^6},\; \frac{8\gamma^2\sqrt{|A|}\kappa_\rho}{(1-\gamma)^6}\left(1 + \frac{1}{(1-\gamma)^2\xi^2}\right),\; \frac{4}{(1-\gamma)^3}\right).
$$

After applying the established inequalities above to (31) and (32), we combine them into (30) as,

$$
\begin{aligned}
&V_{r+\lambda_t g}^{\pi^\star}(\rho) - V_{r+\lambda^\star g}^{\pi_t}(\rho) \\
&\leq \quad \frac{1}{2\eta(1-\gamma)}\sum_s d_\rho^{\pi^\star}(s)\left(\|\pi^\star(\cdot\,|\,s) - \hat{\pi}_t(\cdot\,|\,s)\|^2 - \|\pi^\star(\cdot\,|\,s) - \hat{\pi}_{t+1}(\cdot\,|\,s)\|^2 - \|\hat{\pi}_{t+1}(\cdot\,|\,s) - \hat{\pi}_t(\cdot\,|\,s)\|^2\right) \\
&\quad + \frac{1}{2\eta(1-\gamma)}\sum_s d_\rho^{\pi^\star}(s)\left(\|\hat{\pi}_{t+1}(\cdot\,|\,s) - \hat{\pi}_t(\cdot\,|\,s)\|^2 - \|\hat{\pi}_{t+1}(\cdot\,|\,s) - \pi_t(\cdot\,|\,s)\|^2 - \|\pi_t(\cdot\,|\,s) - \hat{\pi}_t(\cdot\,|\,s)\|^2\right) \\
&\quad + \frac{1}{2\eta}\left((\hat{\lambda}_{t+1} - \hat{\lambda}_t)^2 - (\hat{\lambda}_{t+1} - \lambda_t)^2 - (\lambda_t - \hat{\lambda}_t)^2\right) \\
&\quad + \frac{1}{2\eta}\left((\lambda^\star - \hat{\lambda}_t)^2 - (\lambda^\star - \hat{\lambda}_{t+1})^2 - (\hat{\lambda}_{t+1} - \hat{\lambda}_t)^2\right) \\
&\quad + \eta\iota(\lambda_t - \hat{\lambda}_t)^2 + \eta\iota(\hat{\lambda}_t - \lambda_{t-1})^2 \\
&\quad + \eta\iota\sum_s d_\rho^{\pi^\star}(s)\left(\|\pi_t(\cdot\,|\,s) - \hat{\pi}_t(\cdot\,|\,s)\|^2 + \|\hat{\pi}_t(\cdot\,|\,s) - \pi_{t-1}(\cdot\,|\,s)\|^2\right).
\end{aligned}
$$

We notice that $V_{r+\lambda_t g}^{\pi^\star}(\rho) - V_{r+\lambda^\star g}^{\pi_t}(\rho) \geq 0$ and non-expansiveness of projection operators $\mathcal{P}_{\Pi^\star}$, $\mathcal{P}_{\Lambda^\star}$,

$$
\begin{aligned}
\|\mathcal{P}_{\Pi^\star}(\hat{\pi}_{t+1}(\cdot\,|\,s)) - \hat{\pi}_{t+1}(\cdot\,|\,s)\| &\leq \|\pi^\star(\cdot\,|\,s) - \hat{\pi}_{t+1}(\cdot\,|\,s)\| \\
\left\|\mathcal{P}_{\Lambda^\star}(\hat{\lambda}_{t+1}) - \hat{\lambda}_{t+1}\right\| &\leq \left\|\lambda^\star - \hat{\lambda}_{t+1}\right\|
\end{aligned}
$$

for any $\pi^\star \in \Pi^\star$ and $\lambda^\star \in \Lambda^\star$. If we take $\pi^\star = \mathcal{P}_{\Pi^\star}(\hat{\pi}_t(\cdot \,|\, s))$ and $\lambda^\star = \mathcal{P}_{\Lambda^\star}(\hat{\lambda}_t)$, then after some re-arrangement, we have

$$\frac{1}{2(1-\gamma)} \sum_s d_\rho^{\pi^\star}(s) \|\mathcal{P}_{\Pi^\star}(\hat{\pi}_{t+1}(\cdot \,|\, s)) - \hat{\pi}_{t+1}(\cdot \,|\, s)\|^2 \;+\; \frac{1}{2}(\mathcal{P}_{\Lambda^\star}(\hat{\lambda}_{t+1}) - \hat{\lambda}_{t+1})^2$$

$$\leq \quad \frac{1}{2(1-\gamma)} \sum_s d_\rho^{\pi^\star}(s) \|\mathcal{P}_{\Pi^\star}(\hat{\pi}_t(\cdot \,|\, s)) - \hat{\pi}_{t+1}(\cdot \,|\, s)\|^2 \;+\; \frac{1}{2}(\mathcal{P}_{\Lambda^\star}(\hat{\lambda}_t) - \hat{\lambda}_{t+1})^2$$

$$\leq \quad \frac{1}{2(1-\gamma)} \sum_s d_\rho^{\pi^\star}(s) \|\mathcal{P}_{\Pi^\star}(\hat{\pi}_t(\cdot \,|\, s)) - \hat{\pi}_t(\cdot|s)\|^2 \;+\; \frac{1}{2}(\mathcal{P}_{\Lambda^\star}(\hat{\lambda}_t) - \hat{\lambda}_t)^2$$

$$- \frac{1}{2(1-\gamma)} \sum_s d_\rho^{\pi^\star}(s) \|\hat{\pi}_{t+1}(\cdot \,|\, s) - \pi_t(\cdot \,|\, s)\|^2 \;-\; \frac{1}{2}(\hat{\lambda}_{t+1} - \lambda_t)^2$$

$$- \left(\frac{1}{2(1-\gamma)} - \eta^2 \iota\right) \sum_s d_\rho^{\pi^\star}(s) \|\pi_t(\cdot \,|\, s) - \hat{\pi}_t(\cdot \,|\, s)\|^2 \;-\; \left(\frac{1}{2} - \eta^2 \iota\right)(\lambda_t - \hat{\lambda}_t)^2$$

$$+ \eta^2 \iota \sum_s d_\rho^{\pi^\star}(s) \|\hat{\pi}_t(\cdot \,|\, s) - \pi_{t-1}(\cdot \,|\, s)\|^2 \;+\; \eta^2 \iota(\hat{\lambda}_t - \lambda_{t-1})^2$$

where we also use the assumption $d_\rho^{\pi^\star} = d_\rho^\pi$ for any $\pi \in \Pi^\star$. By taking $\eta^2 \iota \leq \frac{1}{16}$ and using notation $\Theta_t$ and $\zeta_t$, we have $\Theta_{t+1} \leq \Theta_t - \frac{7}{16}\zeta_t$. $\qquad\square$

To show the convergence, we next relate $\zeta_t$ with $\Theta_{t+1}$. An intermediate step is to show that $\zeta_t$ has the following lower bound.

**Lemma 20.** *Let $\kappa_{\rho,\gamma} := \max(\frac{\kappa_\rho}{1-\gamma}, 1)$. If we set the stepsize $\eta \leq \min(\frac{(1-\gamma)^3}{4|A|}, \frac{(1-\gamma)^3}{2\kappa_\rho})$, then the primal-dual iterates of OPG-PD (9) satisfy*

$$\sum_s d_\rho^{\pi^\star}(s) \left(\|\pi_t(\cdot \,|\, s) - \hat{\pi}_t(\cdot \,|\, s)\|^2 \;+\; \|\hat{\pi}_{t+1}(\cdot \,|\, s) - \pi_t(\cdot \,|\, s)\|^2\right) + \left(|\lambda_t - \hat{\lambda}_t|^2 + |\lambda_t - \hat{\lambda}_{t+1}|^2\right)$$

$$\geq \quad \frac{\eta^2}{9\kappa_{\rho,\gamma}^2} \frac{\left[V_{r+\hat{\lambda}_{t+1}g}^\pi(\rho) - V_{r+\lambda g}^{\hat{\pi}_{t+1}}(\rho)\right]_+^2}{\left(\max_s \|\pi(\cdot \,|\, s) - \hat{\pi}_{t+1}(\cdot \,|\, s)\| + |\lambda - \hat{\lambda}_{t+1}|\right)^2} \quad \textit{for all } (\pi, \lambda) \neq (\hat{\pi}_{t+1}, \hat{\lambda}_{t+1}).$$

*Proof.* From the optimality of $\hat{\pi}_{t+1}(\cdot \,|\, s)$ in OPG-PD, we know that for any $\pi \in \Pi$,

$$\langle \hat{\pi}_{t+1}(\cdot \,|\, s) - \hat{\pi}_t(\cdot \,|\, s), \pi(\cdot \,|\, s) - \hat{\pi}_{t+1}(\cdot \,|\, s)\rangle$$

$$\geq \quad \eta\langle Q_{r+\lambda_t g}^{\pi_t}(s, \cdot), \pi(\cdot \,|\, s) - \hat{\pi}_{t+1}(\cdot \,|\, s)\rangle$$

$$= \quad \eta\langle Q_{r+\hat{\lambda}_{t+1}g}^{\hat{\pi}_{t+1}}(s, \cdot), \pi(\cdot \,|\, s) - \hat{\pi}_{t+1}(\cdot \,|\, s)\rangle \;+\; \eta\langle Q_{r+\lambda_t g}^{\pi_t}(s, \cdot) - Q_{r+\lambda_t g}^{\hat{\pi}_{t+1}}(s, \cdot), \pi(\cdot \,|\, s) - \hat{\pi}_{t+1}(\cdot \,|\, s)\rangle$$

$$+ \eta\langle Q_{r+\lambda_t g}^{\hat{\pi}_{t+1}}(s, \cdot) - Q_{r+\hat{\lambda}_{t+1}g}^{\hat{\pi}_{t+1}}(s, \cdot), \pi(\cdot \,|\, s) - \hat{\pi}_{t+1}(\cdot \,|\, s)\rangle.$$

(35a)

Similarly, from the optimality of $\pi_{t+1}(\cdot \,|\, s)$ in OPG-PD, we know that for any $\pi \in \Pi$,

$$\langle \pi_{t+1}(\cdot \,|\, s) - \hat{\pi}_{t+1}(\cdot \,|\, s), \pi(\cdot \,|\, s) - \pi_{t+1}(\cdot \,|\, s)\rangle$$

$$\geq \quad \eta\langle Q_{r+\lambda_t g}^{\pi_t}(s, \cdot), \pi(\cdot \,|\, s) - \pi_{t+1}(\cdot \,|\, s)\rangle$$

$$= \quad \eta\langle Q_{r+\lambda_{t+1}g}^{\pi_{t+1}}(s, \cdot), \pi(\cdot \,|\, s) - \pi_{t+1}(\cdot \,|\, s)\rangle \;+\; \eta\langle Q_{r+\lambda_t g}^{\pi_t}(s, \cdot) - Q_{r+\lambda_t g}^{\pi_{t+1}}(s, \cdot), \pi(\cdot \,|\, s) - \pi_{t+1}(\cdot \,|\, s)\rangle$$

$$+ \eta\langle Q_{r+\lambda_t g}^{\pi_{t+1}}(s, \cdot) - Q_{r+\lambda_{t+1}g}^{\pi_{t+1}}(s, \cdot), \pi(\cdot \,|\, s) - \pi_{t+1}(\cdot \,|\, s)\rangle.$$

(35b)

On the other hand, from the optimality of $\hat{\lambda}_{t+1}$ in OPG-PD, we know that for any $\lambda \in \Lambda$,

$$
\begin{aligned}
(\hat{\lambda}_{t+1} - \hat{\lambda}_t)(\lambda - \hat{\lambda}_{t+1}) &\geq \eta V_g^{\pi_t}(\rho)(\hat{\lambda}_{t+1} - \lambda) \\
&= \eta V_g^{\hat{\pi}_{t+1}}(\rho)(\hat{\lambda}_{t+1} - \lambda) + \eta(V_g^{\pi_t}(\rho) - V_g^{\hat{\pi}_{t+1}}(\rho))(\hat{\lambda}_{t+1} - \lambda)
\end{aligned}
\tag{36a}
$$

and the optimality of $\lambda_{t+1}$ in OPG-PD yields,

$$
\begin{aligned}
(\lambda_{t+1} - \hat{\lambda}_t)(\lambda - \lambda_{t+1}) &\geq \eta V_g^{\pi_t}(\rho)(\lambda_{t+1} - \lambda) \\
&= \eta V_g^{\pi_{t+1}}(\rho)(\lambda_{t+1} - \lambda) + \eta(V_g^{\pi_t}(\rho) - V_g^{\pi_{t+1}}(\rho))(\lambda_{t+1} - \lambda).
\end{aligned}
\tag{36b}
$$

First, we take the expectation of (35a) over some state distribution $d_\rho^\pi$ on both sides and add it to (36a),

$$
\begin{aligned}
&\frac{1}{1-\gamma} \sum_s d_\rho^\pi(s) \langle \hat{\pi}_{t+1}(\cdot \mid s) - \hat{\pi}_t(\cdot \mid s), \pi(\cdot \mid s) - \hat{\pi}_{t+1}(\cdot \mid s) \rangle + (\hat{\lambda}_{t+1} - \hat{\lambda}_t)(\lambda - \hat{\lambda}_{t+1}) \\
\geq\ & \frac{\eta}{1-\gamma} \sum_s d_\rho^\pi(s) \langle Q_{r+\hat{\lambda}_{t+1}g}^{\hat{\pi}_{t+1}}(s, \cdot), \pi(\cdot \mid s) - \hat{\pi}_{t+1}(\cdot \mid s) \rangle \\
&+ \frac{\eta}{1-\gamma} \sum_s d_\rho^\pi(s) \langle Q_{r+\lambda_t g}^{\pi_t}(s, \cdot) - Q_{r+\lambda_t g}^{\hat{\pi}_{t+1}}(s, \cdot), \pi(\cdot \mid s) - \hat{\pi}_{t+1}(\cdot \mid s) \rangle \\
&+ \frac{\eta}{1-\gamma} \sum_s d_\rho^\pi(s) \langle Q_{r+\lambda_t g}^{\hat{\pi}_{t+1}}(s, \cdot) - Q_{r+\hat{\lambda}_{t+1}g}^{\hat{\pi}_{t+1}}(s, \cdot), \pi(\cdot \mid s) - \hat{\pi}_{t+1}(\cdot \mid s) \rangle \\
&+ \eta V_g^{\hat{\pi}_{t+1}}(\rho)(\hat{\lambda}_{t+1} - \lambda) + \eta(V_g^{\pi_t}(\rho) - V_g^{\hat{\pi}_{t+1}}(\rho))(\hat{\lambda}_{t+1} - \lambda) \\
\geq\ & \eta \left( V_{r+\hat{\lambda}_{t+1}g}^\pi(\rho) - V_{r+\hat{\lambda}_{t+1}g}^{\hat{\pi}_{t+1}}(\rho) \right) \\
&- \frac{\eta}{1-\gamma} \sum_s d_\rho^\pi(s) \left\| Q_{r+\lambda_t g}^{\pi_t}(s, \cdot) - Q_{r+\lambda_t g}^{\hat{\pi}_{t+1}}(s, \cdot) \right\|_\infty \|\pi(\cdot \mid s) - \hat{\pi}_{t+1}(\cdot \mid s)\|_1 \\
&- \frac{\eta}{1-\gamma} \sum_s d_\rho^\pi(s) \left\| Q_{r+\lambda_t g}^{\hat{\pi}_{t+1}}(s, \cdot) - Q_{r+\hat{\lambda}_{t+1}g}^{\hat{\pi}_{t+1}}(s, \cdot) \right\| \|\pi(\cdot \mid s) - \hat{\pi}_{t+1}(\cdot \mid s)\| \\
&+ \eta V_g^{\hat{\pi}_{t+1}}(\rho)(\hat{\lambda}_{t+1} - \lambda) - \eta |V_g^{\pi_t}(\rho) - V_g^{\hat{\pi}_{t+1}}(\rho)| |\hat{\lambda}_{t+1} - \lambda| \\
\geq\ & \eta \left( V_{r+\hat{\lambda}_{t+1}g}^\pi(\rho) - V_{r+\hat{\lambda}_{t+1}g}^{\hat{\pi}_{t+1}}(\rho) \right) + \eta V_g^{\hat{\pi}_{t+1}}(\rho)(\hat{\lambda}_{t+1} - \lambda) \\
&- \frac{\eta\gamma|A|}{(1-\gamma)^3} \max_s \|\pi_t(\cdot \mid s) - \hat{\pi}_{t+1}(\cdot \mid s)\| \sum_s d_\rho^\pi(s) \|\pi(\cdot \mid s) - \hat{\pi}_{t+1}(\cdot \mid s)\| \\
&- \frac{\eta}{(1-\gamma)^2} \sum_s d_\rho^\pi(s) |\lambda_t - \hat{\lambda}_{t+1}| \|\pi(\cdot \mid s) - \hat{\pi}_{t+1}(\cdot \mid s)\| \\
&- \frac{\eta\kappa_\rho\sqrt{|A|}}{(1-\gamma)^3} \sum_s d_\rho^{\pi^\star}(s) \|\pi_t(\cdot \mid s) - \hat{\pi}_{t+1}(\cdot \mid s)\| \left|\hat{\lambda}_{t+1} - \lambda\right|
\end{aligned}
\tag{37}
$$

where the second inequality is due to performance difference lemma in Lemma 28 and Cauchy-Schwarz inequality, and the last inequality results from Lemma 22 and the inequality $\|p - p'\|_1 \leq \sqrt{|A|} \|p - p'\|_2$ for two probability distributions $p$ and $p'$. By using the inequality $ac + bd \leq$

$(a+b)(c+d)$ for $a \geq 0$, $b \geq 0$, $c \geq 0$, and $d \geq 0$, with some re-arrangement, we have

$$\left( \max_s \|\pi(\cdot \mid s) - \hat{\pi}_{t+1}(\cdot \mid s)\| + |\lambda - \hat{\lambda}_{t+1}| \right) \times$$
$$\left[ \frac{\kappa_\rho}{1-\gamma} \sum_s d_\rho^{\pi^\star}(s) \left( \frac{1}{1-\gamma} \|\hat{\pi}_{t+1}(\cdot \mid s) - \hat{\pi}_t(\cdot \mid s)\| + \frac{2\eta|A|}{(1-\gamma)^3} \|\hat{\pi}_{t+1}(\cdot \mid s) - \pi_t(\cdot \mid s)\| \right) \right.$$
$$\left. + |\hat{\lambda}_{t+1} - \hat{\lambda}_t| + \frac{\eta\kappa_\rho}{(1-\gamma)^3} |\lambda_t - \hat{\lambda}_{t+1}| \right]$$

$$\geq \max_s \|\pi(\cdot \mid s) - \hat{\pi}_{t+1}(\cdot \mid s)\| \frac{\kappa_\rho}{1-\gamma} \sum_s d_\rho^{\pi^\star}(s) \times$$
$$\left( \frac{1}{1-\gamma} \|\hat{\pi}_{t+1}(\cdot \mid s) - \hat{\pi}_t(\cdot \mid s)\| + \frac{\eta\gamma|A|}{(1-\gamma)^3} \|\hat{\pi}_{t+1}(\cdot \mid s) - \pi_t(\cdot \mid s)\| + \frac{\eta}{(1-\gamma)^2} |\lambda_t - \hat{\lambda}_{t+1}| \right)$$
$$+ |\lambda - \hat{\lambda}_{t+1}| \left( |\hat{\lambda}_{t+1} - \hat{\lambda}_t| + \frac{\eta\kappa_\rho\sqrt{|A|}}{(1-\gamma)^3} \sum_s d_\rho^{\pi^\star}(s) \|\hat{\pi}_{t+1}(\cdot \mid s) - \pi_t(\cdot \mid s)\| \right)$$

$$\geq \max_s \|\pi(\cdot \mid s) - \hat{\pi}_{t+1}(\cdot \mid s)\| \sum_s d_\rho^\pi(s) \times$$
$$\left( \frac{1}{1-\gamma} \|\hat{\pi}_{t+1}(\cdot \mid s) - \hat{\pi}_t(\cdot \mid s)\| + \frac{\eta\gamma|A|}{(1-\gamma)^3} \max_s \|\hat{\pi}_{t+1}(\cdot \mid s) - \pi_t(\cdot \mid s)\| + \frac{\eta}{(1-\gamma)^2} |\lambda_t - \hat{\lambda}_{t+1}| \right)$$
$$+ |\lambda - \hat{\lambda}_{t+1}| \left( |\hat{\lambda}_{t+1} - \hat{\lambda}_t| + \frac{\eta\kappa_\rho\sqrt{|A|}}{(1-\gamma)^3} \sum_s d_\rho^{\pi^\star}(s) \|\hat{\pi}_{t+1}(\cdot \mid s) - \pi_t(\cdot \mid s)\| \right)$$

$$\geq \eta \left( V_{r+\hat{\lambda}_{t+1}g}^\pi(\rho) - V_{r+\lambda g}^{\hat{\pi}_{t+1}}(\rho) \right)$$

where the second inequality comes from the property of $\kappa_\rho$ and the last inequality is straightforward from (37). We take $\eta > 0$ such that $\max \left( \frac{2\eta|A|}{(1-\gamma)^3}, \frac{\eta\kappa_\rho}{(1-\gamma)^3} \right) \leq \frac{1}{2}$ and denote $\kappa_{\rho,\gamma} := \max \left( \frac{\kappa_\rho}{1-\gamma}, 1 \right)$. If we take the square of both sides of the inequality above, then the second product argument has the following upper bound,

$$\left( \sum_s d_\rho^{\pi^\star}(s) \left( \|\hat{\pi}_{t+1}(\cdot \mid s) - \hat{\pi}_t(\cdot \mid s)\| + \frac{2\eta|A|}{(1-\gamma)^3} \|\hat{\pi}_{t+1}(\cdot \mid s) - \pi_t(\cdot \mid s)\| \right) + |\hat{\lambda}_{t+1} - \hat{\lambda}_t| + \frac{\eta\kappa_\rho}{(1-\gamma)^3} |\lambda_t - \hat{\lambda}_{t+1}| \right)^2$$

$$\leq \left( \sum_s d_\rho^{\pi^\star}(s) \left( \|\hat{\pi}_{t+1}(\cdot \mid s) - \hat{\pi}_t(\cdot \mid s)\| + \frac{1}{2} \|\hat{\pi}_{t+1}(\cdot \mid s) - \pi_t(\cdot \mid s)\| \right) + |\hat{\lambda}_{t+1} - \hat{\lambda}_t| + \frac{1}{2}|\lambda_t - \hat{\lambda}_{t+1}| \right)^2$$

$$\leq \left( \sum_s d_\rho^{\pi^\star}(s) \left( \|\pi_t(\cdot \mid s) - \hat{\pi}_t(\cdot \mid s)\| + \frac{3}{2} \|\hat{\pi}_{t+1}(\cdot \mid s) - \pi_t(\cdot \mid s)\| \right) + |\lambda_t - \hat{\lambda}_t| + \frac{3}{2}|\lambda_t - \hat{\lambda}_{t+1}| \right)^2$$

$$\leq \left( \frac{3}{2} \sum_s d_\rho^{\pi^\star}(s) \left( \|\pi_t(\cdot \mid s) - \hat{\pi}_t(\cdot \mid s)\| + \|\hat{\pi}_{t+1}(\cdot \mid s) - \pi_t(\cdot \mid s)\| \right) + \frac{3}{2} \left( |\lambda_t - \hat{\lambda}_t| + |\lambda_t - \hat{\lambda}_{t+1}| \right) \right)^2$$

$$\leq 9 \sum_s d_\rho^{\pi^\star}(s) \left( \|\pi_t(\cdot \mid s) - \hat{\pi}_t(\cdot \mid s)\|^2 + \|\hat{\pi}_{t+1}(\cdot \mid s) - \pi_t(\cdot \mid s)\|^2 \right) + 9 \left( |\lambda_t - \hat{\lambda}_t|^2 + |\lambda_t - \hat{\lambda}_{t+1}|^2 \right)$$

where we use the inequality $(x+y)^2 \leq 2x^2 + 2y^2$ and Jensen's inequality. $\qquad \square$

Recall the definition of $\kappa_\rho$ and $\kappa_\rho \leq 1/\rho_{\min}$, where $\rho_{\min} := \min_s \rho(s)$.

**Lemma 21.** *Assume $\rho_{\min} > 0$. For any $t \geq 1$, the primal-dual iterates of OPG-PD* (9) *satisfy*

$$\sup_{\pi \in \Pi, \lambda \in \Lambda} \frac{V_{r+\hat{\lambda}_{t+1}g}^\pi(\rho) - V_{r+\lambda g}^{\hat{\pi}_{t+1}}(\rho)}{\max_s \|\pi(\cdot \mid s) - \hat{\pi}_{t+1}(\cdot \mid s)\| + |\lambda - \hat{\lambda}_{t+1}|}$$
$$\geq C_{\rho,\xi} \left( \sum_s d_\rho^{\pi^\star}(s) \|\hat{\pi}_{t+1}(\cdot \mid s) - \mathcal{P}_{\Pi^\star}(\hat{\pi}_{t+1}(\cdot \mid s))\| + |\hat{\lambda}_{t+1} - \mathcal{P}_{\Lambda^\star}(\hat{\lambda}_{t+1})| \right)$$

*where $C_{\rho,\xi} := c\rho_{\min}/(2\sqrt{|S||A|})/(1 + 1/((1-\gamma)\xi))$ in which $c > 0$ is described in Lemma 26.*

*Proof.* Denote $V^\star := V^{\pi^\star}_{r+\lambda^\star g}(\rho)$ and $D_{\max} := \max_{\pi,\pi' \in \Pi, \lambda, \lambda' \in \Lambda}(\max_s \|\pi(\cdot \,|\, s) - \pi'(\cdot \,|\, s)\| + |\lambda - \lambda'|)$. We observe that if we can prove that there exist constants $c_1, c_2 > 0$ such that

$$\max_{\pi \in \Pi} V^\pi_{r+\hat{\lambda}_{t+1}g}(\rho) - V^\star \geq c_1 |\hat{\lambda}_{t+1} - \mathcal{P}_{\Lambda^\star}(\hat{\lambda}_{t+1})| \tag{38a}$$

$$V^\star - \min_{\lambda \in \Lambda} V^{\hat{\pi}_{t+1}}_{r+\lambda g}(\rho) \geq c_2 \sum_s d^{\pi^\star}_\rho(s) \|\hat{\pi}_{t+1}(\cdot \,|\, s) - \mathcal{P}_{\Pi^\star}(\hat{\pi}_{t+1}(\cdot \,|\, s))\|, \tag{38b}$$

then,

$$\sup_{\pi \in \Pi, \lambda \in \Lambda} \frac{V^\pi_{r+\hat{\lambda}_{t+1}g}(\rho) - V^{\hat{\pi}_{t+1}}_{r+\lambda g}(\rho)}{\max_s \|\pi(\cdot \,|\, s) - \hat{\pi}_{t+1}(\cdot \,|\, s)\| + |\lambda - \hat{\lambda}_{t+1}|}$$

$$\geq \frac{1}{D_{\max}} \sup_{\pi \in \Pi, \lambda \in \Lambda} V^\pi_{r+\hat{\lambda}_{t+1}g}(\rho) - V^{\hat{\pi}_{t+1}}_{r+\lambda g}(\rho)$$

$$= \frac{1}{D_{\max}} \left( \max_{\pi \in \Pi} V^\pi_{r+\hat{\lambda}_{t+1}g}(\rho) - V^\star \right) + \frac{1}{D_{\max}} \left( V^\star - \min_{\lambda \in \Lambda} V^{\hat{\pi}_{t+1}}_{r+\lambda g}(\rho) \right)$$

$$\geq \frac{\min(c_1, c_2)}{D_{\max}} \left( \sum_s d^{\pi^\star}_\rho(s) \|\hat{\pi}_{t+1}(\cdot \,|\, s) - \mathcal{P}_{\Pi^\star}(\hat{\pi}_{t+1}(\cdot \,|\, s))\| + |\hat{\lambda}_{t+1} - \mathcal{P}_{\Lambda^\star}(\hat{\lambda}_{t+1})| \right)$$

which proves the lemma by taking $C_{\rho,\xi} := \frac{\min(c_1, c_2)}{D_{\max}}$.

We next prove (38) using the bilinear game result in Lemma 26. By the linear program formulation of constrained MDP, we can express the value function in terms of the occupancy measure $q^\pi$, e.g., $V^\pi_{r+\lambda g}(\rho) = \langle q^\pi, r + \lambda g \rangle$, where $q^\pi$ is the occupancy measure that lives in a polytope $\mathcal{Q}$ that is given by (15). Hence, the constrained saddle-point problem (3) reduces to,

$$\underset{q^\pi \in \mathcal{Q}}{\text{maximize}} \; \underset{\lambda \in \Lambda}{\text{minimize}} \; \langle q^\pi, r + \lambda g \rangle = \underset{\lambda \in \Lambda}{\text{minimize}} \; \underset{q^\pi \in \mathcal{Q}}{\text{maximize}} \; \langle q^\pi, r + \lambda g \rangle.$$

We notice that the game value keeps the same as $V^\star$ that is achieved at $(q^{\pi^\star}, \lambda^\star)$, where $q^{\pi^\star}$ is the occupancy measure under the policy $\pi^\star$. Let the set of occupancy measures associated with $\Pi^\star$ be $\mathcal{Q}^\star$. According to Lemma 26, we know that there exists a constant $c > 0$ such that

$$\max_{q^\pi \in \mathcal{Q}} \langle q^\pi, r + \lambda g \rangle - V^\star \geq c |\lambda - \mathcal{P}_{\Lambda^\star}(\lambda)| \tag{39a}$$

$$V^\star - \min_{\lambda \in \Lambda} \langle q^\pi, r + \lambda g \rangle \geq c \|q^\pi - \mathcal{P}_{\mathcal{Q}^\star}(q^\pi)\|. \tag{39b}$$

It is more straightforward to see (38a) from (39a) if we take $\lambda = \hat{\lambda}_{t+1}$ and $c_1 = c$. We next show (38b) using (39b) by taking $\pi^\star(\cdot \,|\, s)$ to be the policy associated with $\mathcal{P}_{\mathcal{Q}^\star}(q^\pi)$,

$$\sum_s d^{\pi^\star}_\rho(s) \|\pi(\cdot \,|\, s) - \pi^\star(\cdot \,|\, s)\|$$

$$= \sum_s d^{\pi^\star}_\rho(s) \left\| \frac{q^\pi(s, \cdot)}{q^\pi(s)} - \frac{q^{\pi^\star}(s, \cdot)}{q^{\pi^\star}(s)} \right\|$$

$$\leq \sum_s d^{\pi^\star}_\rho(s) \frac{\|q^\pi(s, \cdot) - q^{\pi^\star}(s, \cdot)\| q^{\pi^\star}(s)}{q^\pi(s) q^{\pi^\star}(s)} + \sum_s d^{\pi^\star}_\rho(s) \frac{\|q^{\pi^\star}(s, \cdot)\| |q^\pi(s) - q^{\pi^\star}(s)|}{q^\pi(s) q^{\pi^\star}(s)}$$

$$\leq \sum_s \frac{\|q^\pi(s, \cdot) - q^{\pi^\star}(s, \cdot)\|}{q^\pi(s)} + \sum_s \frac{|q^\pi(s) - q^{\pi^\star}(s)|}{q^\pi(s)}$$

$$\leq \frac{1}{\rho_{\min}} \sum_s \left( \|q^\pi(s, \cdot) - q^{\pi^\star}(s, \cdot)\| + |q^\pi(s) - q^{\pi^\star}(s)| \right)$$

$$\leq \frac{1}{\rho_{\min}} \left( \sqrt{|S|} \sqrt{\sum_s \|q^\pi(s, \cdot) - q^{\pi^\star}(s, \cdot)\|^2} + \sqrt{|S||A|} \sqrt{\sum_{s,a} |q^\pi(s, a) - q^{\pi^\star}(s, a)|^2} \right)$$

$$= \frac{2\sqrt{|S||A|}}{\rho_{\min}} \left\| q^\pi - q^{\pi^\star} \right\|$$

$$= \frac{2\sqrt{|S||A|}}{\rho_{\min}} \|q^\pi - \mathcal{P}_{\mathcal{Q}^\star}(q^\pi)\|$$

where the first inequality is due to triangle inequality, we use the fact: $(1-\gamma)q^{\pi^*}(s) = d_\rho^{\pi^*}(s)$, $d_\rho^{\pi^*}(s) \leq 1$, and $\left\| q^{\pi^*}(s,\cdot) \right\| \leq 1$ in the second inequality, the third inequality is due to that $q^\pi(s) \geq \rho_{\min}$, and we apply Cauchy-Schwarz inequality in the last inequality. Hence, we can further lower bound (39b),

$$
\begin{aligned}
V^\star - \min_{\lambda \in \Lambda} \langle q^\pi, r + \lambda g \rangle \;&\geq\; c \left\| q^\pi - \mathcal{P}_{\mathcal{Q}^\star}(q^\pi) \right\| \\
&\geq\; \frac{c\rho_{\min}}{2\sqrt{|S||A|}} \sum_s d_\rho^{\pi^\star}(s) \left\| \pi(\cdot \mid s) - \pi^\star(\cdot \mid s) \right\| \\
&\geq\; \frac{c\rho_{\min}}{2\sqrt{|S||A|}} \sum_s d_\rho^{\pi^\star}(s) \left\| \pi(\cdot \mid s) - \mathcal{P}_{\Pi^\star}(\pi(\cdot \mid s)) \right\|
\end{aligned}
$$

which yields (38b) using $c_2 = \frac{c\rho_{\min}}{2\sqrt{|S||A|}}$.

Finally, we combine all selected constants and take $D_{\max} = 1 + \frac{1}{(1-\gamma)\xi}$ to conclude the proof. $\qquad\square$

**Lemma 22.** *For any policies $\pi$ and $\pi'$, we have*

$$
\left\| Q_r^\pi(\cdot,\cdot) - Q_r^{\pi'}(\cdot,\cdot) \right\|_\infty \;\leq\; \frac{\gamma}{(1-\gamma)^2} \max_s \left\| \pi(\cdot \mid s) - \pi'(\cdot \mid s) \right\|_1
$$

$$
\left| V_g^\pi(\rho) - V_g^{\pi'}(\rho) \right| \;\leq\; \frac{\kappa_\rho}{(1-\gamma)^3} \sum_s d_\rho^{\pi^\star}(s) \left\| \pi(\cdot \mid s) - \pi'(\cdot \mid s) \right\|_1 .
$$

*Proof.* By the Bellman equations, for each pair $(s,a)$,

$$
\begin{aligned}
Q_r^\pi(s,a) \;&=\; r(s,a) + \gamma \sum_{s',a'} P(s' \mid s, a)\pi(a' \mid s')Q_r^\pi(s',a') \\
Q_r^{\pi'}(s,a) \;&=\; r(s,a) + \gamma \sum_{s',a'} P(s' \mid s, a)\pi'(a' \mid s')Q_r^{\pi'}(s',a').
\end{aligned}
$$

Hence, for each pair $(s,a)$,

$$
\begin{aligned}
\left| Q_r^\pi(s,a) - Q_r^{\pi'}(s,a) \right| \;&\leq\; \gamma \sum_{s',a'} P(s' \mid s, a) \left| \pi(a' \mid s')Q_r^\pi(s',a') - \pi'(a' \mid s')Q_r^{\pi'}(s',a') \right| \\
&\leq\; \gamma \sum_{s',a'} P(s' \mid s, a) \left| \pi(a' \mid s') - \pi'(a' \mid s') \right| Q_r^{\pi'}(s',a') \\
&\quad + \gamma \sum_{s',a'} P(s' \mid s, a)\pi'(a' \mid s') \left| Q_r^\pi(s',a') - Q_r^{\pi'}(s',a') \right| \\
&\leq\; \frac{\gamma}{1-\gamma} \max_s \left\| \pi(\cdot \mid s) - \pi'(\cdot \mid s) \right\|_1 + \gamma \left\| Q_r^\pi(\cdot,\cdot) - Q_r^{\pi'}(\cdot,\cdot) \right\|_\infty
\end{aligned}
$$

which yields the first inequality.

To show the second inequality, we employ the performance difference lemma to show that,

$$
\begin{aligned}
\left| V_g^\pi(\rho) - V_g^{\pi'}(\rho) \right| \;&\leq\; \frac{1}{1-\gamma} \sum_{s,a} d_\rho^\pi(s)|\pi(a \mid s) - \pi'(a \mid s)||Q_g^{\pi'}(s,a)| \\
&\leq\; \frac{1}{(1-\gamma)^2} \sum_s d_\rho^\pi(s) \left\| \pi(\cdot \mid s) - \pi'(\cdot \mid s) \right\|_1 \\
&\leq\; \frac{1}{(1-\gamma)^2} \sum_s d_\rho^\pi(s) \left\| \pi(\cdot \mid s) - \pi'(\cdot \mid s) \right\|_1
\end{aligned}
$$

where we replace $d_\rho^\pi$ by $d_\rho^{\pi^\star}$ using the inequality $\left\| d_\rho^\pi / d_\rho^{\pi^\star} \right\|_\infty \leq \kappa_\rho/(1-\gamma)$ for any policy $\pi \in \Pi$. $\qquad\square$

## D.2 Proof of Theorem 6

*Proof.* From the non-increasing relation (29) in Lemma 19, we have

$$\frac{1}{2(1-\gamma)}\sum_s d_\rho^{\pi^\star}(s)\,\|\hat\pi_{t+1}(\cdot|s)-\pi_t(\cdot|s)\|^2 \;+\; \frac{1}{2}(\hat\lambda_{t+1}-\lambda_t)^2 \;\le\; \zeta_t \;\le\; \frac{16}{7}\Theta_t \;\le\; \frac{16}{7}\Theta_1.$$

Meanwhile, from Lemma 20, we obtain that

$$
\begin{aligned}
\zeta_t \;\ge\;& \frac{1}{4}\sum_s d_\rho^{\pi^\star}(s)\,\|\hat\pi_{t+1}(\cdot\,|\,s)-\pi_t(\cdot\,|\,s)\|^2 \;+\; \frac{1}{4}(\hat\lambda_{t+1}-\lambda_t)^2 \\
&+\frac{1}{4}\sum_s d_\rho^{\pi^\star}(s)\left(\|\hat\pi_{t+1}(\cdot\,|\,s)-\pi_t(\cdot\,|\,s)\|^2 + \|\pi_t(\cdot\,|\,s)-\hat\pi_t(\cdot\,|\,s)\|^2\right) \\
&+\frac{1}{4}\left((\hat\lambda_{t+1}-\lambda_t)^2 + (\lambda_t-\hat\lambda_t)^2\right) \\
\;\ge\;& \frac{1}{4}\sum_s d_\rho^{\pi^\star}(s)\,\|\hat\pi_{t+1}(\cdot\,|\,s)-\pi_t(\cdot\,|\,s)\|^2 \;+\; \frac{1}{4}(\hat\lambda_{t+1}-\lambda_t)^2 \\
&+\frac{\eta^2}{36\kappa_{\rho,\gamma}^2}\sup_{\pi\in\Pi,\lambda\in\Lambda}\frac{\left[V_{r+\hat\lambda_{t+1}g}^{\pi}(\rho)-V_{r+\lambda g}^{\hat\pi_{t+1}}(\rho)\right]_+^2}{\left(\max_s\|\pi(\cdot\,|\,s)-\hat\pi_{t+1}(\cdot\,|\,s)\|+|\lambda-\hat\lambda_{t+1}|\right)^2} \\
\;\ge\;& \frac{1}{4}\sum_s d_\rho^{\pi^\star}(s)\,\|\hat\pi_{t+1}(\cdot\,|\,s)-\pi_t(\cdot\,|\,s)\|^2 \;+\; \frac{1}{4}(\hat\lambda_{t+1}-\lambda_t)^2 \\
&+\frac{\eta^2 C_{\rho,\xi}'}{36\kappa_{\rho,\gamma}^2}\left(\sum_s d_\rho^{\pi^\star}(s)\,\|\hat\pi_{t+1}(\cdot\,|\,s)-\mathcal{P}_{\Pi^\star}(\hat\pi_{t+1}(\cdot\,|\,s))\|^2 + (\hat\lambda_{t+1}-\mathcal{P}_{\Lambda^\star}(\hat\lambda_{t+1}))^2\right) \\
\;\ge\;& (1-\gamma)\min\left(2,\frac{4\eta^2 C_{\rho,\xi}'}{9\kappa_{\rho,\gamma}^2}\right)\Theta_{t+1}
\end{aligned}
$$

where the third inequality is due to Lemma 21, the inequalities: $(x+y)^2\ge x^2+y^2$ for $x,\,y\ge 0$, and $d_\rho^{\pi^\star}(s)\ge(1-\gamma)\rho_{\min}$, and $C_{\rho,\xi}':=(1-\gamma)C_{\rho,\xi}^2\rho_{\min}$. Hence, the relation (29) reduces into,

$$\Theta_{t+1}\;\le\;\Theta_t\;-\;(1-\gamma)\min\left(\frac{7}{8},\frac{7\eta^2 C_{\rho,\xi}'}{36\kappa_{\rho,\gamma}^2}\right)\Theta_{t+1}$$

which implies that $\Theta_t$ is decreasing to zero at an exponential rate. $\qquad\square$

## D.3 Proof of Corollary 7

*Proof.* According to Theorem 6, if we take the same stepsize $\eta$, then for any $t=\Omega(\log\frac{1}{\epsilon})$,

$$\frac{1}{2(1-\gamma)}\sum_s d_\rho^{\pi^\star}(s)\,\|\mathcal{P}_{\Pi^\star}(\hat\pi_t(\cdot\,|\,s))-\hat\pi_t(\cdot\,|\,s)\|^2 \;=\; O(\epsilon) \quad\text{and}\quad \frac{1}{2}(\mathcal{P}_{\Lambda^\star}(\hat\lambda_t)-\hat\lambda_t)^2 \;=\; O(\epsilon).$$

Let $\hat\pi_t^\star(\cdot\,|\,s):=\mathcal{P}_{\Pi^\star}(\hat\pi_t(\cdot\,|\,s))$ and $\hat\lambda_t^\star:=\mathcal{P}_{\Lambda^\star}(\hat\lambda_t)$. Because of the interchangeability of saddle points from Lemma 9, $(\hat\pi_t^\star,\hat\lambda_t^\star)$ is a saddle point in the set $\Pi^\star\times\Lambda^\star$ for any $t\ge 0$.

First, we have

$$
\begin{aligned}
V_r^{\hat\pi_t^\star}(\rho)-V_r^{\hat\pi_t}(\rho) \;=\;& \frac{1}{1-\gamma}\sum_{s,a} d_\rho^{\hat\pi_t^\star}(s)\,(\hat\pi_t^\star(a\,|\,s)-\hat\pi_t(a\,|\,s))\,Q_r^{\pi_t}(s,a) \\
\;\le\;& \frac{1}{(1-\gamma)^2}\sum_s d_\rho^{\hat\pi_t^\star}(s)\,\|\hat\pi_t^\star(\cdot\,|\,s)-\hat\pi_t(\cdot\,|\,s)\|_1 \\
\;\le\;& \frac{\sqrt{|A|}}{(1-\gamma)^2}\sum_s d_\rho^{\hat\pi_t^\star}(s)\,\|\hat\pi_t^\star(\cdot\,|\,s)-\hat\pi_t(\cdot\,|\,s)\| \\
\;\le\;& \frac{\sqrt{|A|}}{(1-\gamma)^2}\sqrt{\sum_s d_\rho^{\hat\pi_t^\star}(s)\,\|\hat\pi_t^\star(\cdot\,|\,s)-\hat\pi_t(\cdot\,|\,s)\|^2}
\end{aligned}
$$

where we use Cauchy–Schwarz inequality in the first and third inequalities, and the second inequality is due to $\|x\|_1 \leq \sqrt{d}\,\|x\|_2$ for $x \in \mathbb{R}^d$, which shows $V_r^{\hat{\pi}_t^\star}(\rho) - V_r^{\hat{\pi}_t}(\rho) \leq O(\sqrt{\epsilon})$. By the saddle-point property of $(\hat{\pi}_t^\star, \hat{\lambda}_t^\star)$, $\hat{\pi}_t^\star$ is also an optimal constrained policy, i.e., $V_r^{\hat{\pi}_t^\star}(\rho) = V_r^{\pi^\star}(\rho)$. Therefore, $V_r^{\pi^\star}(\rho) - V_r^{\hat{\pi}_t}(\rho) \leq O(\sqrt{\epsilon})$.

Second, we have

$$-V_g^{\hat{\pi}_t}(\rho) \;=\; \underbrace{-V_g^{\hat{\pi}_t^\star}(\rho)}_{(i)} + \underbrace{V_g^{\hat{\pi}_t^\star}(\rho) - V_g^{\hat{\pi}_t}(\rho)}_{(ii)}.$$

Similar to bounding $V_r^{\hat{\pi}_t^\star}(\rho) - V_r^{\hat{\pi}_t}(\rho)$, we can show that $(ii) \leq O(\sqrt{\epsilon})$. By the saddle-point property of $(\hat{\pi}_t^\star, \hat{\lambda}_t^\star)$, $V_g^{\hat{\pi}_t^\star}(\rho) \geq 0$. Therefore, $-V_g^{\hat{\pi}_t}(\rho) \leq O(\sqrt{\epsilon})$.

Finally, we replace the accuracy $\sqrt{\epsilon}$ by $\epsilon$ and combine all big O notation to conclude the proof. $\qquad\square$

### D.4 Zero constraint violation of OPG-PD (9)

**Corollary 23** (Zero constraint violation). *Let Assumption 1 hold and the optimal state visitation distribution be unique, i.e., $d_\rho^\pi = d_\rho^{\pi^\star}$ for any $\pi \in \Pi^\star$, and $\rho_{\min} > 0$. For small $\epsilon$, there exists $\delta > 0$ such that if we instead use the conservative constraint $V_{g'}^\pi(\rho) \geq 0$ for $g' = g - (1-\gamma)\delta$, and take the stepsize $\eta$ from Theorem 6, then the policy iterates of OPG-PD satisfy*

$$V_r^{\pi^\star}(\rho) - V_r^{\hat{\pi}_t}(\rho) \;\leq\; \epsilon \;\;\text{and}\;\; -V_g^{\hat{\pi}_t}(\rho) \;\leq\; 0 \;\;\text{for any } t = \Omega\left(\log^2 \frac{1}{\epsilon}\right)$$

*where $\Omega(\cdot)$ only omits some problem-dependent constant.*

*Proof.* We apply the conservatism to the translated constraint $V_g^\pi(\rho) \geq 0$ in Problem (1). Specifically, for any $\delta < \min(\xi, 1)$, we let $g' := g - (1-\gamma)\delta$ and define a conservative constraint,

$$V_{g'}^\pi \;:=\; V_g^\pi(\rho) - \delta \;\geq\; 0.$$

It is straightforward to see that Assumption 1 ensures that $V_{g'}^{\pi_t}(\rho) \geq 0$ is strictly feasible for a new slack variable $\xi' := \xi - \delta$. We now can apply OPG-PD (9) to a new Lagrangian,

$$L'(\pi, \lambda) \;:=\; V_{r+\lambda g'}^\pi(\rho)$$

and Corollary 7 holds if we replace $g$ in OPG-PD by $g'$. Thus,

$$V_r^{\pi_\delta^\star}(\rho) - V_r^{\hat{\pi}_t}(\rho) \;\leq\; O(\epsilon) \;\;\text{and}\;\; -V_{g'}^{\hat{\pi}_t}(\rho) \;\leq\; O(\epsilon) \;\;\text{for any } t = \Omega\left(\log^2 \frac{1}{\epsilon}\right)$$

where $\Omega(\cdot)$ hides some problem-dependent constant, and $\pi_\delta^\star$ is an optimal policy to the $\delta$-perturbed constrained policy optimization problem,

$$\underset{\pi \in \Pi}{\text{maximize}} \; V_r^\pi(\rho) \qquad \text{subject to } V_g^\pi(\rho) - \delta \;\geq\; 0. \tag{40}$$

We notice that the above $\Omega(\cdot)$ has some problem-dependent constant $\Upsilon > 0$. Thus, we select $\delta$ such that $\delta \geq \epsilon\Upsilon$, which is always possible for small enough $\epsilon$, for instance $\delta = \frac{\xi}{2}$ and $\xi' = \frac{\xi}{2}$. Hence, if we take $\delta = \frac{\xi}{2}$ and such small $\epsilon$, then,

$$-V_{g'}^{\hat{\pi}_t}(\rho) \;=\; -V_g^{\hat{\pi}_t}(\rho) + \delta \;\leq\; \epsilon\Upsilon \;\;\text{for any } t = \Omega\left(\log^2 \frac{1}{\epsilon}\right)$$

which shows that $V_g^{\hat{\pi}_t}(\rho) \geq 0$ for some large $t$.

The rest is to show that $V_r^{\pi^\star}(\rho) - V_r^{\hat{\pi}_t}(\rho) \leq O(\epsilon)$. We notice that $\pi^\star$ is an optimal policy to Problem (40) when $\delta = 0$. Let $q^\star$ and $q_\delta^\star$ be associated occupancy measures of policies $\pi^\star$ and $\pi_\delta^\star$. In the occupancy measure space, Problem (40) becomes a linear program and it has a solution $q_\delta^\star$. Thus, we can view $q_\delta^\star$ as a $\delta$-perturbed solution of a convex optimization problem in which all functions are continuous differentiable and the domain is convex and compact. It is known from [129, Theorem 3.1] that the optimal solution $q_\delta^\star$ is continuous in $\delta$, which implies that for any

$\epsilon > 0$, there exists $\delta'$ such that $|\langle r, q^\star \rangle - \langle r, q_\delta^\star \rangle| \leq O(\epsilon)$ for any $\delta < \delta'$. Thus, $|V_r^{\pi^\star}(\rho) - V_r^{\pi_\delta^\star}(\rho)| \leq O(\epsilon)$ for small enough $\epsilon$. Therefore,

$$V_r^{\pi^\star}(\rho) - V_r^{\hat{\pi}_t}(\rho) \leq V_r^{\pi_\delta^\star}(\rho) - V_r^{\hat{\pi}_t}(\rho) + |V_r^{\pi^\star}(\rho) - V_r^{\pi_\delta^\star}(\rho)| \leq O(\epsilon)$$

for some large $t$. Collecting all conditions on $\delta$ leads to our final choice of $\delta = \min(\frac{\xi}{2}, 1, \delta')$. Finally, we combine all big O notation to complete the proof. □

## E    Other Computational Experiments

In this section, we report details of our experimental setup and additional experimental results that verify merits and effectiveness of our methods: RPG-PD (6) and OPG-PD (9). We implement RPG-PD in the form of NPG [49] and the restricted probability simplex $\hat{\Delta}(A)$ by restraining policy parameter to be bounded.

To properly assess the convergence performance, our experiment is a tabular constrained MDP with a randomly generated transition kernel, a discount factor $\gamma = 0.9$, uniform rewards $r \in [0, 1]$ and utilities $g \in [-1, 1]$, and a uniform initial state distribution $\rho$. The constraint is $V_g^\pi(\rho) \geq 0$. To check feasibility, we employ the standard policy iteration procedure to solve a standard MDP problem with respect to $V_g^\pi(\rho)$. If feasible, we then solve a linear program in occupancy-measure space to find the optimal reward value $V_r^{\pi^\star}(\rho)$ at an optimal policy $\pi^\star$ induced by the optimal occupancy measure. Throughout all experiments, the random seed is fixed and $V_r^{\pi^\star}(\rho)$ takes the value 8.16 at an optimal policy $\pi^\star$.

We compare our methods RPG-PD and OPG-PD with three typical learning algorithms in the constrained MDP literature: (i) primal-dual methods [23, 29]; (ii) dual methods [28, 24, 34]; and (iii) primal method [77]. We have reported our comparison for primal-dual methods in Section 5 and Figure 1. We now report our comparison experiments on other two baseline methods: (ii) dual methods [28, 24, 34] and (iii) primal method [77], in Section E.1. In addition, we showcase the last-iterate zero constraint violation performance of our methods in Section E.2, and conduct sensitivity analysis of our methods to regularization and stepsize in Section E.3 and Section E.4, together with a variant of OPG-PD and policy-based ReLOAD [40]. All the experiments were conducted on an Apple MacBook Pro 2017 laptop equipped with a 2.3 GHz Dual-Core Intel Core i5 in Jupyter Notebook.

### E.1    Last-iterate convergence comparison with other baselines

We report our comparison for dual methods in Figure 4. We notice that dual methods [28, 24, 34] work in a double-loop fashion, where a (regularized) NPG subroutine is executed to perform the dual update. To make a fair comparison, the number of dual updates and the number of NPG steps are set to be 53 and 20 to ensure that dual methods take the same number of policy gradient updates: 1060, as RPG-PD and OPG-PD, and we evaluate all policy iterates inside the NPG subroutines of dual methods.

In Figure 4, RPG-PD (– –) and OPG-PD (—) outperform PMD-PD [24] (– –), AR-CPO [28] (–·–), and Accelerated Dual [34] (····) in several aspects. First, we see that the known oscillation behavior in primal-dual methods also shows up in these dual methods, which could result from that we can't exactly evaluate the search direction of the dual update via a NPG subroutine. Thus, dual methods can be viewed an instantiation of two-time-scale methods in which the primal update performs faster than the dual update. Regarding this, RPG-PD and OPG-PD show outstanding performance in suppressing oscillation behavior. Second, since RPG-PD and OPG-PD are single-time-scale primal-dual methods, it is easy to tune algorithmic parameters compared with double-loop dual methods. For instance, in Accelerated Dual, it is relatively difficult to find a set of algorithmic parameters that avoid the sub-optimal performance in Figure 4. Third, OPG-PD reaches the maximum reward value 8.16 and RPG-PD converges to a slightly smaller reward value because of regularization, while both methods enjoy the utility constraint satisfaction in the last-iterate fashion, which seems to be difficult for dual methods because of oscillating utility values. Hence, using the same number of policy gradient updates, we have verified that OPG-PD and RPG-PD also can outperform several dual methods by yielding an optimal constrained policy in the last-iterate fashion.

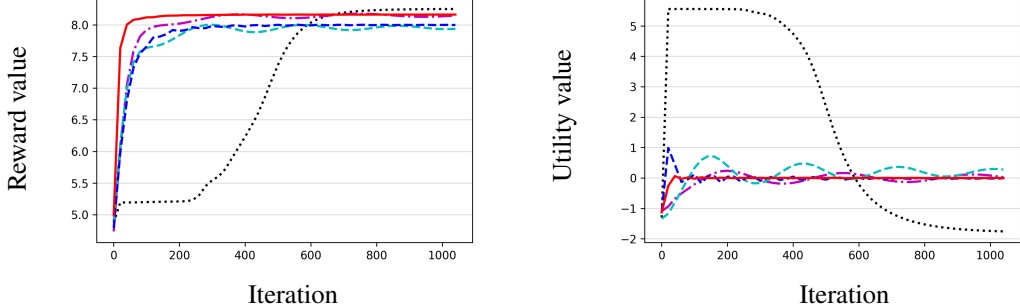

Figure 4: Convergence performance of RPG-PD, OPG-PD, and dual methods. Learning curves of our RPG-PD (− −) and OPG-PD (—), and PMD-PD [24] (− −), AR-CPO [28] (−·−), and Accelerated Dual [34] (····) methods. The horizontal axes represent the policy iterations $\{\pi_t\}_{t \geq 0}$ that are generated by each method and the vertical axes mean the value functions of the policy iterates $\{\pi_t\}_{t \geq 0}$: reward value $V_r^{\pi_t}(\rho)$ (Left) and utility value $V_g^{\pi_t}(\rho)$ (Right). In this experiment, for RPG-PD and OPG-PD, we use the stepsize $\eta = 0.1$ and the regularization parameter $\tau = 0.08$ for RPG-PD, and the initial distribution $\rho$ is uniform. For PMD-PD, AR-CPO, and Accelerated Dual, we use the stepsize $\eta = 0.1$ for the dual update, the regularized NPG stepsize $\alpha = 1$, and the regularization parameter $\tau = 0.08$, and the uniform initial distribution $\rho$.

We report our comparison for a primal method in Figure 5. In Figure 5, RPG-PD (− −) and OPG-PD (—) outperform CRPO [77] (−·−). We notice that although CRPO [77] works in a single-time-scale fashion as RPG-PD and OPG-PD, the policy has to be updated by alternatively using the gradient directions of reward/utility value functions. To ensure constraint satisfaction, CRPO often switches between the gradient directions of reward/utility value functions depending on the amount of constraint violation. As a result, we see that CRPO reaches a slightly lower reward value than OPG-PD's, and the constraint satisfaction is relatively conservative and has mild oscillation behavior. In contrast, OPG-PD achieves the maximum reward value 8.16 and RPG-PD converges to a slightly smaller reward value because of regularization, while both methods enjoy the utility constraint satisfaction in the last-iterate fashion. Last but not least, we have supported RPG-PD and OPG-PD with a policy last-iterate convergence theory, while such theory is unknown for CRPO as far as we know.

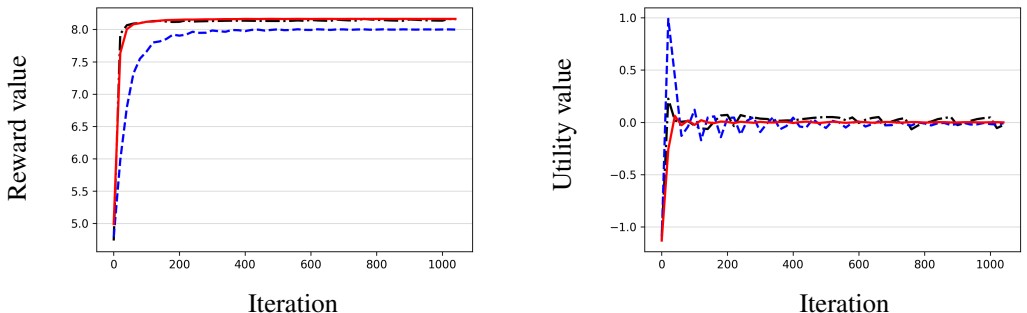

Figure 5: Convergence performance of RPG-PD, OPG-PD, and primal methods. Learning curves of our RPG-PD (− −) and OPG-PD (—), and CRPO [77] (−·−) methods. The horizontal axes represent the policy iterations $\{\pi_t\}_{t \geq 0}$ that are generated by each method and the vertical axes mean the value functions of the policy iterates $\{\pi_t\}_{t \geq 0}$: reward value $V_r^{\pi_t}(\rho)$ (Left) and utility value $V_g^{\pi_t}(\rho)$ (Right). In this experiment, for RPG-PD and OPG-PD, we use the stepsize $\eta = 0.1$ and the regularization parameter $\tau = 0.08$ for RPG-PD, and the initial distribution $\rho$ is uniform. For CRPO, we update the policy via the NPG step with stepsize $\eta = 0.1$ and the uniform initial distribution $\rho$.

## E.2 Last-iterate zero constraint violation comparison

In this experiment, we continue our previous tabular constrained MDP with a random transition, a discount factor $\gamma = 0.9$, uniform rewards $r \in [0,1]$ and utilities $g \in [-1,1]$, and an uniform initial state distribution $\rho$. Instead of the nominal constraint $V_g^\pi(\rho) \geq 0$, we use a conservative constraint $V_{g'}^\pi(\rho) \geq 0$ in RPG-PD and OPG-PD, where $g' := g - (1-\gamma)\delta$ is a conservative utility function and $\delta$ is the conservative parameter. To get zero constraint violation regarding the nominal constraint, we apply RPG-PD and OPG-PD to the conservative constraint $V_g^\pi(\rho) \geq \delta$ where we take the conservative parameter $\delta = 0.1$. As above, we compare conservative RPG-PD and OPG-PD with three typical learning algorithms in the constrained MDP literature: (i) primal-dual methods [23, 29]; (ii) dual methods [28, 24, 34]; and (iii) primal approach [77]. We report our comparison for primal-dual methods in Figure 6, for dual methods [28, 24, 34] in Figure 7, and for primal approach [77] in Figure 8. We observe that conservative RPG-PD and OPG-PD achieve similar performance regarding the oscillation suppression and the optimality of reward values as shown in Figure 1, Figure 4, and Figure 5. Interestingly, in Figure 6, Figure 7, and Figure 8, RPG-PD and OPG-PD converge to a utility value that is strictly above zero, i.e., $V_g^{\pi_t}(\rho) > 0$ for large $t$, which is not guaranteed in many of other methods. To sum up, we have confirmed that RPG-PD and OPG-PD can ensure zero constraint violation of instantaneous policy iterates in a finite number of training time.

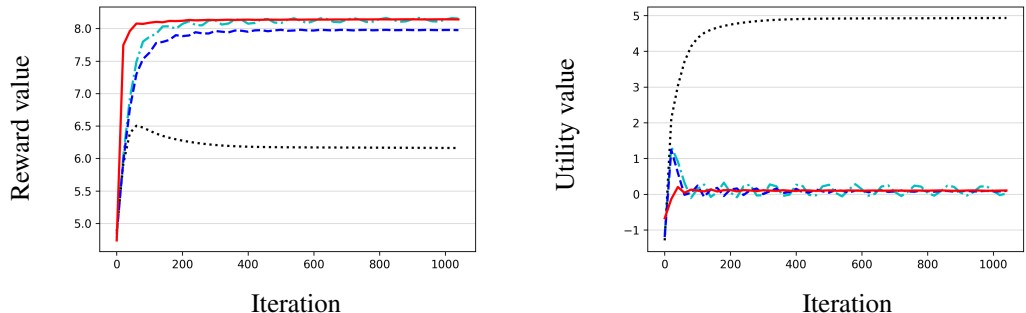

Figure 6: Convergence performance of RPG-PD, OPG-PD, and primal-dual methods. Learning curves of our RPG-PD (– –) and OPG-PD (——), and NPG-PD [23] (–·–) and PID Lagrangian [29] (····) methods. The horizontal axes represent the policy iterations $\{\pi_t\}_{t \geq 0}$ that are generated by each method and the vertical axes mean the value functions of the policy iterates $\{\pi_t\}_{t \geq 0}$: reward value $V_r^{\pi_t}(\rho)$ (Left) and utility value $V_g^{\pi_t}(\rho)$ (Right). In this experiment, we apply RPG-PD and OPG-PD to a conservative constraint $V_g^\pi(\rho) \geq \delta$, and we take the conservative parameter $\delta = 0.1$, the same stepsize $\eta = 0.1$ for all methods, the regularization parameter $\tau = 0.08$ for RPG-PD, and the uniform initial distribution $\rho$.

## E.3 Sensitivity of RPG-PD (6) to regularization and stepsize

In this experiment, we use our previous tabular constrained MDP with a random transition, a discount factor $\gamma = 0.9$, uniform rewards $r \in [0,1]$ and utilities $g \in [-1,1]$, and an uniform initial state distribution $\rho$. The constraint is $V_g^\pi(\rho) \geq 0$. We recall that when we set the regularization parameter $\tau = 0$, RPG-PD becomes a policy-based primal-dual method [23] that often suffers the oscillation issue as shown in Figure 1. However, larger regularization usually bias regularized methods more to sub-optimal solutions. Hence, it is important to reveal how the last-iterate convergence of RPG-PD depends on the regularization parameter $\tau$ and the stepsize $\eta$.

We first repeat executing RPG-PD with a fixed stepsize $\eta = 0.1$, but varying three different regularization parameters $\tau \in \{0.1, 0.05, 0.01\}$. In Figure 9, RPG-PD damps initial oscillations successfully for $\tau = 0.1$ and $0.05$, and oscillates for $\tau = 0.01$. When we decrease $\tau$ from $0.1$ to $0.05$, the reward value RPG-PD converges to becomes higher, and the utility value's oscillation is slightly amplified initially, but damped eventually. When $\tau$ is further reduced to $0.01$, although the reward value reaches a value around the optimal reward value $8.16$, the utility value behaves oscillating.

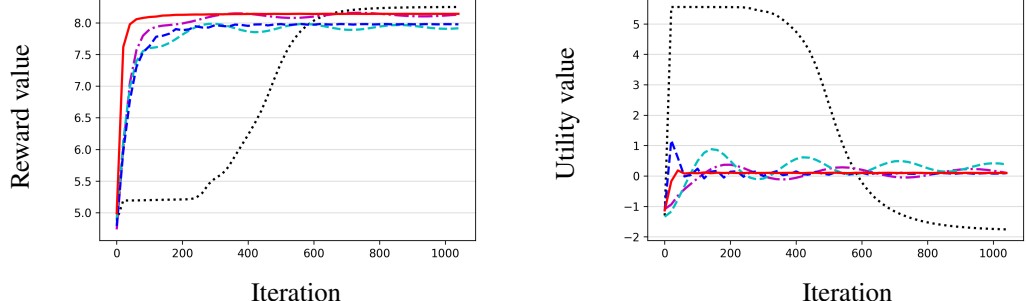

Figure 7: Convergence performance of RPG-PD, OPG-PD, and dual methods. Learning curves of our RPG-PD (--) and OPG-PD (—), and PMD-PD [24] (--), AR-CPO [28] (-·-), and Accelerated Dual [34] (····) methods. The horizontal axes represent the policy iterations $\{\pi_t\}_{t \geq 0}$ that are generated by each method and the vertical axes mean the value functions of the policy iterates $\{\pi_t\}_{t \geq 0}$: reward value $V_r^{\pi_t}(\rho)$ (Left) and utility value $V_g^{\pi_t}(\rho)$ (Right). In this experiment, we apply RPG-PD and OPG-PD to a conservative constraint $V_g^\pi(\rho) \geq \delta$, and we take the stepsize $\eta = 0.1$, the conservative parameter $\delta = 0.1$, the regularization parameter $\tau = 0.08$ for RPG-PD, and the initial distribution $\rho$ is uniform. For PMD-PD, AR-CPO, and Accelerated Dual, we use the stepsize $\eta = 0.1$ for the dual update, the regularized NPG stepsize $\alpha = 1$, and the regularization parameter $\tau = 0.08$, and the uniform initial distribution $\rho$.

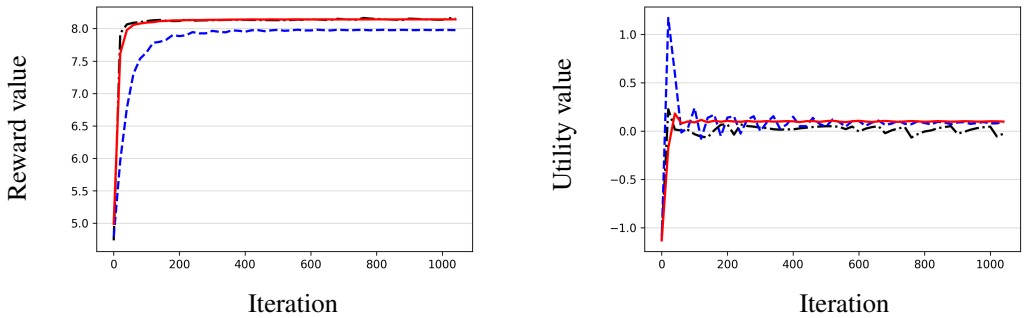

Figure 8: Convergence performance of RPG-PD, OPG-PD, and primal methods. Learning curves of our RPG-PD (--) and OPG-PD (—), and CRPO [77] (-··) methods. The horizontal axes represent the policy iterations $\{\pi_t\}_{t \geq 0}$ that are generated by each method and the vertical axes mean the value functions of the policy iterates $\{\pi_t\}_{t \geq 0}$: reward value $V_r^{\pi_t}(\rho)$ (Left) and utility value $V_g^{\pi_t}(\rho)$ (Right). In this experiment, we apply RPG-PD and OPG-PD to a conservative constraint $V_g^\pi(\rho) \geq \delta$, and we take the conservative parameter $\delta = 0.1$, the stepsize $\eta = 0.1$, the regularization parameter $\tau = 0.08$ for RPG-PD, and the initial distribution $\rho$ is uniform. For CRPO, we update the policy via the NPG step with stepsize $\eta = 0.1$ and the uniform initial distribution $\rho$.

A reason for this is that RPG-PD with a relatively small regularization parameter (compared to the stepsize) works as usual un-regularized single-time-scale primal-dual methods. Hence, increasing the regularization parameter, $\tau = 0.1$ can accelerate the convergence and attenuate the oscillation more effectively, although it makes the reward value more sub-optimal, which is also clearly shown in Figure 10.

To demonstrate the convergence of RPG-PD's policy iterates, we measure the policy optimality gap via the squared norm distance of policy iterates to an optimal policy $\pi^\star$ that is obtained from an occupancy-measure-based linear program. From the optimality gap in Figure 10, we see that three policy optimality gaps decrease sublinearly to some constants in the logarithmic scale plot, which verifies the sublinear last-iterate convergence of RPG-PD's policy iterates in Theorem 2. Hence, we

conjecture that it is impossible to improve the order of RPG-PD's sublinear rate, without introducing new algorithmic design.

To reduce the oscillation behavior, we repeat this experiment with a smaller stepsize $\eta = 0.01$ as suggested by Corollary 3. We report our result in Figure 11 and Figure 12. The oscillation in the utility value previously happened for $\tau = 0.01$ becomes less frequent, and the best reward value is achieved in this case. A noticeable loss is that the convergence has been slowed down, apparently in Figure 12. Hence, we have shown that balancing the stepsize and the regularization parameter (e.g., Corollary 3) is important for RPG-PD to achieve better oscillation attenuation and last-iterate convergence in practice.

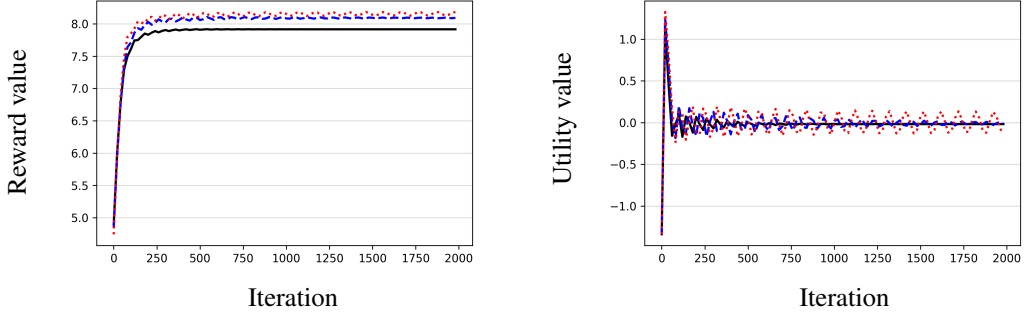

Figure 9: Convergence performance of RPG-PD with regularization parameter $\tau$: ($\tau = 0.1$, —), ($\tau = 0.05$, − −), ($\tau = 0.01$, ····). The horizontal axes represent the policy iterations $\{\pi_t\}_{t \geq 0}$ that are generated by RPG-PD and the vertical axes mean the value functions of the policy iterates $\{\pi_t\}_{t \geq 0}$: reward value $V_r^{\pi_t}(\rho)$ (Left) and utility value $V_g^{\pi_t}(\rho)$ (Right). In this experiment, we fix the same stepsize $\eta = 0.1$ and take the regularization parameter $\tau$ among $0.1$, $0.05$, and $0.01$, and the uniform initial distribution $\rho$.

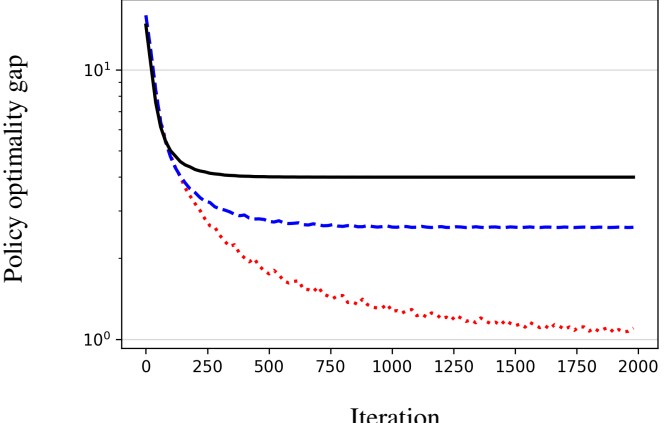

Figure 10: Convergence performance of RPG-PD with regularization parameter $\tau$: ($\tau = 0.1$, —), ($\tau = 0.05$, − −), ($\tau = 0.01$, ····). The horizontal axis represents the policy iterations $\{\pi_t\}_{t \geq 0}$ that are generated by RPG-PD and the vertical axis means the policy optimality gap that measures the distance of the policy iterates $\{\pi_t\}_{t \geq 0}$ to an optimal policy $\pi^\star$: $\sum_s \|\pi_t(\cdot \,|\, s) - \pi^\star(\cdot \,|\, s)\|^2$. In this experiment, we fix the stepsize $\eta = 0.1$ and take the regularization parameter among $0.1$, $0.05$, and $0.01$, and the uniform initial distribution $\rho$.

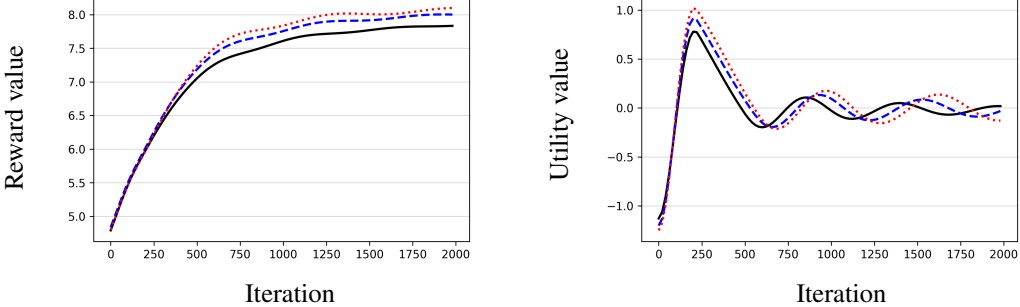

Figure 11: Convergence performance of RPG-PD with regularization parameter $\tau$: ($\tau = 0.1$, —), ($\tau = 0.05$, $--$), ($\tau = 0.01$, $\cdots\cdots$). The horizontal axes represent the policy iterations $\{\pi_t\}_{t \geq 0}$ that are generated by RPG-PD and the vertical axes mean the value functions of the policy iterates $\{\pi_t\}_{t \geq 0}$: reward value $V_r^{\pi_t}(\rho)$ (Left) and utility value $V_g^{\pi_t}(\rho)$ (Right). In this experiment, we fix the same stepsize $\eta = 0.01$ and take the regularization parameter $\tau$ among 0.1, 0.05, and 0.01, and the uniform initial distribution $\rho$.

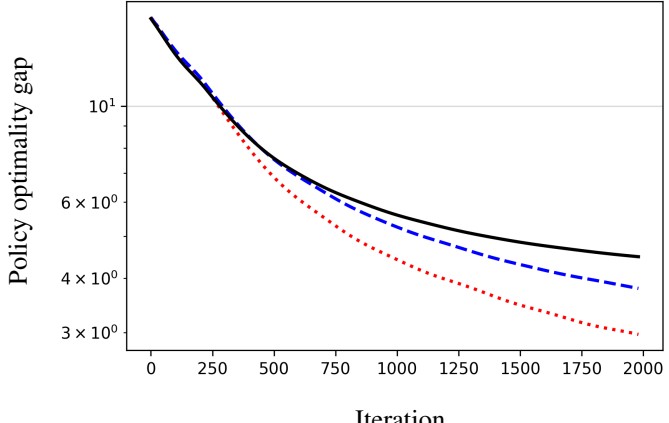

Figure 12: Convergence performance of RPG-PD with regularization parameter $\tau$: ($\tau = 0.1$, —), ($\tau = 0.05$, $--$), ($\tau = 0.01$, $\cdots\cdots$). The horizontal axis represents the policy iterations $\{\pi_t\}_{t \geq 0}$ that are generated by RPG-PD and the vertical axis means the policy optimality gap that measures the distance of the policy iterates $\{\pi_t\}_{t \geq 0}$ to an optimal policy $\pi^\star$: $\sum_s \|\pi_t(\cdot \,|\, s) - \pi^\star(\cdot \,|\, s)\|^2$. In this experiment, we fix the stepsize $\eta = 0.01$ and take the regularization parameter among 0.1, 0.05, and 0.01, and the uniform initial distribution $\rho$.

### E.4 Sensitivity of OPG-PD (9) to stepsize

In this experiment, we use our previous tabular constrained MDP with a random transition, a discount factor $\gamma = 0.9$, uniform rewards $r \in [0,1]$ and utilities $g \in [-1,1]$, and an uniform initial state distribution $\rho$. The constraint is $V_g^\pi(\rho) \geq 0$. We repeat executing OPG-PD by varying three different stepsizes $\eta \in \{0.05, 0.1, 0.2\}$. To demonstrate the optimality of OPG-PD's policy iterates, we measure the policy optimality gap via the squared norm distance of policy iterates to an optimal policy $\pi^\star$ that is obtained from an occupancy-measure-based linear program. To demonstrate the optimality of OPG-PD's policy iterates, we measure the policy optimality gap via the squared norm distance of policy iterates to an optimal policy $\pi^\star$ that is obtained from an occupancy-measure-based linear program. From the policy optimality gap in Figure 2, we see that three policy optimality gaps decrease linearly in the logarithmic scale plot, which verifies the linear last-iterate convergence of

OPG-PD's policy iterates in Theorem 6. Furthermore, in Figure 13 we show the optimality of OPG-PD's policy iterates by plotting the optimality gap $|V_r^{\pi^\star}(\rho) - V_r^{\pi_t}(\rho)|$ and the constraint violation $|V_g^{\pi_t}(\rho)|$, where $V_r^{\pi^\star}(\rho) = 8.16$. We see that the optimality gap and the constraint violation both decay asymptotically in linear rates in spite of some oscillations, and larger stepsize enjoys faster convergence. It is worth mentioning that, these descending oscillations actually show that value functions are approaching the optimal ones. Hence, we have verified the linear last-iterate convergence of OPG-PD's policy iterates in Theorem 6 using a range of constant stepsizes.

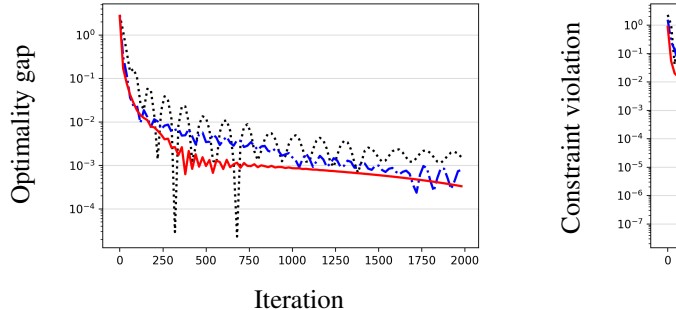 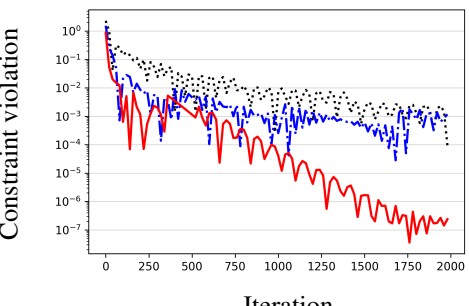

Figure 13: Convergence performance of OPG-PD with stepsize $\eta$: ($\eta = 0.05$, ⋯⋯), ($\eta = 0.1$, -⋅-), ($\eta = 0.2$, —). The horizontal axes represent the policy iterations $\{\pi_t\}_{t \geq 0}$ that are generated by OPG-PD and the vertical axes mean the optimality of the policy iterates $\{\pi_t\}_{t \geq 0}$: optimality gap of reward value $|V_r^{\pi^\star}(\rho) - V_r^{\pi_t}(\rho)|$ (Left) and constraint violation of utility value $|V_g^{\pi_t}(\rho)|$ (Right). In this experiment, we take the stepsize $\eta$ among $0.05$, $0.1$, and $0.2$, and the uniform initial distribution $\rho$.

Last but not least, we extend this experiment for a variant of OPG-PD that is based on the multiplicative weights update (MWU). We call this variant as an optimistic multiplicative weights update primal-dual (OMWU-PD) method which maintains two sequences for policy and dual variables each: $\{\pi_t\}_{t \geq 1}$ and $\{\hat{\pi}_t\}_{t \geq 1}$ for the policy-player, and $\{\lambda_t\}_{t \geq 1}$ and $\{\hat{\lambda}_t\}_{t \geq 1}$ for the dual-player,

$$
\begin{aligned}
\pi_t(\cdot \,|\, s) &= \operatorname*{argmax}_{\pi(\cdot \,|\, s) \in \Delta(A)} \left\{ \sum_a \pi(a \,|\, s) Q_{r + \lambda_{t-1}g}^{\pi_{t-1}}(s, a) - \frac{1}{\eta} \operatorname{KL}(\pi(\cdot \,|\, s), \hat{\pi}_t(\cdot \,|\, s)) \right\} \\
\hat{\pi}_{t+1}(\cdot \,|\, s) &= \operatorname*{argmax}_{\pi(\cdot \,|\, s) \in \Delta(A)} \left\{ \sum_a \pi(a \,|\, s) Q_{r + \lambda_t g}^{\pi_t}(s, a) - \frac{1}{\eta} \operatorname{KL}(\pi(\cdot \,|\, s), \hat{\pi}_t(\cdot \,|\, s)) \right\}
\end{aligned}
\tag{41a}
$$

$$
\begin{aligned}
\lambda_t &= \operatorname*{argmin}_{\lambda \in \Lambda} \left\{ \lambda V_g^{\pi_{t-1}}(\rho) + \frac{1}{2\eta}(\lambda - \hat{\lambda}_t)^2 \right\} \\
\hat{\lambda}_{t+1} &= \operatorname*{argmin}_{\lambda \in \Lambda} \left\{ \lambda V_g^{\pi_t}(\rho) + \frac{1}{2\eta}(\lambda - \hat{\lambda}_t)^2 \right\}
\end{aligned}
\tag{41b}
$$

where $\eta$ is the stepsize and $(\hat{\pi}_0, \hat{\lambda}_0) = (\pi_0, \lambda_0) \in \Pi \times \Lambda$ is the initial point. OMWU-PD concurrently works with two primal iterates and two dual iterates, which is similar to OPG-PD. The only difference is that Primal update (41a) works as the NPG or MWU updates [23, 54], instead of the projected $Q$-ascent [58, 65], which is also different from the one-step optimistic MWU in policy-based ReLOAD [40].

To further investigate the applicability of optimistic gradient methods, we execute OMWU-PD in the same setting and report our result in Figure 14 and Figure 15. We see that the policy optimality gaps decay sublinearly in Figure 14, and the optimality gap and the constraint violation asymptotically decay in sulinear rates in Figure 15. As a comparison, we repeat the same experiment for policy-based ReLOAD [40], which is different from our OMWU-PD in using one-step optimistic gradient updates. In Figure 16 and Figure 17, we observe sublinear decay of policy optimality gap, optimality gap and constraint violation, where are similar as shown in Figure 14 and Figure 15.

The empirical results from two MWU-based optimistic primal-dual algorithms are suggestive. First, MWU-based optimistic algorithms can also have policy last-iterate convergence to an optimal policy. Second, convergence rates of MWU-based optimistic algorithms look slower than OPG-PD's under the same constant stepsize, which we leave as an immediate future work to establish sublinear convergence rates for MWU-based optimistic primal-dual algorithms.

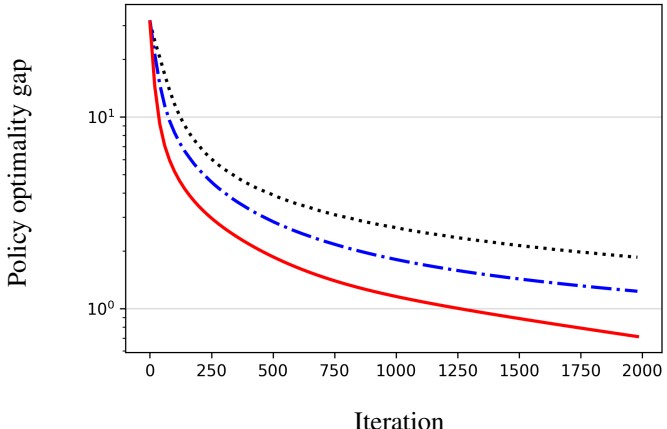

Figure 14: Convergence performance of OMWU-PD with stepsize $\eta$: ($\eta = 0.05$, ⋯⋯), ($\eta = 0.1$, ⎯·⎯), ($\eta = 0.2$, ⎯). The horizontal axis represents the policy iterations $\{\pi_t\}_{t \geq 0}$ that are generated by OMWU-PD and the vertical axis means the policy optimality gap that measures the distance of the policy iterates $\{\pi_t\}_{t \geq 0}$ to an optimal policy $\pi^\star$: $\sum_s \text{KL}(\pi^\star(\cdot \,|\, s), \pi_t(\cdot \,|\, s))$. In this experiment, we take the stepsize $\eta$ among $0.05$, $0.1$, and $0.2$, and the uniform initial distribution $\rho$.

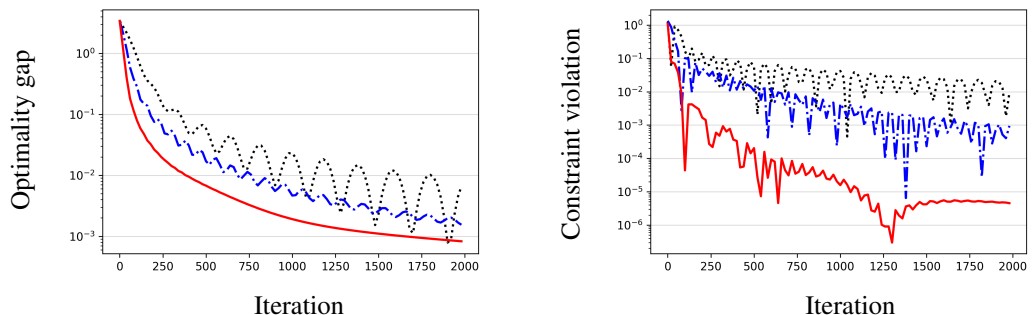

Figure 15: Convergence performance of OMWU-PD with stepsize $\eta$: ($\eta = 0.05$, ⋯⋯), ($\eta = 0.1$, ⎯·⎯), ($\eta = 0.2$, ⎯). The horizontal axes represent the policy iterations $\{\pi_t\}_{t \geq 0}$ that are generated by OMWU-PD and the vertical axes mean the optimality of the policy iterates $\{\pi_t\}_{t \geq 0}$: optimality gap of reward value $|V_r^{\pi^\star}(\rho) - V_r^{\pi_t}(\rho)|$ (Left) and constraint violation of utility value $|V_g^{\pi_t}(\rho)|$ (Right). In this experiment, we take the stepsize $\eta$ among $0.05$, $0.1$, and $0.2$, and the uniform initial distribution $\rho$.

## F  Supporting Lemmas

In this section, we collect some lemmas that are helpful to our analysis.

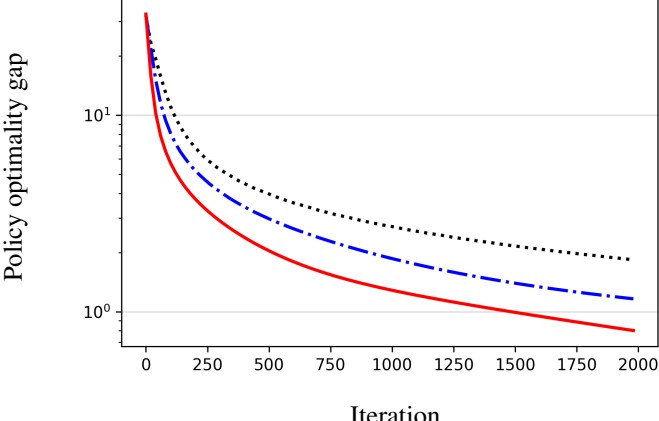

Figure 16: Convergence performance of policy-based ReLOAD [40] with stepsize $\eta$: ($\eta = 0.05$, $\cdots$), ($\eta = 0.1$, $-\cdot-$), ($\eta = 0.2$, —). The horizontal axis represents the policy iterations $\{\pi_t\}_{t \geq 0}$ that are generated by ReLOAD and the vertical axis means the policy optimality gap that measures the distance of the policy iterates $\{\pi_t\}_{t \geq 0}$ to an optimal policy $\pi^\star$: $\sum_s \text{KL}(\pi^\star(\cdot \,|\, s), \pi_t(\cdot \,|\, s))$. In this experiment, we take the stepsize $\eta$ among $0.05$, $0.1$, and $0.2$, and the uniform initial distribution $\rho$.

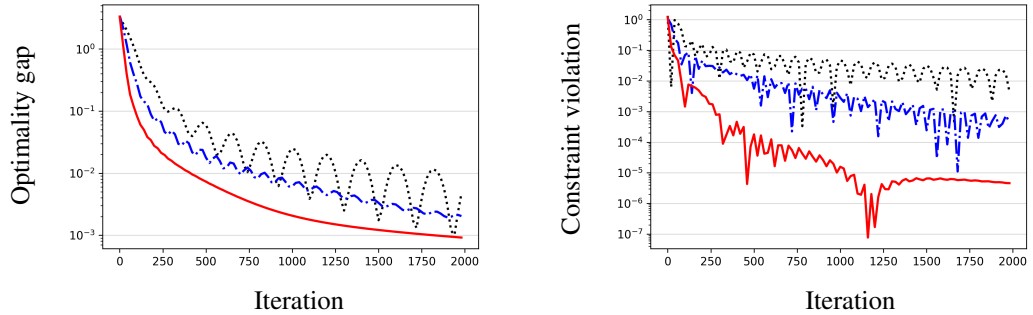

Figure 17: Convergence performance of policy-based ReLOAD [40] with stepsize $\eta$: ($\eta = 0.05$, $\cdots$), ($\eta = 0.1$, $-\cdot-$), ($\eta = 0.2$, —). The horizontal axes represent the policy iterations $\{\pi_t\}_{t \geq 0}$ that are generated by ReLOAD and the vertical axes mean the optimality of the policy iterates $\{\pi_t\}_{t \geq 0}$: optimality gap of reward value $|V_r^{\pi^\star}(\rho) - V_r^{\pi_t}(\rho)|$ (Left) and constraint violation of utility value $|V_g^{\pi_t}(\rho)|$ (Right). In this experiment, we take the stepsize $\eta$ among $0.05$, $0.1$, and $0.2$, and the uniform initial distribution $\rho$.

### F.1 Lemmas in optimization

For any convex differentiable function $\psi\colon X \to \mathbb{R}$, the Bregman divergence of $x, x' \in X$ is given by $D_\psi(x', x) := \psi(x') - \psi(x) - \langle \nabla\psi(x), x' - x \rangle$. When $\psi$ is $\sigma$-strongly convex, $D_\psi(x', x) \geq \frac{\sigma}{2} \|x' - x\|^2$ for any $x', x \in X$. Specifically, when $\psi(x) = \frac{1}{2} \|x\|^2$, $D_\psi(x', x) = \frac{1}{2} \|x' - x\|^2$.

**Lemma 24.** *Let $X$ be a convex set. If $x' = \text{argmin}_{\bar{x} \in X} \langle \bar{x}, g \rangle + D_\psi(\bar{x}, x)$, then for any $x^\star \in X$,*

$$\langle x' - x^\star, g \rangle \leq D_\psi(x^\star, x) - D_\psi(x^\star, x') - D_\psi(x', x).$$

*Proof.* See [38, Lemma 10]. $\qquad\square$

**Lemma 25.** *Assume that $D_\psi(x, x') \geq \frac{1}{2} \|x - x'\|_p^2$ for some $\psi$ and $p \geq 1$. If*

$$x_1 = \underset{\bar{x} \in X}{\text{argmin}} \, \langle \bar{x}, g_1 \rangle + D_\psi(\bar{x}, x) \quad \text{and} \quad x_2 = \underset{\bar{x} \in X}{\text{argmin}} \, \langle \bar{x}, g_2 \rangle + D_\psi(\bar{x}, x)$$

*then*

$$\|x_1 - x_2\|_p \ \leq \ \|g_1 - g_2\|_q$$

*where* $\frac{1}{p} + \frac{1}{q} = 1$.

*Proof.* See [38, Lemma 11]. □

**Lemma 26.** *Let* $X \subset \mathbb{R}^m$ *and* $Y \subset \mathbb{R}^n$ *be polytopes and* $M \in \mathbb{R}^{m \times n}$ *be a matrix. Then, there exists a problem-dependent constant* $c > 0$ *such that*

$$\max_{y' \in Y} x^\top My' - V^\star \ \geq \ c \, \|x - \mathcal{P}_{X^\star}(x)\|$$

$$V^\star - \min_{x' \in X} (x')^\top My \ \geq \ c \, \|y - \mathcal{P}_{Y^\star}(y)\|$$

*where* $V^\star$ *is the game value,*

$$V^\star \ := \ \operatorname*{minimize}_{x \in X} \operatorname*{maximize}_{y \in Y} x^\top My \ = \ \operatorname*{maximize}_{y \in Y} \operatorname*{minimize}_{x \in X} x^\top My$$

*and* $(X^\star, Y^\star)$ *is the set of minimax optimal strategies.*

*Proof.* See [38, Theorem 5]. □

**Lemma 27.** *Let* $X \subseteq \Delta(A)$ *be a convex set and* $g$ *be a bounded vector in* $\mathbb{R}^{|A|}$. *If* $x' = \operatorname{argmin}_{\bar{x} \in X} \langle \bar{x}, g \rangle + \frac{1}{\eta} KL(\bar{x}, x)$, *then for any* $x^\star \in X$,

$$\langle x - x^\star, g \rangle \ \leq \ \frac{KL(x^\star, x) - KL(x^\star, x')}{\eta} + \eta \sum_{a \in A} x_a (g_a)^2$$

*where* $\eta$ *satisfies* $\eta g_a \geq -1$ *for* $a \in A$.

*Proof.* See [132, Theorem 2]. □

### F.2 Properties of policy gradient

**Lemma 28** (Performance difference lemma)**.** *For any two policies* $\pi$ *and* $\pi'$, *and any state* $s_0$,

$$V^\pi(s_0) \ - \ V^{\pi'}(s_0) \ = \ \frac{1}{1-\gamma} \mathbb{E}_{s \sim d_{s_0}^\pi} \mathbb{E}_{a \sim \pi(\cdot \,|\, s)} \left[ A^{\pi'}(s, a) \right]$$

*Proof.* See [49, Lemma 3.2]. □

**Lemma 29** (Regularized PG and NPG under softmax parametrization)**.** *Let an entropy-regularized value function be* $V_\tau^\pi(\rho) := V_r^\pi(\rho) + \tau \mathcal{H}(\pi)$, *and define* $Q_\tau^\pi \colon S \times A \to \mathbb{R}$ *and* $V_\tau^\pi \colon S \to \mathbb{R}$ *via Bellman equations,*

$$Q_\tau^\pi(s, a) \ = \ r(s, a) + \lambda g(s, a) + \gamma \mathbb{E}_{s' \sim P(\cdot \,|\, s, a)} \left[ V_\tau^\pi(s') \right]$$

$$V_\tau^\pi(s) \ = \ \mathbb{E}_{a \sim \pi(\cdot, |\, s)} \left[ -\tau \log \pi(a \,|\, s) + Q_\tau^\pi(s, a) \right].$$

*Let a parametrized policy be* $\pi_\theta$ *for some parameter* $\theta \in \mathbb{R}^d$. *If the policy* $\pi_\theta$ *is differentiable and* $\sum_a \pi_\theta(a \,|\, s) = 1$, *then,*

$$\frac{\partial V_\tau^{\pi_\theta}(\rho)}{\partial \theta_{s,a}} \ = \ \frac{1}{1-\gamma} d_\rho^{\pi_\theta}(s) \cdot \pi_\theta(a \,|\, s) \cdot A_\tau^{\pi_\theta}(s, a)$$

$$= \ \frac{1}{1-\gamma} d_\rho^{\pi_\theta}(s) \cdot \pi_\theta(a \,|\, s) \cdot (Q_\tau^{\pi_\theta}(s, a) - \tau \log \pi_\theta(a \,|\, s))$$

*for all* $(s, a) \in S \times A$, *where* $A_\tau^{\pi_\theta}(s, a) := Q_\tau^{\pi_\theta}(s, a) - \tau \log \pi_\theta(a \,|\, s) - V_\tau^{\pi_\theta}(s)$. *Moreover, if the policy* $\pi_\theta$ *is in form of the softmax function* $\pi_\theta(a \,|\, s) = \frac{\exp(\theta_{s,a})}{\sum_{a'} \exp(\theta_{s,a'})}$ *for all* $(s, a) \in S \times A$, *then*

$$\left[ (F_\rho(\theta))^\dagger \cdot \nabla_\theta V_\tau^{\pi_\theta}(\rho) \right]_{s,a} \ = \ \frac{1}{1-\gamma} A_\tau^{\pi_\theta}(s, a) + c(s)$$

*where* $c(s)$ *is an action-independent constant.*

*Proof.* See [52, Lemma 6]. □

