# OpenReview forum: "Last-Iterate Convergent Policy Gradient Primal-Dual Methods for Constrained MDPs"
_NeurIPS.cc/2023/Conference — NeurIPS 2023 poster_

### Official Review · Reviewer_tcAd · 2023-06-27

**Soundness:** 3 good
**Presentation:** 3 good
**Contribution:** 3 good
**Rating:** 6
**Confidence:** 4

**Summary:**

This paper studies the problem of policy searching for constrained MDPs. The authors devise two Lagrangian-based named regularized policy gradient primal-dual (RPG-PD) method and optimistic policy gradient primal-dual (OPG-PD) method, respectively. Their methods are single-time-scale and thus insensitive to hyperparameter changes. They also show that the proposed methods have the property of last-iterate convergence and give the convergence rate in theoretical analysis.

**Strengths:**

The paper is well written and easy to follow. The related works section is particularly commendable, providing a comprehensive overview of relevant works in this field. The authors are the first to give a non-asymptotic rates for last-iterate convergence of single-timescale primal-dual methods. The theoretical analysis is technically sound. From my view point this paper can be a nice addition to the literature on constrained policy optimization.

**Weaknesses:**

* The convergence of the proposed methods relies on the strong duality property of CMDPs, which may not be true for general parametrized policy classes (like NNs).
* The numerical experiments are only conducted on tabular cases. And I suggest the authors compare their results to more baselines, for example, primal methods like CRPO in [102].
* The paper seems a little bit too long to appear in a conference proceeding. I suggest the author remove some redundant explanations or examples to shorten the paper.

**Questions:**

* The paper only considered constrained MDP with a single constraint as a simplified case. Can the results be generalized to the case of multiple constraints?
* Could the homotopic strategy (gradually shrink the regularization term, see [Li et al. 2022]) be applied to the RPG-PD method? Would that yield better theoretical guarantees?


[Li et al. 2022] Homotopic Policy Mirror Descent: Policy Convergence, Implicit Regularization, and Improved Sample Complexity
Yan Li, Guanghui Lan, Tuo Zhao arXiv:2201.09457

**Limitations:**

See comments above.

---

> ### Author Rebuttal · Authors · 2023-08-09
>
> We thank the reviewer for the time and effort in reviewing our paper, and the valuable feedback. Please find our specific remarks as follows.
>
> ---
>
> ## Weaknesses
>
> > - *The convergence of the proposed methods relies on the strong duality property of CMDPs, which may not be true for general parametrized policy classes (like NNs).*
>
> **Response**: This is an important point. First, strong duality is a `structural property` of constrained MDPs when no function approximation is used, i.e., it can be `proved` under the strict feasibility/Slater condition; see e.g.,  [R1, R2]. Slater condition has been commonly used in the literature to develop convergence theory for constrained MDPs, see e.g., [R2, R3] and many follow-up works.
>
> Second, in the function approximation setting we do not require strong duality in the `parametrized policy class`, and the function approximation error has captured the possible duality error caused by the inexpressiveness of the function class; see also [R3]. Interestingly, we would like to point out that for our regularized method: RPG-PD (see  Equation (8)), the theory (see Theorem 4 and Corollary 5) only assumes the saddle-point property/strong duality for the regularized Lagrangian, without requiring the strong duality for the original un-regularized Lagrangian. Hence, even if the original strong duality fails, when the saddle point of the regularized Lagrangian is close to the solution of the original constrained MDP problem, our regularized method is still applicable.
>
> Thanks for the comment, and we will remark this point in the final version.
>
> [R1] *Constrained Markov Decision Processes. Routledge. 2021.*
>
> [R2] *Safe policies for reinforcement learning via primal-dual methods. TAC 2022.*
>
> [R3] *Convergence and sample complexity of natural policy gradient primal-dual methods for constrained MDPs. arXiv:2206.02346. 2022.*
>
> > - *The numerical experiments are only conducted on tabular cases. And I suggest the authors compare their results to more baselines, for example, primal methods like CRPO in [102].*
>
> **Response**: We have compared RPG-PD and OPG-PD with a primal method CRPO [R1] in Figure 4 in Appendix E.1. To control constraint satisfaction, CRPO switches between the gradient directions of reward/utility value functions depending on the amount of constraint violation. As a result, Figure 4 shows that CRPO reaches a slightly lower reward value than OPG-PD's, and the constraint satisfaction is relatively conservative and has mild oscillation behavior. Hence, we conjecture that the policy last-iterate convergence holds for CRPO, up to some error caused by the switching overhead.
>
> Thanks for the suggestion, and we will check other baselines and add more experiments in the final version.
>
> > - *The paper seems a little bit too long to appear in a conference proceeding. I suggest the author remove some redundant explanations or examples to shorten the paper.*
>
> **Response**: Thanks for the suggestion, and we will shorten the final version according to your suggestion.
>
> ---
>
> ## Questions
>
> > - *The paper only considered constrained MDP with a single constraint as a simplified case. Can the results be generalized to the case of multiple constraints?*
>
> **Response**: We can generalize the constrained saddle-point formulation (see Equation (3)) for constrained MDPs with a finite number of constraints by introducing a vector form of Lagrangian multiplier. Thus, the dual updates of RPG-PD and OPG-PD are in vector form and the last-iterate convergence analysis carries over to this general case. We notice that we are not the first using such a simplification, as it has been used in several other studies, e.g., [R1, R2, R3].
>
> Thanks for the question, and we will remark this point in the final version.
>
> [R1] *Provably efficient model-free constrained rl with linear function approximation.  NeurIPS 2022.*
>
> [R2] *DOPE: Doubly optimistic and pessimistic exploration for safe reinforcement learning. NeurIPS 2022.*
>
> [R3] *Natural policy gradient primal-dual method for constrained markov decision processes. NeurIPS 2020.*
>
> > - *Could the homotopic strategy (gradually shrink the regularization term, see [Li et al. 2022]) be applied to the RPG-PD method? Would that yield better theoretical guarantees?*
>
> **Response**: Thank you for providing this important reference [R1]. We believe that the homotopic strategy can be adopted in our regularized method, and hopefully with better convergence guarantee. We notice two challenges in applying the homotopic strategy: (i) an optimal policy is not necessarily induced by some deterministic optimal policies that usually do not exist for a constrained MDP; (ii) since instability of saddle-point gradient dynamics often results from large stepsize, monotonically increasing stepsizes could make primal-dual algorithms divergent. Hence, additional effort is needed to prove the homotopic strategy for solving constrained MDPs.
>
> Thanks for the suggestion, and we will remark this reference, and include this important future direction in the final version.
>
> [R1] *Homotopic policy mirror descent: Policy convergence, implicit regularization, and improved sample complexity. arXiv:2201.09457. 2022.*
>
> ---
>
> We would like to thank the reviewer again for the helpful comments. Please feel free to let us know if there are any other concerns we can address that could improve your assessment of our work.

---

> > ### Comment · Reviewer_tcAd · 2023-08-13
> >
> > Thanks for the clarification, I choose to retain my positive evaluation of the manuscript.

---

### Official Review · Reviewer_V7nr · 2023-06-30

**Soundness:** 3 good
**Presentation:** 3 good
**Contribution:** 3 good
**Rating:** 7
**Confidence:** 4

**Summary:**

This paper studied the policy gradient primal-dual approach for constraint MDP with last iterate convergence guarantee. A regularized policy gradient primal-dual method is first proposed where regularization on the policy and dual variables are introduced to add curvature to the minimax problem and with appropriately chosen regularization factor, sublinear last-iterate convergence is obtained. An optimistic poicy gradient primal-dual method is then proposed with linear convergence of the squared distance between the last iterate (policy, dual variable) pair and the saddle region under some additional assumptions. The key analysis technique is to bridge the policy update and update in occupancy measure, and CMDP is convex w.r.t. the occupancy measure.

**Strengths:**

The last iterate convergence of algorithm for CMDP is meaningful. The proposed methods with counterparts in minimax optimization are intuitive and explained very well. Multiple varations of CMDP has been studied, i.e., linear function approximation, zero constraint violations. The I went over a majority part of the proofs and have some questions. I will increase my score after the my questions are addressed. The majority of proofs are sound to me.

**Weaknesses:**

I did  not find  perticular weaknesses of the current paper. The ideas are explained clearly and easy to follow. The limitations are also discussed.

**Questions:**

The first inequality in Page 29 between Line 1045 - Line 1046: could you justify the usage of Lemma 27, since \pi^*_\tau may not be the optimal w.r.t. to dual variable \lambda_t?

It seems that there is some order mismatch of the terms (i), (ii) in Eq. (20), where (i) is O(\sqrt{epsilon}) and (ii) is O(epsilon). It is possible to further reduce the sample complexity by carefully selecting \tau?

**Limitations:**

Yes.

---

> ### Author Rebuttal · Authors · 2023-08-09
>
> We thank the reviewer for the time and effort in reviewing our paper, and the valuable feedback. Please find our specific remarks as follows.
>
> ---
> ## Questions
>
> > *The first inequality in Page 29 between Line 1045 - Line 1046: could you justify the usage of Lemma 27, since \pi^*_\tau may not be the optimal w.r.t. to dual variable \lambda_t?*
>
> **Response**: Thank you for raising this important question. We apologize for a typo in Lemma 27: $x'$ on the left-hand side of the inequality should be $x$. We repeat Lemma 27 in a more convenient argmax-form here: if $x' \in argmax_{\bar{x}\in X} \langle \bar{x}, g \rangle - \frac{1}{\eta} \text{KL}(\bar{x},x)$, where $X$ is a probability simplex $\Delta(A)$ and $g$ is a bounded vector in $\mathbb{R}^{|A|}$, then for any $x^\star\in X$, $\langle x^\star - x, g\rangle \leq \frac{1}{\eta} (\text{KL}(x^\star,x)-\text{KL}(x^\star,x'))  + \eta \sum_a x_a (g_a)^2$. Since the explicit form of $x'$ is the standard update of Hedge: $x_a'\propto x_a {\rm e}^{\eta g_a}$, its proof follows from the proof of Theorem 2 in the note [R1] by flipping the sign of the loss.
>
> Application of Lemma 27 to the primal update of RPG-PD per state (see Equation (6a)) can be verified by taking $x' = \pi_{t+1}(\cdot \vert s)$, $\bar{x} = \pi(\cdot \vert s)$, $g =  Q_{r+\lambda_t g +\tau \psi_t}^{\pi_t}(s,\cdot)$, $x = \pi_t(\cdot \vert s)$, and $x^\star = \pi_{\tau}^\star(\cdot \vert s)$.  We notice that the left-hand side of the inequality in Lemma 27 does not require the optimality of $x^\star$. Then, the boundedness of the value function $g$ leads to the inequality between line 1045 and line 1046.
>
> Thanks for the question, and we will remove this typo and explicitize the application of Lemma 27 in the final version.
>
> [R1] Lecture 1, Introduction to Online Optimization/Learning, [Link](https://haipeng-luo.net/courses/CSCI659/2022_fall/lectures/lecture1.pdf)
>
> > *It seems that there is some order mismatch of the terms (i), (ii) in Eq. (20), where (i) is O(\sqrt{epsilon}) and (ii) is O(epsilon). It is possible to further reduce the sample complexity by carefully selecting \tau?*
>
> **Response**: Given a desired accuracy $\epsilon\in(0,1)$, adding two big O notations $O(\epsilon)$ and $O(\sqrt{\epsilon})$ leads to $O(\epsilon) + O(\sqrt{\epsilon}) \leq O(\sqrt{\epsilon})$ when $\epsilon\to 0$, which summarizes the way we combine the terms (i), (ii) in Equation (20).
>
> We note that the regularization parameter $\tau$ has been carefully selected. The KL distance between the policy iterate and the optimal regularized policy is determined by a sum of ${\rm e}^{-\eta \tau t}$ and $\eta/\tau$, where $\eta$ is the stepsize and $\tau$ is the regularization parameter. To ensure $\epsilon$-optimality gap and $\epsilon$-constraint violation, the KL distance must be $O(\epsilon^2)$ and the regularization parameter needs to be $\Theta(\epsilon^2)$  (see Proof of Corollary 3). Hence, the stepsize has to be $\Theta(\epsilon^4)$ and the iteration complexity thus becomes $\Omega  (1/\epsilon^6)$.
>
> ---
>
> We would like to thank the reviewer again for the helpful comments. Please feel free to let us know if there are any other concerns we can address that could improve your assessment of our work.

---

> > ### Author Response · Authors · 2023-08-18
> >
> > Thanks again for providing very positive comments and constructive questions. It would be very much appreciated if you could review our response again. If you have further questions, please feel free to notify us.

---

> > > ### Comment · Reviewer_V7nr · 2023-08-18
> > >
> > > Thanks for addressing my concerns. I keep my positive rating.

---

> > > > ### Author Response · Authors · 2023-08-18
> > > >
> > > > Thank you very much for keeping the positive rating. As we noticed in your previous review that ''I will increase my score after the my questions are addressed'', we were wondering if there are any clarifying issues regarding the questions we could address them better. Many thanks!

---

> > > > > ### Comment · Reviewer_V7nr · 2023-08-18
> > > > >
> > > > > Yes, sure! I am positive about the work and will increase my score.

---

> > > > > > ### Author Response · Authors · 2023-08-18
> > > > > > **Thank you**
> > > > > >
> > > > > > Thank you!

---

### Official Review · Reviewer_NJhb · 2023-07-01

**Soundness:** 4 excellent
**Presentation:** 4 excellent
**Contribution:** 3 good
**Rating:** 7
**Confidence:** 4

**Summary:**

This work shows the first non-asymptotic and policy last-iterate convergence for single-time-scale algorithms in the CMDP literature. In particular, it provides nearly dimension-free sublinear last-iterate policy convergence, sublinear last-iterate policy convergence with function approximation, and problem-dependent linear last-iterate policy convergence.



**Strengths:**

1. This work strengthened the prior works which only guarantee asymptotic last-iterate convergence or value-average or policy-mixture non-asymptotic convergence.
2. It sets up a new framework for analyzing policy-based primal-dual algorithms via the distance of primal-dual iterates to a saddle point.

**Weaknesses:**

1. It is better to give a more formal definition of the "single-time-scale" and "two-time-scale" at the beginning of the paper to help readers better understand the introduction.


**Questions:**

1. Compared with the references [95, 24], if there is any other technical difficulty apart from replacing the convex inequality conditions in constrained convex optimization with the specialty property of CMDP, such as the performance difference lemma?
2. How is the last-iterate converge rate established in this paper compared with the last-iterate converge rates using two-time-scale methods? Are the convergence rates in this paper better? If not, why not, and if it is possible to reduce the gap?
3. For problem (4), why do you directly show that the strong duality holds? Lemma 1 can not be directly applied to the problem (4).
4. In Theorem 6, how strong is the assumption that optimal state visitation distribution is unique given that there may exist multiple optimal policies for CMDP.



**Limitations:**

See questions

---

> ### Author Rebuttal · Authors · 2023-08-09
>
> We thank the reviewer for the time and effort in reviewing our paper, and the valuable feedback. Please find our specific remarks as follows.
>
> ---
> ## Weaknesses
>
> >1. *...  a more formal definition of the "single-time-scale" and "two-time-scale" ... help readers better understand the introduction.*
>
> **Response**: This is an important point. `single-time-scale` refers to the classical gradient-based methods that update multiple iterates, concurrently, using constant stepsize [R1, R2]. `two-time-scale` is from stochastic approximation [R3]: stepsizes for different iterates are relatively large/small (or iterates change relatively fast/slow). We notice that methods with two nested gradient loops are in the `two-time-scale` type, since one iterate waiting for another iteration loop is slower. We will explicitize this notion in the final version.
>
> [R1] *Studies in linear and non-linear programming. Stanford University Press, 1958.*
>
> [R2] *A modification of the Arrow-Hurwicz method for search of saddle points. USSR. 1980.*
>
> [R3] *Stochastic approximation: a dynamical systems viewpoint. Springer. 2009.*
>
> ---
>
> ## Questions
>
> >1. *Compared with the references [95, 24] ... any other technical difficulty apart from replacing the convex inequality conditions in constrained convex optimization with the specialty property of CMDP, such as the performance difference lemma?*
>
> **Response**: Besides handling non-convexity, our first technical contribution, we see other two considerable technical difficulties we have addressed. First, our constrained saddle-point problem is not symmetric: one takes a stochastic policy that affects transition dynamics and the other selects an action in a continuous interval that changes payoff. Our last-iterate analysis works for the asymmetric saddle points, which departs from the symmetric setting [R1, R2]. Second, a saddle-point policy is not uniformly max-min optimal, i.e., being optimal across all states, since an optimal policy depends on the initial state distribution in a constrained MDP. Our duality analysis goes beyond the per-state analysis [R3, R4], which could be of independent interest for analyzing constrained Markov games.
>
> Thanks for the question, and we will emphasize these points in the final version.
>
> [R1] *Tight last-iterate convergence of the extragradient and the optimistic gradient descent-ascent algorithm for constrained monotone variational inequalities. arXiv:2204.09228. 2022.*
>
> [R2] *On linear convergence of iterative methods for the variational inequality problem. JCAM. 1995.*
>
> [R3] *Last-iterate convergence of decentralized optimistic gradient descent/ascent in infinite-horizon competitive Markov games. COLT 2021.*
>
> [R4] *Can We Find Nash Equilibria at a Linear Rate in Markov Games? ICLR 2023.*
>
> >2. *... compared with the last-iterate converge rates using two-time-scale methods? ... better? ... reduce the gap?*
>
> **Response**: The last-iterate rate of RPG-PD is sublinear in time, which is worse than the linear rates of two-time-scale methods [R1, R2]. We notice that RPG-PD converges to a regularized saddle point at a linear rate, up to a neighborhood that is dictated by the stepsize and regularization parameters. The slow rate results from proper parameters that set the neighborhood to be a desired accuracy. In our experiment, RPG-PD converges to the optimal saddle point sublinearly; see Figure 9 in Appendix E.3. So, we conjecture that it is impossible to improve the order of rate without new algorithmic design.
>
> The last-iterate rate of OPG-PD is linear in time, which matches the linear rate of the two-time-scale method [R1] and improves the linear rate in [R2]. We note that our last-iterate convergence captures the stability of primal-dual iterates, while the last policy iterates in [R1, R2] come from NPG subroutines. We also notice that problem-dependent constants occur in all three linear rates, which we leave as future work of uncovering the optimal rate.
>
> Please see more rate comparison in Table 1 in Appendix A. Thanks for the questions, and we will emphasize these points in the final version.
>
> [R1] *A dual approach to constrained markov decision processes with entropy regularization. AISTATS 2022.*
>
> [R2] *Algorithm for constrained Markov decision process with linear convergence. AISTATS 2023.*
>
> >3. *... problem (4), why do you directly show that the strong duality holds? ...*
>
> **Response**: Because the strong duality in Lemma 1 is for the un-regularized problem (see Equation (1)), it is not relevant to the regularized constrained saddle-point problem (see Equation (4)). So, it is crucial to check well-definedness of this regularized saddle-point problem, by showing the existence and uniqueness of a saddle point; see Appendix C.1 for proof.
>
> >4. *...  Theorem 6, how strong is the assumption that optimal state visitation distribution is unique ...*
>
> **Response**: This is an important point. To measure the proximity of primal-dual iterates to a saddle point, we assume the uniqueness of the optimal state visitation distribution to define a distance metric supported by this distribution. Compared with the unique optimal policy, it is a mild assumption, since different optimal policies can share the same state visitation. This can be viewed from the linear program formulation of MDPs: occupancy measures $\{q^{\pi_k^\star}(s,a), k =1,2,\cdots\}$ associated with optimal policies $\{\pi_k^\star, k =1,2,\cdots\}$ share the same state visitation $q(s) = \sum_a q^{\pi_k^\star}(s,a)$ for all $k$. We notice that when we restrict the set of optimal policies $\Pi^\star$ with this property (any such optimal policy induces the same state visitation), our theory (see Theorem 6 and Corollary 7) still holds (or algorithms indeed converge to a `set` of optimal policies that share the same state visitation). Therefore, we believe that this uniqueness assumption is arguably mild. We leave further relaxing this assumption as our immediate future work.

---

> > ### Author Response · Authors · 2023-08-18
> >
> > Thanks again for providing valuable comments and insightful questions. It would be very much appreciated if you could review our response again. If you have further questions, please feel free to notify us.

---

### Official Review · Reviewer_jUv5 · 2023-07-26

**Soundness:** 4 excellent
**Presentation:** 4 excellent
**Contribution:** 4 excellent
**Rating:** 8
**Confidence:** 3

**Summary:**

Two single-timescale algorithms (RPG-PD and OPG-PD) are proposed, and their finite-time convergence rates have been derived. The iteration complexity guarantees for RPG-PD are (nearly) dimension-free but are sublinear. Guarantees for OPG-PD are linear but depend on problem-dependent quantities.



**Strengths:**

1. Last-iterate convergence and constraint satisfaction are useful and generally more challenging. Unlike average and mixture performance measures, these do not hide possible oscillations in objective/constraint functions of immediate policy iterates, which can be undesirable for constrained dynamic systems.

2. Empirical studies showing improved performance over existing algorithms. Most notably -  RPG-PD and OPG-PD suppress oscillation in utility values while achieving optimal reward values.

3. Very thorough literature survey.

**Weaknesses:**

See Questions.

 Minor typos
1. At several places, the references are to the results in the appendix, e.g., Theorem 18 below the statement of Thm. 4.


**Questions:**

1. I could not find any explanations/insights on why the previous approaches lead to oscillations while the newly proposed approaches do not. In particular, why does OPG-PD more or less have no oscillations? Also, in Figure 8, the oscillations are more pronounced for smaller parameter choices. Why does that happen?

2. It is unclear how the proposed ideas can be implemented in practice: they involve (a) computing exact expectations or Monte-carlo estimates from a priori unknown distributions and (b) optimizing certain objective functions over the set of all policy distributions, e.g., 6a, 9a, or 27a. These seem very challenging. Can the authors comment on it?

3. While the authors have mentioned the online version as part of future work, do they expect this version to also have the benefits of damped oscillations?

4. In practice, one often may require that the constraints not be violated with high probability. Would the proposed approach extend to cover that scenario as well?

5. What is `optimistic' about the OPG-PD method? Some insights would be helpful for the reader.

**Limitations:**

Limitations of the proposed techniques have been adequately discussed.

---

> ### Author Rebuttal · Authors · 2023-08-09
>
> We thank the reviewer for the time and effort in reviewing our paper, and the valuable feedback. Please find our specific remarks as follows.
>
> ---
> ## Weaknesses
>
> >1. *Minor typos: At several places, the references are to the results in the appendix, e.g., Theorem 18 below the statement of Thm. 4.*
>
> **Response**: Below the statement of Theorem 4, `Theorem 18` should be `Theorem 4`. We will correct this typo and check all cross-references in the final version.
>
> ---
>
> ## Questions
>
> >1. *... why does OPG-PD more or less have no oscillations? ... in Figure 8, the oscillations are more pronounced for smaller parameter choices. Why does that happen?*
>
> **Response**: A key insight into the oscillation of Lagrangian-based primal-dual algorithms is from the game-theoretic view. Primal/dual iterates in a primal-dual method are two players: max-/min-players are primal/dual updates, respectively. Gradient-based learning dynamics are not necessarily contractive at the stationary points. The simplest case is the cyclic behavior, e.g., iterates cycle in a bilinear game: $\max_x \min_y x^\top y$ (see Figure 2 of [R1]) in which $(0,0)$ is not a contracting point of usual gradient descent-ascent methods.
>
> A key reason why almost no oscillation is in our OPG-PD (see Equation (9)) is that its gradient-based learning dynamics is `contracting` to the set of stationary points. Particularly, our theory (see Theorem 6) shows that the distance from primal-dual iterates to a set of optimal ones decreases to zero, linearly. Due to fast last-iterate convergence, any oscillation is damped exponentially fast, yielding `more or less have no oscillations.'
>
> Figure 8 evaluates the sensitivity of our RPG-PD (see Equation (6)) using different regularizations $\tau \in \{ 0.1, 0.05, 0.01 \}$ and a fixed stepsize $\eta$. Decreasing $\tau$ from $0.1$ to $0.01$ causes the utility value function in Figure 8 (Right) to oscillate more and more. A reason for this is that RPG-PD with small regularization works as an un-regularized primal-dual method. However, oscillation is inherent to naive primal-dual methods, e.g., un-regularized one [R2].
>
> Thanks for the questions, and we will emphasize these points in the final version.
>
> [R1] *ReLOAD: Reinforcement learning with optimistic ascent-descent for last-iterate convergence in constrained MDPs. ICML 2023.*
>
> [R2] *Responsive safety in reinforcement learning by PID Lagrangian methods. ICML 2020.*
>
> >2. *... unclear how the proposed ideas can be implemented in practice ... Can the authors comment on it?*
>
> **Response**: In practice, our primal-dual algorithms: RPG-PD (see Equation (6)) and OPG-PD (see Equation (9)) can be implemented using policy simulators. Executing primal-dual updates in RPG-PD and OPG-PD needs 2 things: (i) estimation of state and state-action value functions of a current policy; and (ii) projection onto a probability simplex. We can implement them efficiently. For instance, an version of RPG-PD with linear function approximation is given in Algorithm 1 (see it in Appendix C.9), where the policy gradient direction is a linear function. Since current policy is from the last update, we simulate it to form unbiased estimates of value functions in Algorithm 2 and Algorithm 3 (see them in Appendix C.9) and the best linearly approximated state-action value function serves as a policy gradient direction.
>
> The policy gradient steps in Equation (6a), Equation (9a), and Equation (27a) can be evaluated explicitly.  Since Equation (6a) and Equation (27a) perform the classical mirror descent step with KL divergence; see their closed-form expressions in [R1]. For Equation (9a), it is the classical mirror descent step with Euclidean distance, i.e., projected gradient, where probability simplex projection can be solved with linear complexity [R2].
>
> Thanks for the questions, and we will emphasize these points in the final version.
>
> [R1] *Mirror descent, Large-Scale Optimization for Data Science,
> 	[Link](https://yuxinchen2020.github.io/ele522_optimization/lectures/mirror_descent.pdf)*
>
> [R2] *Fast projection onto the simplex and the $l_1$ ball. MP 2016.*
>
> >3. *... online version ... do they expect this version to also have the benefits of damped oscillations?*
>
> **Response**: An open question raised by our work is: can we have a policy-based primal-dual algorithm with online exploration and prove its last-iterate convergence? Due to the last-iterate convergence, it would inherit the benefits of damped oscillations. We conjecture that the answer to this question is positive, considering last-iterate results for zero-sum games [R1, R2].
>
> [R1] *Last-Iterate Convergence with Full and Noisy Feedback in Two-Player Zero-Sum Games. AISTATS 2023.*
>
> [R2] *Uncoupled and Convergent Learning in Two-Player Zero-Sum Markov Games. arXiv:2303.02738. 2023.*
>
> >4. *... constraints not be violated with high probability ... extend to cover that scenario ...*
>
> **Response**: Our constrained MDP takes an inequality constraint on a value function. Such a constraint can approximate the feasible region of a high probability constraint, e.g., obstacle-avoidance [R1]. Hence, we can apply our algorithms to high probability constraints. We will remark this in the final version.
>
> [R1] *Safe policies for reinforcement learning via primal-dual methods. IEEE TAC 2022.*
>
> >5. *What is `optimistic' about the OPG-PD method? ...*
>
> **Response**: The notion `optimistic` is from optimization, e.g., [R1]. If we view Equation (9b) as a real policy gradient step that gives a policy $\hat\pi_{t+1}$, then Equation (9a) serves as a prediction step that gives an intermediate policy $\pi_t$. Not policy gradient at $\hat\pi_t$, a real step uses a policy gradient at $\pi_t$ from prediction.  Thus, OPG-PD is `optimistic` on the predicted policy $\pi_t$, as it leads the direction of policy search. We will remark this in the final version.
>
> [R1] *Optimization, learning, and games with predictable sequences. NeurIPS 2013.*

---

> > ### Comment · Reviewer_jUv5 · 2023-08-13
> >
> > Thanks for your response. I choose to retain my positive score.

---

### Decision · Program_Chairs · 2023-09-21

**Decision:**

Accept (poster)

**Comment:**

The paper makes a nice contribution in achieving both non-asymptotic and last-iterate convergence for single-time-scale primal-dual algorithms in solving CMDP. All reviewers are positive about the the paper and agree it should be accepted.